# Geographical controls and anthropogenic impacts on dissolved organic carbon from mountainous rivers: Insights from optical properties and carbon isotopes

Shuai Chen[1], Jun Zhong[2], Lishan Ran[1], Yuanbi Yi[2], Wanfa Wang[3], Zelong Yan[4], Si-liang Li[2,5], Khan M.G. Mostofa[2]

[1]Department of Geography, The University of Hong Kong, Pokfulam Road, Hong Kong, China

[2]Institute of Surface-Earth System Science, School of Earth System Science, Tianjin University, Tianjin, 300072, China

[3]College of Resources and Environmental Engineering, Key Laboratory of Karst Georesources and Environment, Ministry of Education, Guizhou University, Guiyang, 550025, China

[4]School of Environmental Science and Technology, Dalian University of Technology, Dalian, 116081, China

[5]State Key Laboratory of Hydraulic Engineering Simulation and Safety, Tianjin University, Tianjin 300072, China

*Correspondence to*: Jun Zhong (jun.zhong@tju.edu.cn) and Lishan Ran (lsran@hku.hk)

**Abstract.** Mountainous rivers are critical in transporting dissolved organic carbon (DOC) from terrestrial environments to downstream ecosystems. However, how geographical factors and anthropogenic impacts control the composition and export of DOC in mountainous rivers remains largely unclear. Here, we explore DOC dynamics in three subtropical mountainous catchments (i.e., the Yinjiang, Shiqian, and Yuqing catchments) in southwest China which are heavily influenced by anthropogenic activities. Water chemistry, stable and radioactive carbon isotopes of DOC ($\delta^{13}C_{DOC}$ and $\Delta^{14}C_{DOC}$), and optical properties (UV absorbance and fluorescence spectra) were employed to assess the biogeochemical processes and controlling factors on riverine DOC. The radiocarbon ages of DOC in the Yinjiang River varied widely from 928 years before present to modern. Catchments with higher catchment slope gradients and lower annual air temperature were characterized by lower DOC concentrations, enriched $\delta^{13}C_{DOC}$ and $\Delta^{14}C_{DOC}$, and more aromatic dissolved organic matter (DOM), which were opposite to those with gentle catchment slopes and higher temperature. Variabilities in DOC concentrations were also regulated by land use with higher DOC concentrations in urban and agricultural areas. Furthermore, DOM in catchments with a higher proportion of urban and agricultural land uses was less aromatic, less recently produced and exhibited a higher degree of humification and more autochthonous humic-like DOM. This research highlights the significance of incorporating geographical controls on DOC sources and anthropogenic impacts on DOM composition into the understanding of DOC dynamics and quality of DOM in mountainous rivers which are globally abundant.

## 1 Introduction

Dissolved organic carbon (DOC) plays a fundamental role in the riverine carbon cycle with approximately 0.26 Pg ($1Pg = 10^{15}g$) of DOC exported from global rivers to the ocean each year, accounting for more than half of the total organic carbon export (Cai, 2011; Raymond and Spencer, 2015). Owing to continued climate warming and rapid land use changes, it is important to

gain a better understanding of the spatial and temporal dynamics of DOC transport in river systems (Butman et al., 2014; Fasching et al., 2016; Zhong et al., 2021). For example, the elevated temperature has a dominant effect on DOC concentration and dissolved organic matter (DOM) composition by enhancing decomposition and photochemical degradation rates of DOM (Zhou et al., 2018), contributing to significant $CO_2$ emissions from inland waters (Raymond et al., 2013). Additionally, DOM provides energy and nutrient sources for aquatic biota (Findlay et al., 1998), adsorbing heavy metals and organic pollutants (Aiken et al., 2011). Riverine DOC can also restrict in-stream primary production by reducing light penetration and lowering temperature in the water column, thereby serving as an important determinant in shaping the ecological and biogeochemical processes in aquatic environments (Ask et al., 2009). Therefore, disentangling the processes controlling riverine DOC dynamics is crucial for a greater understanding of aquatic ecosystem functioning and the global carbon cycle. Recent advances in spectroscopic techniques, especially the UV-visible spectrophotometry and fluorescence spectroscopy, and widespread application of stable and radioactive carbon isotopes on bulk DOC have provided insights into the composition, source, and age of DOM in freshwater ecosystems (Coble, 1996; Fellman et al., 2010; Marwick et al., 2015; Minor et al., 2014; Sanderman et al., 2009). These new techniques have led to significant improvements in our understanding of the biogeochemical processes of DOC in river systems, which will continue to be effective tools for researchers to gain deeper insights into the riverine carbon cycle.

The biogeochemical processes of DOM in river systems have been extensively studied, which depend largely on the sources and composition of DOM (Toming et al., 2013). Riverine DOM has both internal and external origins, namely autochthonous and allochthonous DOM. Autochthonous DOM is a pool of dead and living microbial and algal biomass that is derived within the aquatic ecosystem (Devesa-Rey and Barral, 2011). Autochthonous DOM is mainly consisted of non-humic substances that are more bioavailable (Toming et al., 2013). In comparison, allochthonous DOM refers to DOM that originates from outside of the aquatic ecosystem and is typically composed of higher plants and soil organic matter (Toming et al., 2013; Zhang et al., 2023). It may also contain organic waste of anthropogenic origin (Ramos et al., 2006; Toming et al., 2013). Consequently, allochthonous DOM is generally characterized by high lignin content and high molecular weight, making it refractory to decomposition (Devesa-Rey and Barral, 2011).

Recent studies have shown that geographical (e.g., elevation and catchment slope) controls on DOC export may also be important for riverine carbon cycling (Connolly et al., 2018; Li Yung Lung et al., 2018). Compared with high-relief catchments, low relief regions with longer water residence time, stronger hydrologic connectivity to rivers, and greater development of wetlands are typically characterized by greater releases of DOC (Harms et al., 2016; McGuire et al., 2005). A recent global study on lakes and rivers found that increasing elevation is associated with greater protein-like fluorescence DOM and lower specific ultraviolet absorbance at 254 nm ($SUVA_{254}$), which indicate the effect of enhanced UV radiation and accumulation of autochthonous DOM in higher elevation areas with low temperatures (Zhou et al., 2018). More specifically, DOC supply is

likely regulated by the amount of stored soil organic carbon (SOC) in a catchment (Lee et al., 2019; Rawlins et al., 2021). However, this supply is limited by shallow soil depth and high water flow velocity (Lee et al., 2019). In addition, the varying extent of hydrologic connectivity due to changing water residence time with different catchment slopes may have significant influences on DOC dynamics (Connolly et al., 2018). Typically, it is anticipated that as the slope increases towards higher elevation areas, where residence time is relatively short and soil organic matter is well-connected to hydrologic pathways, the composition of DOM pools in inland waters will shift towards a more "terrestrial" character, comprising of larger molecules with high molecular weight and aromatic structures (Creed et al., 2018; Xenopoulos et al., 2021). Although geographical characteristics have proved to be useful in estimating DOC concentrations (Harms et al., 2016; Mzobe et al., 2020), the underlying mechanisms that regulate DOC dynamics in small mountainous rivers remain poorly understood. Therefore, a deep understanding of the geographical controls on DOC dynamics is urgently needed. Subtropical small mountainous rivers are characterized by steep catchment slopes, high erosion rates, frequent rainfall events in wet seasons, and rapid change in hydrology during these rainfall events (Lee et al., 2019; Leithold et al., 2006; Qiao et al., 2019), yet have received little research attention regarding their DOC dynamics. Moreover, runoff, catchment slope gradient, and SOC have been recognized as good predictors for DOC export in small mountainous rivers (Lee et al., 2019). Yet, the extent to which these factors, along with land use patterns, effectively regulate the DOC dynamic is still far from well-understood (Lee et al., 2019; Moyer et al., 2013).

Anthropogenic impacts, such as urban and agricultural land uses, have led to significant alterations to the flux of DOC and the fate and quality of DOM in global streams and rivers (Coble et al., 2022; Pilla et al., 2022; Wilson and Xenopoulos, 2008; Xenopoulos et al., 2021). Agricultural streams and rivers are dominated by microbial-derived, protein-like DOM, while urban freshwater ecosystems are characterized by microbial, humic-like or protein-like, and autochthonous DOM (Hosen et al., 2014; Williams et al., 2016; Xenopoulos et al., 2021). Agricultural and urban land uses tend to increase nutrient loading in streams, resulting in enhanced bacterial production and DOM decomposition (Quinton et al., 2010; Williams et al., 2010). Therefore, microbial-derived DOM plays a crucial role in agricultural and urban rivers. In addition, DOM tends to have a more reduced redox state and is likely more labile and accessible to the microbial community in agricultural streams when compared to the DOM found in natural streams (Williams et al., 2010). Although the DOM in urban rivers shares some similarities with agricultural rivers (such as microbial origins), the sources of DOM in urban rivers are much more complex, which may originate from urban point-source inputs (e.g., wastewater treatment facilities) and nonpoint source inputs (e.g., household sewage and petroleum-based hydrocarbons) (Hosen et al., 2014). On the scale of years to decades, anthropogenic impacts can accelerate terrestrially sourced DOC export to aquatic ecosystems (Xenopoulos et al., 2021). On the scale of decades to centuries, however, anthropogenic impacts would shift natural DOM to forms of low-molecular weight, enhanced redox state with potentially increased lability, or increased aromaticity due to warmer climate and altered hydrology (Stanley et al., 2012; Xenopoulos et al., 2021). In addition, a warmer climate can enhance microbial activity and in-stream production of DOM (He

et al., 2016), and may simultaneously increase the microbial degradation rate of soil DOM, thus reducing the potential input of DOM into rivers (Voss et al., 2015). Consequently, this will increase the relative contribution of autochthonous DOM. Nevertheless, elevated temperature can also enhance photo-chemical degradation and reduce autochthonous microbial humic-like DOM (Henderson et al., 2009; Zhou et al., 2018), thus potentially limiting the accumulation of autochthonous DOM in inland waters. These two seemingly contradictory findings are due to the complex effect of temperature on organic matter. Clearly, it remains largely unknown how these impacts have regulated riverine DOC dynamics due to their complex regulating mechanisms and changing influencing factors.

In this study, we evaluated how geographical controls (i.e., mean catchment slope, mean drainage elevation, and annual air temperature) and anthropogenic impacts (i.e., land use patterns) affect the DOC dynamics and DOM characteristics in three subtropical catchments that contain many small to medium mountainous rivers in southwest China. Our prior observations from these catchments showed that particulate organic carbon (POC) and dissolved inorganic carbon (DIC) dynamics were highly affected by in-stream photosynthesis, as evidenced by stable carbon isotope and radioactive carbon isotope of POC and DIC (Chen et al., 2021). We hypothesize that catchments with a higher proportion of agricultural and urban land use, more gentle catchment slope, and lower elevation would exhibit higher riverine DOC concentrations and more autochthonous microbial humic-like DOM than steeper catchments at high elevations with fewer influences by agricultural and urban land uses. Relationships of DOC concentrations, stable isotopic values of DOC, DOM quality assessed through optical metric, nutrient concentrations, and land use patterns versus geographical characteristics (i.e., mean catchment slope, mean drainage elevation, and annual air temperature) were examined. We also examined relationships between geographical characteristics and radiocarbon for nine sampling sites in the Yinjiang River. This study allows us to gain a deeper insight into the geographical controls and anthropogenic impacts on DOC dynamics and DOM quality in the subtropical, anthropogenically influenced mountainous rivers.

## 2 Materials and Methods

### 2.1 Study area

Yinjiang River (Y), Shiqian River (S), and Yuqing River (Q) are tributaries of the Wujiang River (Fig. 1a), the largest tributary on the south bank of the upper Changjiang River. The drainage area is 1231, 2101, and 1561 $km^2$ for the Yinjiang, Shiqian, and Yuqing rivers, respectively. Data on land use types and air temperature in 2015, as well as a 90 m digital elevation model (Shuttle Radar Topography Mission, SRTM) were obtained from the Resource and Environment Data Cloud Platform of the Chinese Academy of Sciences (http://www.resdc.cn/). Information on dams was retrieved from Wang et al. (2022), and their location was identified by Google Earth. Furthermore, the distance from the river mouth (i.e., the Yinjiang, Shiqian, and Yuqing rivers) to the sampling sites was also estimated using Google Earth. We further delineated the sub-catchments, which constitute

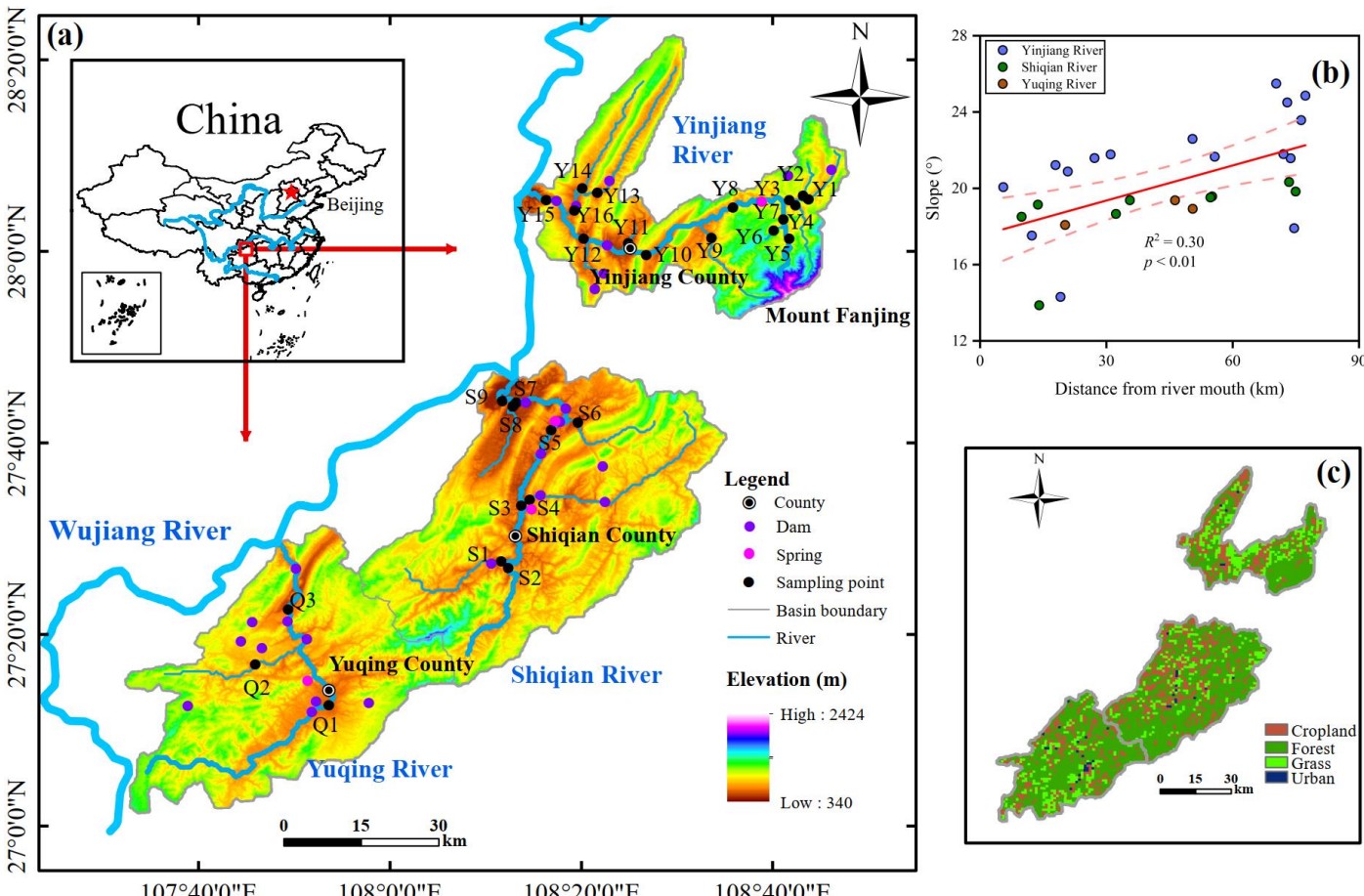

Figure 1. Map of the study area. (**a**) Overview of the sampling sites and elevation characteristics in the three study catchments, including the Yinjiang, Shiqian, and Yuqing catchments, (**b**) correlation between mean catchment slope and the distance from the river mouth (i.e., the Yinjiang, Shiqian, and Yuqing rivers) to the sampling site, and (**c**) spatial variation in land-use patterns.

the contributing area upstream of sampling sites, by spatial analyst tools of ArcGIS (version 10.2). The mean catchment slope (degrees; 3D analysis tools) and elevation for sub-catchments were extracted from the digital elevation model using ArcGIS. Annual air temperature, catchment slope, and proportion of urban and agricultural land uses for these sub-catchments were also determined using ArcGIS. The mean drainage elevation of these three catchments ranges from 340 m to 2424 m with the lowest and highest elevation both reported in the Yinjiang River catchment, showing the greatest change in relief (Figs. 1a and S1a). Similar to elevation, the Yinjiang River catchment has a greater variation in mean catchment slope (from 14.3° to 25.5°), while the Shiqian and Yuqing river catchments have a mean catchment slope of approximately 20°, except the segment above site S8 (13.9°; Figs. 1b and S1b). Carbonate rock is widely distributed in the three catchments, accounting for a large proportion of the exposed strata (Han and Liu, 2004). The remaining areas are mainly covered by clastic rocks, igneous rocks, and low-grade metamorphic rocks. Forest, agriculture, and urban areas are the three dominant land uses in these studied catchments (Fig. 1c). Forest is generally distributed in high-elevation regions, while urban and agricultural land uses are mainly located in low-elevation regions. The proportion of urban and agricultural land uses in the Yuqing River catchment varies from 17.3% to

23.1% (Figs. 1c and S1c). This catchment has higher % urban/agriculture land uses than other studied catchments, and less variability in land use compared to the Yinjiang and Shiqian river catchments (from 4.5% to 46.5% and from 9.6% to 41.3%, respectively). There are three mountainous agricultural counties (i.e., Yinjiang, Shiqian, and Yuqing; Fig. 1a) in this study area, where crops are mainly C4 (e.g., corn and sorghum) and C3 (e.g., rice, wheat, and potato) plants. Dams and reservoirs are

145 widely distributed in the three catchments, and these dams are primarily used for agricultural irrigation and power generation (Fig. 1a). This study area is highly affected by monsoon-influenced humid subtropical climate with April to October being the rainy season, and the average annual precipitation and discharge are 1100 mm and 14.4 $m^3$/s, respectively, in the Yinjiang River catchment. Further details on regional setting of the study area and the sources and methods for catchment characteristics delineation are provided in our previous study (Chen et al., 2021).

**2.2 Field sampling**

Surface water samples (n = 28) along the mainstem and major tributaries of the Yinjiang River, Shiqian River, and Yuqing River and spring water samples (n = 4) were collected in September 2018 (Fig. 1a). During the sampling period, two water samples (sites Y12 and Y15) were significantly affected by rainfall events, and an additional sample was collected at site Y12 before the rainfall event as it is close to the hydrological station. Unless stated otherwise, the data used in this study from site

Y12 are based on the sample collected after rainfall event due to the availability of carbon isotopes. Electrical conductivity (EC) and dissolved oxygen (DO) were measured by a multi-parameter water quality probe (WTW, pH 3630/Cond 3630, Germany) in the field. For the analysis of ion concentrations, total phosphorus (TP), ammonium ($NH_4^+$-N), and total nitrogen (TN), water samples were filtered through 0.45 μm cellulose acetate membranes. Water samples for the concentrations and isotopes of DOC and DOM absorbance and fluorescence were filtered through pre-combusted glass fibre filters (Whatman,

0.7 μm). The filtered water was stored in a Milli-Q water and sampling water pre-washed brand-new low-density polyethylene container at low temperature (4℃) in the dark before optical properties analysis and acidified by phosphoric acid to pH = 2 for DOC analysis. Water samples were also filtered for determining dissolved inorganic carbon (DIC; through 0.45 μm cellulose acetate membranes) through titration with hydrochloric acid and analyzing POC using retained suspended particles on the filter membranes. The water samples filtered through 0.22 μm cellulose-acetate filter membranes were used to determine

water isotopes ($\delta^{18}O$ and $\delta D$). Detailed information on sampling methods was provided in Chen et al. (2021) and Zhong et al. (2020).

**2.3 Laboratory analysis**

The main cations ($K^+$, $Na^+$, $Ca^{2+}$, and $Mg^{2+}$) were measured by inductively coupled plasma emission spectrometer (ICP-OES), and the main anions ($Cl^-$, $SO_4^{2-}$, and $NO_3^-$) were measured by ion chromatography (Thermo Aquion; Chen et al., 2020). The

170 normalized inorganic charge balance is within 5%, indicating the accuracy of the measured data. The concentrations of $NH_4^+$-

N were analyzed using an automatic flow analyzer (Skalar Sans Plus Systems), and the relative deviations of the results of $NH_4^+$-N and were less than 5%. DOC concentrations were determined with a total organic carbon analyser (OI Analytical, Aurora 1030W, USA) with duplicates (±1.5%, analytical error) and a detection limit at 0.01 mg $L^{-1}$. Water isotopes were measured by a Liquid Water Isotope Analyzer (Picarro L2140-i, USA) with measurement precisions at ± 0.3 ‰ for $\delta^{18}O$. The above analyses were carried out at the Institute of Surface Earth System Science, Tianjin University.

For the determination of stable carbon isotope and radiocarbon isotope of DOC ($\delta^{13}C_{DOC}$ and $\Delta^{14}C_{DOC}$), water samples were first concentrated using a rotary evaporation and then oxidized through the wet oxidation method (Leonard et al., 2013). In this study, nine water samples collected from the Yinjiang River were selected for $\Delta^{14}C_{DOC}$ analysis as the Yinjiang River catchment has the greatest change in geographical characteristics (i.e., elevation and catchment slope) and the highest proportion of agricultural and urban land uses among the three catchments. The generated $CO_2$ was purified in a vacuum system for $\delta^{13}C_{DOC}$ and $\Delta^{14}C_{DOC}$ analyses, respectively. $\delta^{13}C_{DOC}$ was directly determined by the MAT 253 mass spectrometer with an analysis accuracy of ±0.1 ‰. For $\Delta^{14}C_{DOC}$ analysis, the purified $CO_2$ was transformed into graphite following the same method of $\Delta^{14}C_{POC}$ analysis (Chen et al., 2021) and measured by an accelerator mass spectrometry (AMS) system within 24 hours with an analytical error of ±3 ‰ (Dong et al., 2018).

Optical analyses on DOM were conducted on river samples. DOM absorbance of river water samples was measured from 250 to 750 nm using a UV (ultraviolet)-visible spectrophotometer (UV-2700, Shimadzu) with a 1 cm quartz cuvette. The UV-visible spectrophotometer was blanked with Milli-Q water prior to data collection. Decadic absorbance values were used to calculate absorption coefficients as below (Poulin et al., 2014):

$$a_{254} = Abs_{254}/L, \tag{1}$$

Where, $a_{254}$ is the absorption coefficients ($cm^{-1}$), $Abs_{254}$ is the absorbance at 254 nm, and L represents the path length (cm). Specific UV absorbance at 254 nm ($SUVA_{254}$; reported in units of L mg $C^{-1}$ $m^{-1}$) was determined according to Weishaar et al. (2003; Table 1):

$$SUVA_{254} = a_{254}/DOC. \tag{2}$$

DOM fluorescence was determined with a fluorescence spectrophotometer (F-7000, Hitachi, Japan) to quantify humic-like, fulvic-like, and protein-like fluorescences (Fellman et al., 2010). Refrigerated water samples for DOM absorbance and fluorescence were analyzed within one week after sampling. The fate of humic-like fluorescences may be self-assembly particles or be adsorbed onto minerals, while protein-like fluorescences are tightly associated with biological processes, and biodegraded into inorganic matter (Fellman et al., 2010; He et al., 2016). The excitation wavelengths ranged from 220 to 400 nm at 5 nm increments and emission wavelength from 280 to 500 nm at 2 nm increments. Blanks were measured daily with the same settings to correct excitation-emission matrices (EEMs). Parallel factor analysis (PARAFAC) was performed using N-way toolbox in Matlab (MathWorks, USA) to determine peaks (Andersson and Bro, 2000; Mostofa et al., 2019; Stedmon

and Bro, 2008). Detailed procedures and criteria for application and validation of PARAFAC model are available in Yi et al. (2021). Identified PARAFAC model components were further compared with relevant published and reported fluorophores in the OpenFluor database (Table 1; Murphy et al., 2014). Several common indices of DOM composition were determined from EEMs, including fluorescence index (FI; McKnight et al., 2001), humification index (HIX; Ohno, 2002), and freshness index ($\beta/\alpha$; Parlanti et al., 2000; Table 2).

**Table 1.** Description of the three components identified by PARAFAC and comparison with previous studies from the OpenFluor database with a minimum similarity score of 0.95 (Murphy et al., 2014).

| Component | Ex$_{max}$ (nm) | Em$_{max}$ (nm) | Description and likely structure | Number of matches in Openfluor | Previous studies |
|---|---|---|---|---|---|
| C1 | 295 | 402 | Similar to traditionally defined peak M, marine humic-like component, the products from microbial processes or autochthonous production. | 6 | C6 (Walker et al., 2009); C4 (Kim et al., 2022); C4 (Li et al., 2016) |
| C2 | 275 | 338 | Protein-like (Tryptophan-like) components, commonly found in anthropogenically affected rivers, associated with recent biological production and breakdown products of lignin. | 30 | C3 (DeFrancesco and Guéguen, 2021); C7 (Lambert et al., 2017); C2 (Du et al., 2019) |
| C3 | 325 | 440 | Traditional fulvic-like peaks A and C, humic-like and terrestrial delivered OM, authochthonous, or microbial source. | 70 | C1 (Amaral et al., 2016); C1 (Ryan et al., 2022); C1 (Shutova et al., 2014) |

## 2.4. Statistical analysis

Normality of the data was first examined by a Shapiro-Wilk test using SPSS 26. Normally distributed data were analyzed by one-way ANOVA with Tukey's post-test for multiple comparisons. Nonparametric data with three or more comparisons were made by Kruskal–Wallis test followed by Holm's Stepdown Bonferroni correction. The Mann–Whitney U test was used for comparison of distributions between two groups. Linear regression was applied using OriginPro 2021 (student version) to quantify the relationship of DOC concentrations, DOM properties, carbon isotopes, and ion concentrations versus catchment characteristics (i.e., mean catchment slope, proportion of different land uses, mean annual air temperature, and mean drainage elevation) to identify the predominant influencing factors on DOC dynamics. Moreover, the correlation among water chemistry and catchment characteristics was computed by Pearson's correlation coefficients (R) by OriginPro 2021 (student version). Values are presented as the mean ± standard deviation. All statistical tests were performed at 0.05 significance level. In addition,

all the statistical analyses were performed again after data from site Y12 were removed to test the possible skew of findings as the sample was significantly affected by rainfall events. If not mentioned otherwise, the results from site Y12 did not skew the findings at the significance level of 0.05.

**Table 2.** DOM optical parameters used in this study.

| Index Name | Calculation | Description | Reference |
|---|---|---|---|
| SUVA$_{254}$ | SUVA$_{254}$ = a$_{254}$/DOC concentration. a$_{254}$ is the decadic UV absorbance at 254 nm. | An indicator for the degree of aromaticity. It is positively correlated with aromaticity. | Weishaar et al. (2003) |
| Fluorescence index (FI) | FI = Em450/Em500, at Ex 370 nm. | A proxy for DOM source. Higher values (~1.9) associated with microbial source and lower values (~1.4) correlated with terrestrial source. | McKnight et al. (2001) |
| Humification index (HIX) | HIX = $\sum$435-480/($\sum$300-345+$\sum$435-480), at Ex 254 nm. | Indicator of humification status of DOM. Higher HIX values indicate an increasing degree of humification. | Ohno (2002) |
| Freshness index (β/α) | β/α = Em380 (β) / the Em intensity maximum between 420 and 435 nm at Ex 310 nm (α). | Higher β/α values are commonly associated with increasing contribution of recently microbially produced DOM. | Parlanti et al. (2000) |

# 3 Results

## 3.1 Spatial variations in water chemistry, DOC concentrations, and isotopes of DOC

The average DO presented similar values between the Yinjiang River, Shiqian River, Yuqing River, and springs with the majority of the river water samples being DO supersaturated (i.e., higher than 100%; Fig. 2a). A strong positive correlation was found between EC and $\delta^{18}O$ for the river water and spring water ($p < 0.001$; Fig. S2a), and the $\delta^{18}O$ showed an increasing trend from upstream to downstream in the Yinjiang River (Fig. S2b). The Cl$^-$ concentration showed an increasing trend in the Yinjiang, Shiqian, and Yuqing rivers, with an average of 2.56 ± 1.03 mg L$^{-1}$, 3.76 ± 0.83 mg L$^{-1}$, and 4.55 ± 0.81 mg L$^{-1}$, respectively (Fig. 2b). In addition, the Cl$^-$ concentration in the spring water (4.48 ± 2.08 mg L$^{-1}$) was significantly higher than that in the Yinjiang River ($p < 0.05$; Fig. 2b). Within the rivers and springs, the water displayed similar NH$_4^+$-N concentrations with the mean value at 0.04 ± 0.03 mg L$^{-1}$, 0.07 ± 0.05 mg L$^{-1}$, 0.04 ± 0.03 mg L$^{-1}$, and 0.03 ± 0.04 mg L$^{-1}$ in the Yinjiang,

Shiqian, Yuqing rivers, and spring water (Fig. 2c). In springs, the average $NO_3^--N$ concentration was $1.93 \pm 0.93$ mg $L^{-1}$, higher than the average in the three rivers ($1.15 \pm 0.36$ mg $L^{-1}$), though there were no significant differences for the overall $NO_3^--N$ concentration between the rivers and springs ($p > 0.05$; Figs. 2d).

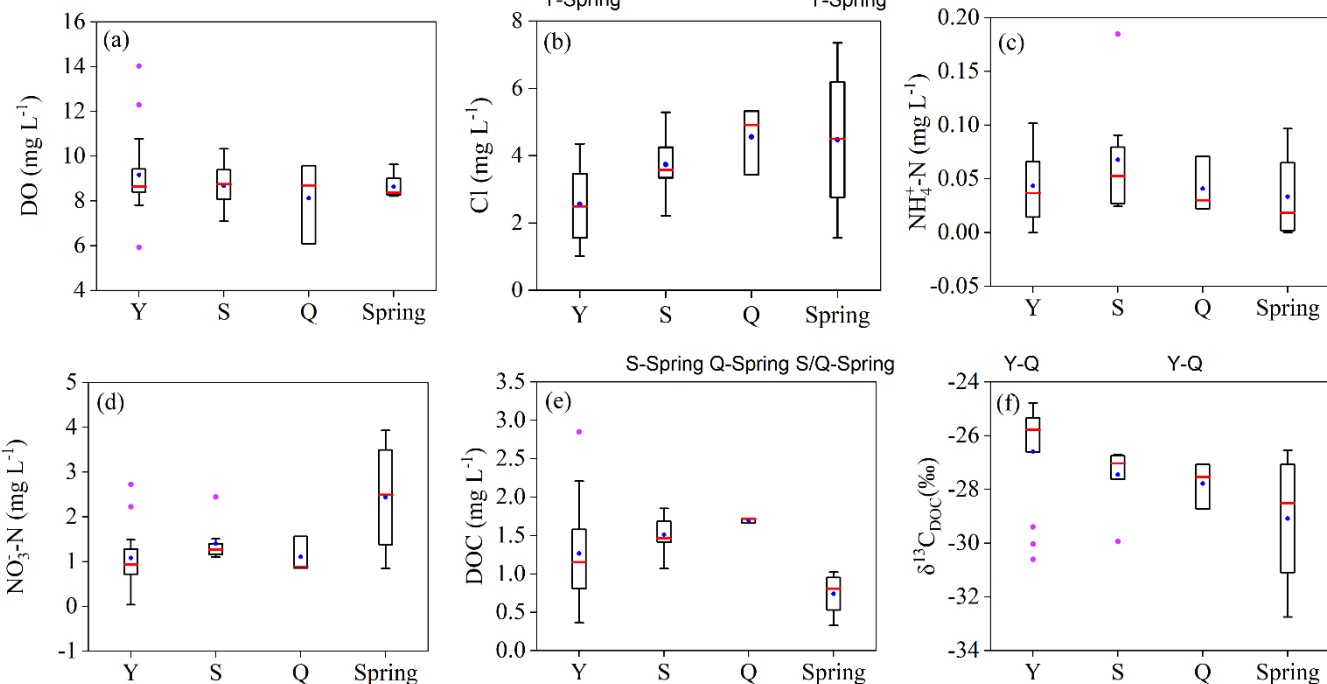

**Figure 2.** Spatial variations in water chemistry in the Yinjiang (Y), Shiqian (S), and Yuqing (Q) rivers and springs. (**a**) DO, (**b**) $Cl^-$, (**c**) $NH_4^+-N$, (**d**) $NO_3^--N$, (**e**) DOC, and (**f**) $\delta^{13}C_{DOC}$. In each box plot, the end of the box represents the 25th and 75th percentiles, the blue solid dot represents the mean, the horizontal line inside the box represents the median, and whiskers represent 1.5 times the upper and lower interquartile ranges (IQR). The magenta solid dot represents the outlier (data points outside of the 1.5 interquartile ranges). Letters above the boxes represent significant differences between the grouping of river and/or spring water based on statistical analyses at the significance level of 0.05 (e.g., Y-Spring above panel (b) indicates that the $Cl^-$ in river water of the Yinjiang River was significantly different from that in the spring water).

DOC concentrations in the three study rivers varied from 0.36 to 2.85 mg $L^{-1}$ with the highest mean concentration in the Yuqing River ($1.70 \pm 0.04$ mg $L^{-1}$; Fig. 2e), followed by the Shiqian River ($1.51 \pm 0.22$ mg $L^{-1}$) and the Yinjiang River ($1.27 \pm 0.66$ mg $L^{-1}$). The DOC concentrations in spring water were significantly lower than that in the surface water of the Shiqian and Yuqing rivers ($p < 0.05$; Fig. 2e), and the average DOC concentration in spring water ($0.74 \pm 0.30$ mg $L^{-1}$) was also lower than the average DOC concentration in the Yinjiang River, indicating there must be other sources of DOC besides groundwater.

For $\delta^{13}C_{DOC}$, although the average $\delta^{13}C_{DOC}$ values showed a decreasing trend in the Yinjiang River, Shiqian River, Yuqing River, and springs, averaging at $-26.6 \pm 1.8$ ‰, $-27.5 \pm 1.1$ ‰, $-27.8 \pm 0.9$ ‰, and $-29.1 \pm 2.7$ ‰, respectively, there were no statistically significant differences on the overall $\delta^{13}C_{DOC}$ values between the three rivers and springs ($p > 0.05$; Fig. 2f). The $\Delta^{14}C_{DOC}$ of the Yinjiang River varied widely from $-109$ ‰ to 33 ‰ with an average of $-54.7 \pm 39.9$ ‰ (Table 3). The radiocarbon ages of the DOC ranged from 928 years BP (i.e., before present) to present, and the youngest $\Delta^{14}C_{DOC}$ (33.3 ‰)

 was found at site Y12.

**Table 3.** $\Delta^{14}C_{DOC}$ and age of DOC in the Yinjiang River.

| River | Samples | $\Delta^{14}C_{DOC}$ (‰) | DOC-Age (yr BP) | SD of DOC-Age (yr BP) |
|-------|---------|--------------------------|-----------------|-----------------------|
| | Y1 | -92 | 774 | 25 |
| | Y2 | -74 | 616 | 23 |
| | Y3 | -52 | 430 | 27 |
| | Y5 | -40 | 326 | 27 |
| Yinjiang | Y9 | -59 | 491 | 27 |
| River | Y11 | -51 | 417 | 27 |
| | Y12 | 33 | Modern | 28 |
| | Y13 | -49 | 401 | 24 |
| | Y14 | -109 | 928 | 28 |

### 3.2. Riverine DOM Optical Properties

Two humic-like fluorescence components (C1 and C3) and one protein-like fluorescence component (C2) were identified by PARAFAC model in these three rivers (Fig. S3; Table 1). Component C1 is similar to traditionally defined peak M and sourced form microbial processes or autochthonous production (Kim et al., 2022; Li et al., 2016; Walker et al., 2009). Component C2 was previously related to recent biological production (DeFrancesco and Guéguen, 2021; Du et al., 2019; Lambert et al., 2017). C3 was the most widely found component in previous research among three fluorescence components and was identified as traditional fulvic-like peaks A and C, representing terrestrial delivered OM or autochthonous microbial sourced OM (Amaral et al., 2016; Ryan et al., 2022; Shutova et al., 2014). Although C1 and C2 varied more widely in the Yinjiang River compared with the Shiqian and Yuqing rivers, the two fluorescence components did not show a statistical difference among the three rivers ($p > 0.05$; Figs. S3a and b). However, a greater proportion of C3 was found in the Shiqian River, exhibiting a distinctive signature compared with the Yinjiang River (Fig. S3c). The proportion of C3 did not show any significant difference between the Yuqing River and the other two rivers (the Yinjiang and Shiqian rivers).

The average $SUVA_{254}$ were $3.3 \pm 1.1$, $3.1 \pm 1.8$, and $2.8 \pm 0.3$ L mg$^{-1}$ m$^{-1}$ in the Yinjiang, Shiqian, and Yuqing rivers, respectively, without significant spatial differences across the three rivers ($p > 0.05$; Fig. 3a). For the fluorescence indexes, the overall fluorescence property did not vary significantly among the three rivers ($p > 0.05$; Figs. 3b, c, and d). FI varied

in a narrow range compared with β/α and HIX. FI of DOM ranged from 1.66 to 1.94, averaging 1.78 (Fig. 3d), indicating

a mixture of DOM of terrestrial and microbial origins. In comparison, β/α varied from 0.70 to 1.22 (Fig. 3b) and HIX

varied from 0.33 to 0.65 (Fig. 3c), with greater variability among the three rivers.

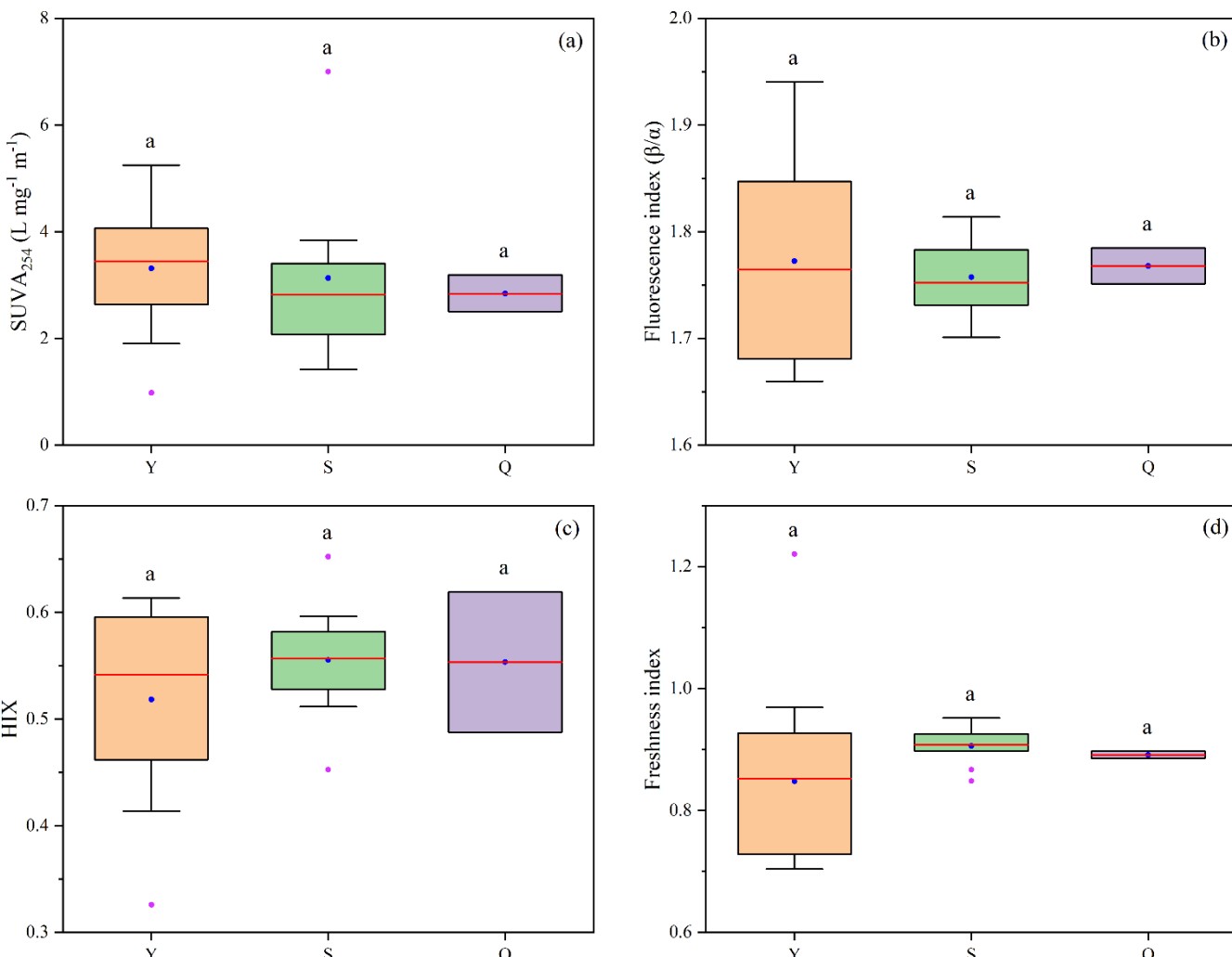

**Figure 3** Spatial variations in DOM property in the Yinjiang (Y), Shiqian (S), and Yuqing (Q) catchments. (a) SUVA$_{254}$, (b) freshness index
(β/α), (c) HIX, and (d) fluorescence index. In each box plot, the end of the box represents the 25th and 75th percentiles, the blue solid dot
represents the average, the horizontal red line represents the median, and whiskers represent 1.5 IQR. The magenta solid dot represents the
outlier, which is outside of the 1.5 interquartile ranges. Different lowercase letters above the boxes denote significant differences across
rivers based on statistical analysis with $p < 0.05$.

**3.3. Factors influencing on DOC**

Significant pairwise interdependencies between DOC and catchment characteristics were identified in the three study rivers

(Fig. S4). There is a strong negative correlation between DOC and average catchment slope ($p < 0.01$; Fig. 4a). Conversely,

the proportion of urban and agricultural land uses displayed a positive correlation with DOC ($p < 0.01$; Fig. 5a). In addition,

DOC was positively related to increasing anthropogenically derived Cl$^-$ ($p < 0.05$; Fig. 5b) and NH$_4^+$-N ($p < 0.001$; Fig. S5a),

indicating anthropogenic impacts on DOC export. Unlike DOC, a significant positive correlation with mean catchment slope

was found for $\delta^{13}C_{DOC}$ ($p < 0.001$; Fig. 4b). In addition, there was a significant negative correlation between $\delta^{13}C_{DOC}$ and $NO_3^-$-N ($p < 0.001$; Fig. S5b). Moreover, $\delta^{13}C_{DOC}$ was negatively correlated with DOC concentrations ($p < 0.01$; Fig. 6a), but

positively correlated with $\delta^{13}C_{POC}$ in these three rivers ($p < 0.05$; Fig. 6b). Similar to $\delta^{13}C_{DOC}$, $\Delta^{14}C_{DOC}$ was positively related

to mean catchment slope ($p < 0.01$; Fig. 4c), and there was a positive correlation between catchment slope and $\Delta^{14}C_{POC}$ ($p <$

0.001; Fig. S6). Additionally, $\Delta^{14}C_{DOC}$ was positively correlated with $\Delta^{14}C_{POC}$ ($p < 0.01$; Fig. 6c), and no significant correlations

were detected between $\Delta^{14}C_{POC}$ and proportion of urban and agricultural land uses or ions which reflect human disturbances

(e.g., anthropogenically derived $Cl^-$ concentration, $NH_4^+$-N, and $NO_3^-$-N; $p > 0.05$; Fig. S4).

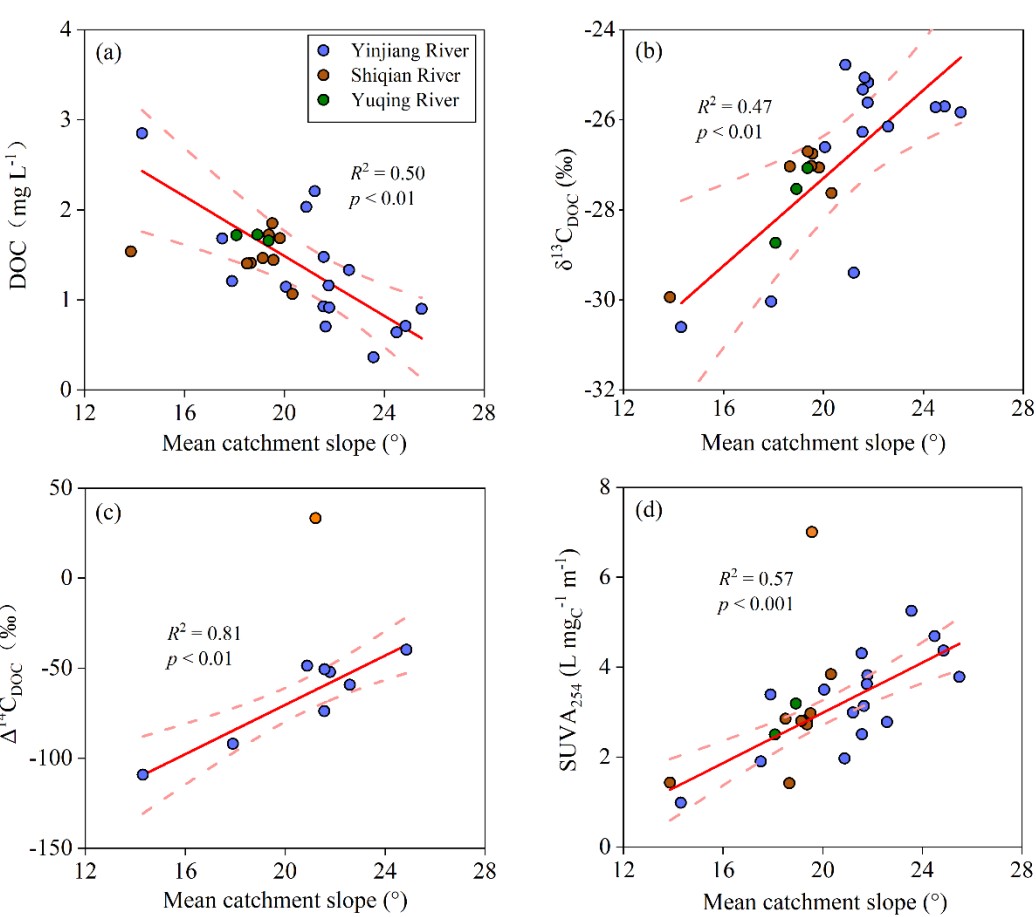

**Figure 4.** Mean catchment slope (°) controls on (**a**) DOC concentrations, (**b**) stable carbon isotopes of DOC ($\delta^{13}C_{DOC}$), (**c**) radiocarbon isotope of DOC ($\Delta^{14}C_{DOC}$), and (**d**) SUVA$_{254}$. The $\Delta^{14}C_{DOC}$ is only available for the Yinjiang River. Outliers in orange were excluded from analyses as they were samples at site Y12 (Fig. 1a) collected after a rainfall event in panel (**c**) and the sample collected at site S3 (Fig. 1a) in panel (**d**) due to the high influence by road construction, which was evidenced by high POC and TSM concentration (Chen et al., 2021). The statistical test used a significance level of 0.05.

    SUVA$_{254}$ showed an increasing trend with increasing mean catchment slope ($p < 0.001$; Fig. 4d). Furthermore, there was a

significant negative correlation between SUVA$_{254}$ and proportion of urban and agricultural land uses ($p < 0.001$; Fig. 7a).

Although no significant correlation was observed between the fluorescence indexes and catchment slope, they (except for FI)

were found to be closely related to land use pattern ($p < 0.05$; Figs. 7b, c, d, and S4). For example, HIX had a positive correlation with urban and agricultural land uses ($p < 0.001$; Fig.7e), while β/α had a negative correlation with urban and agricultural land uses ($p < 0.01$; Fig.7d). In addition, the fluorescence components did not exhibit significant variations with changing catchment slope ($p > 0.05$; Fig. S4), but the percentage of C1 and C2 were positively ($p < 0.05$; Fig. 7b) or negatively ($p < 0.01$; Fig. 7c) related to the proportion of urban and agricultural land uses. The less recently produced DOM (β/α) in the urban and agricultural streams was also characterized by a higher proportion of C1 and a lower proportion of C2 ($p < 0.001$; Fig. S4). However, unlike C1 and C2, C3 was not significantly correlated with urban and agricultural land uses ($p > 0.05$; Fig. S4).

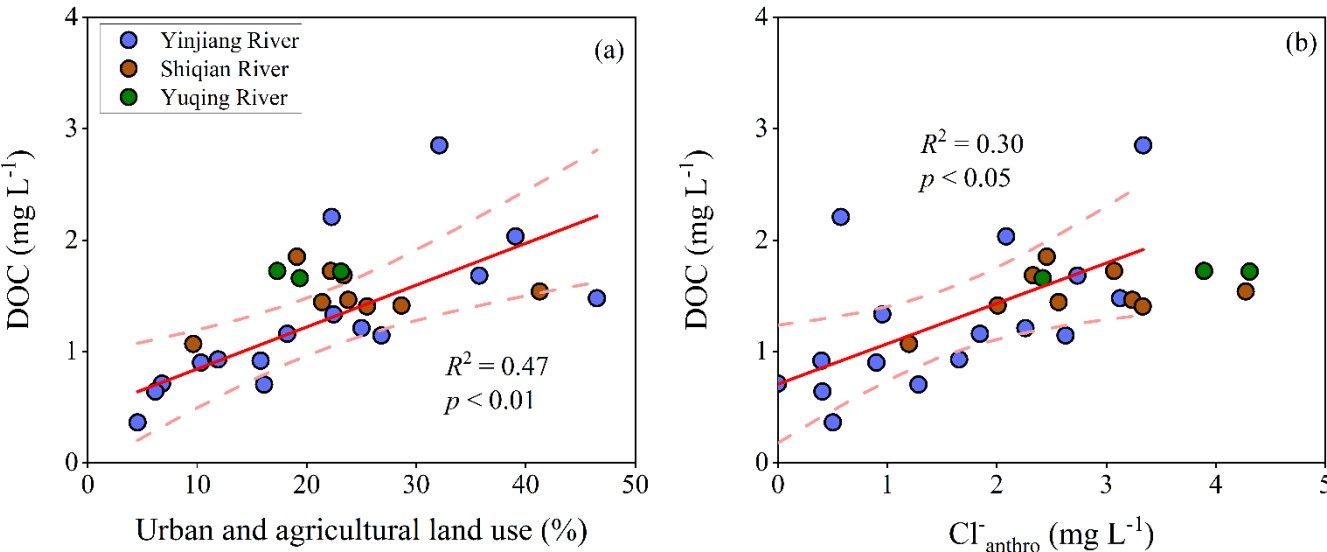

**Figure 5.** Land use pattern and anthropogenic impacts on DOC concentrations, indicated by relationships between DOC and (**a**) proportion of urban and agricultural land uses, and (**b**) anthropogenic Cl⁻ concentration (i.e., Cl⁻anthro, calculated as the total Cl⁻ concentration minus atmospheric contributed Cl⁻ concentration, which is the lowest Cl⁻ concentration at site Y5 in the Yinjiang River). The statistical test used a significance level of 0.05.

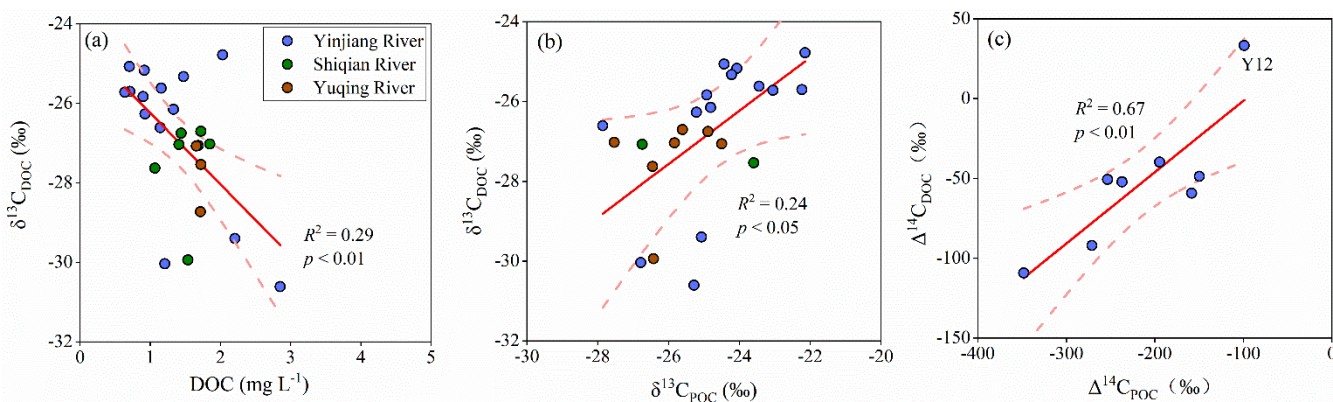

**Figure 6.** Scatter plot showing (**a**) $\delta^{13}C_{DOC}$ versus DOC in river water, (**b**) $\delta^{13}C_{DOC}$ against $\delta^{13}C_{POC}$ in the Yinjiang River (Y), Shiqian River (S), and Yuqing River (Q), and (**c**) relationship between $\Delta^{14}C_{POC}$ and $\Delta^{14}C_{DOC}$ in the Yinjiang River. For panel (**c**), the DOC with modern age at site Y12 was shown in the top-right corner. The statistical test used a significance level of 0.05. Details on $\delta^{13}C_{POC}$ and $\Delta^{14}C_{POC}$ are

available in our earlier work (Chen et al., 2021).

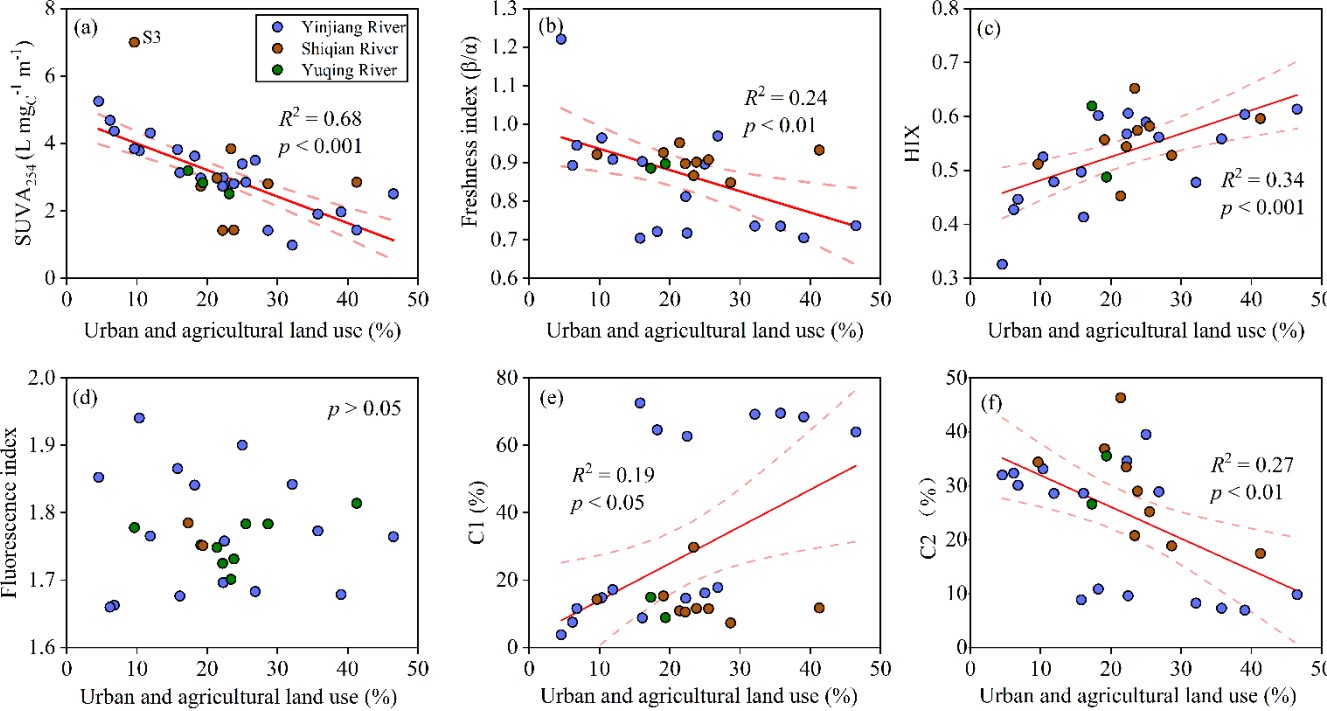

**Figure 7.** Land use pattern impacts on DOM character. (**a**) $SUVA_{254}$, (**b**) freshness index ($\beta/\alpha$), and (**f**) C2 decreased with the increasing proportion of urban and agricultural land uses. Outlier (site S3) was excluded from analysis in panel (**a**) as the sample was strongly influenced by road construction, which was evidenced by high POC and TSM concentration (Chen et al., 2021). (**c**) humification index (HIX) and (**e**) C1 were positively related to the increasing proportion of urban and agricultural land uses. However, there was no significant correlation between (**d**) fluorescence index and the proportion of urban and agricultural land uses.

**4 Discussion**

**4.1 Geomorphologic controls on DOC export**

Catchment slope is an important predictor of DOC concentrations since catchment slope is a key factor in affecting runoff velocity and thus controlling water retention time (Harms et al., 2016; Mu et al., 2015; Mzobe et al., 2020). A shorter water

retention time in high relief regions can reduce DOC export from soil organic matter stocks (Fig. 4a; Connolly et al., 2018) and mobilize organic carbon with younger ages (Figs. 4c and S6), which can be rapidly mineralized (Catalán et al., 2016). In comparison, when relatively $^{14}$C-depleted DIC and $CO_2$ (aq) derived from carbonate weathering is incorporated into primary production in low relief regions, it would also produce aged organic carbon (Fig. S6; Chen et al., 2021). Furthermore, the aged DOC in river systems has been attributed to input of deeper, older soil organic matter through deeper flow paths (Barnes et al.,

2018; Masiello and Druffel, 2001). This also indicates that low relief regions with higher hydrologic connectivity in river network are likely the major source of riverine DOC (Connolly et al., 2018; Mzobe et al., 2020). The shallow soil depth in steeper regions may also be an influencing factor in constraining DOC generation (Lee et al., 2019). The correlation of $SUVA_{254}$

with mean catchment slope suggests that steeper catchments tend to export DOC with more aromaticity (Fig. 4d), indicating

the geomorphologic effects on DOM characteristics (Harms et al., 2016). Previous research has reported that the aromatic

content of DOM tends to decline if DOM is derived from deeper soil profiles (Inamdar et al., 2011), which is attributed to the

sorption of aromatic DOM when subsurface flow water percolates through the soil profile (Inamdar, 2011). Catchment slope

in a number of Arctic watersheds has also been found to be positively associated with the FI, which was possibly due to the

differences in soil characteristics (e.g., volumetric water content and soil temperature) and the resulting changing extent of

microbial processing (Harms et al., 2016). Moreover, catchment slope was negatively associated with terrestrial humic-like

organic material due to the effects of climate and organic layer thickness (Harms et al., 2016). However, there were no similar

correlation between catchment slope and fluorescence components/indexes in this study, demonstrating the likely complicated

mechanisms (e.g., soil property, catchment characteristics, and anthropogenic activities) in regulating DOM composition. DOC

in low relief regions was characterized by more $^{13}$C depleted values, which may be due to the greater inputs of C3-derived

organic carbon (e.g., from rice).

Microbial degradation has been well recognized as a critical factor in controlling organic material preservation in soils

(Barnes et al., 2018; Eusterhues et al., 2003). Previous studies have reported a decreasing $\delta^{13}C_{DOC}$ with increasing DOC

concentrations (Fig. 6a) in spring water due to microbial degradation (Nkoue Ndondo et al., 2020) and for TOC in soil profiles

(Lloret et al., 2016; Nkoue Ndondo et al., 2020). This can be explained by the lateral transport of DOC from microbially active

soil horizons into rivers (Lambert et al., 2011), resulting in the enhanced biodegradation of DOC with the preferential removal

of $^{12}$C. As a result, the remaining DOC with lower concentrations is typically characterized by a heavier $\delta^{13}C_{DOC}$ (Fig. 5a;

Nkoue Ndondo et al., 2020; Opsahl and Zepp, 2001), which further indicates that the low-concentration DOC in the three

rivers is the result of substantial microbial degradation.

Groundwater with large SOC inputs due to highly active microbial activities has long been recognized as a significant

source of DOC (McDonough et al., 2020; Shen et al., 2014). Several studies have reported increased groundwater contributions

with distance downstream at the watershed scale (Asano et al., 2020; Cowie et al., 2017; Iwasaki et al., 2021). The positive

relationship between conductivity and $\delta^{18}$O is largely due to the mixing of two end-members (i.e., high-conductivity with $^{18}$O-

enriched groundwater and low-conductivity with $\delta^{18}$O-depleted headstream water) for river water (Lambs, 2004), though it

may also indicate the impact of evaporation in the catchment (Zhong et al., 2020). In addition, the $\delta^{18}$O values increased

progressively from upstream to downstream (Fig. S2b), which also validates the two sources (i.e., headstream water and

groundwater) of downstream river water, indicating that groundwater was likely an important contributor to downstream river

water. However, groundwater was likely not the primary source of riverine DOC due to the relatively low groundwater DOC

concentrations as compared with riverine DOC concentrations (Fig. 2e; groundwater is shown as "spring"). Moreover, the

groundwater contribution was probably much less significant in the wet season, even in catchments where DOC is mainly

derived from groundwater (Lloret et al., 2016). Thus, we infer that groundwater is an important but not a primary source of riverine DOC in the three study rivers.

Furthermore, the decrease in DOC concentration with increasing mean catchment slope (Fig. 4a) may also be controlled by annual air temperature and land use pattern (Fig. S4). Changing exports of terrestrially-derived DOC have been widely reported with increasing air temperature (Creed et al., 2018; Parr et al., 2019; Voss et al., 2015). Lower altitude is generally associated with a higher annual air temperature (Fig. S4), which promotes terrestrial primary production and degradation of POC (Mayer et al., 2006) and subsequently the accumulation of large quantities of OC in soils (Creed et al., 2018). It is also worth noting that the higher temperature may facilitate microbial degradation of soil DOC (Voss et al., 2015).

## 4.2 Anthropogenic impacts on DOC

Previous research has found significant changes in DOC concentrations and DOM composition in agricultural and urban landscapes (Spencer et al., 2019; Stanley et al., 2012; Williamson et al., 2021). Conversion of native forest and pasture to row crop agriculture may lead to substantial losses of SOC stores as a result of greatly accelerated erosion rates and decomposition rates (Guo and Gifford, 2002; Montgomery, 2007; Stanley et al., 2012). In comparison, natural vegetation could greatly reduce SOC input into rivers by effectively reducing soil erosion through the consolidation effect of roots on soil and the interception of rainfall by stems and leaves (Zhang et al., 2019). Anthropogenic activities are closely related to mean drainage elevation (Fig. S4). Agricultural activities mainly occur in low-elevation areas (Fig. S4), which tend to liberate SOC through erosion over longer timescales and cause an elevated DOC export into rivers (Fig. 5a), although DOC of urban origin can also make a huge contribution to the riverine DOC pool (Sickman et al., 2007). Yet, anthropogenic impacts can also decrease DOC concentrations (Spencer et al., 2019; Williams et al., 2010) or lead to undetectable changes in DOC concentrations (Veum et al., 2009). These uncertain responses of DOC concentration are mainly due to diverse farming practices and associated changing effects on terrestrial and aquatic carbon dynamics (Stanley et al., 2012).

Anthropogenic activities are important factors for the pervasive increase in nutrient and ion concentrations (Chetelat et al., 2008; Smith and Schindler, 2009). For catchments without evaporite outcrops, their riverine $Cl^-$ excluding atmospheric contribution can be regarded as mainly of anthropogenic origin ($Cl^-_{anthro}$; Meybeck, 1983). The positive relationship between DOC concentrations and $Cl^-_{anthro}$ as shown in Fig. 5b also demonstrated anthropogenic impacts on DOC export. Nutrient enrichment has been a well-known contributor to eutrophication (Paerl, 2009). Together with increasing water residence time due to damming and reservoirs (Fig. 1a), our results show that enhanced nutrient inputs into rivers will promote algae production (Chen et al., 2021) and eventually accumulation of DOC (Fig. S5a). A recent study conducted in the Longtan Reservoir in the Xijiang River basin (China) with widespread karst landscape found that a majority of its POC was intercepted or degraded within the reservoir, with the POC primarily originating from phytoplankton (Yi et al., 2022). Its carbon isotope

composition of POC ($\delta^{13}C_{POC}$) ranged from -35‰ to -30‰, which is relatively depleted, and the POC was found to be a significant contributor to the reservoir's DOC (Yi et al., 2022). Thus, the lower $\delta^{13}C_{DOC}$ with increasing $NO_3^-$-N further indicated the greater algae or C3 plant derived DOC accumulation with a higher level of nutrients (Fig. S5b). Anthropogenic impacts on DOM characters and age have been widely proposed in the last two decades (Butman et al., 2014; Coble et al., 2022; Vidon et al., 2008; Zhou et al., 2021). There are no clear relationships between land use and $^{14}$C ages in our study area, which may be the result of large variations in soil characteristics and limited $^{14}$C data. However, DOM characters were found to be closely related to land use patterns (Fig. 7). Although significant relationships with urban and agricultural land uses were found for C1 and C2 (Figs. 7e and f), it remains unclear how the autochthonous contribution to riverine DOC pool varied with land use change because C1 and C2 are both likely derived from autochthonous production but exhibit opposing trends with increasing urban and agricultural land uses. Overall DOM in catchments with a higher proportion of urban and agricultural land uses were distinct from other catchments as it was less aromatic (SUVA$_{254}$, Fig. 7a), less recently produced ($\beta/\alpha$, Fig. 7b), and had a higher degree of humification (HIX, Fig. 7c). SUVA$_{254}$ values for the three study rivers were comparable with those reported in coastal glacier mountainous streams with late succession in southeast Alaska (3.4 ± 0.5 L mg$^{-1}$ m$^{-1}$, n = 5; Holt et al. 2021) and in the anthropogenic influenced downstream of the Yangtze River (3.4 ± 1.1 L mg$^{-1}$ m$^{-1}$, n = 82; Zhou et al. 2021). Lower DOM aromaticity in the urban and agricultural streams and rivers was consistent with previous studies (Hosen et al., 2014; Kadjeski et al., 2020), though it was not a universal phenomenon (Zhou et al., 2021). Furthermore, the less aromatic and recently produced DOM could be due to soil organic materials from deep soil profiles as a result of increased soil erosion by anthropogenic activities (Inamdar et al., 2011; Stanley et al., 2012).

## 4.3 Combined effects of geomorphologic and anthropogenic controls on DOC and comparison of $\Delta^{14}C_{DOC}$ in mountainous rivers

In this study, geomorphologic characteristics and anthropogenic activities were identified as significant drivers of DOC export and DOM composition across broad spatial scales. Here, we further examine how these two factors regulate riverine DOC. The riverine DOC age ranged from modern to greater than 1000 years, which is mainly determined by its sources and aquatic processing (Butman et al., 2012; Koch et al., 2022; Moyer et al., 2013). Generally, DOC is mainly derived from surface soil, decaying terrestrial plants, and autochthonous production, which usually contain a modern carbon pool (Findlay and Sinsabaugh, 2004; Zhou et al., 2018). In addition, old carbon is usually exported from pools such as deep soil layers, peat, shale, groundwater, and terrestrial organisms which incorporate inorganic carbon from weathering sources (Butman et al., 2014; Moyer et al., 2013; Raymond et al., 2004). Petroleum-based carbon export through wastewater, urban, and agricultural runoff has been recognized as a potentially important source of old carbon (Butman et al., 2012; Griffith et al., 2009; Sickman et al., 2010). The DOC ages of the Yinjiang River (548 ± 195 yr BP; the modern sample at site Y12 was excluded) were younger than that of DOC reported in agricultural rivers (Moyer et al.,

2013; Sickman et al., 2010) or treated wastewater (Griffith et al., 2009), which is typically more than 1000 years old. However, we cannot conclude that DOC in the Yinjiang River was not influenced by anthropogenic activities as the wide range of anthropogenically-impacted DOC ages (Butman et al., 2014) and the various sources of anthropogenic DOC as discussed above, which led to the insignificant correlation between the proportion of urban and agricultural land uses and isotopes of organic carbon (Fig. S4). The input of large amounts of young, terrestrially derived organic matter through surface runoff during rainfall events could explain the young age of the DOC and POC at site Y12 (Table 3 and Fig. 6c), where the sample was collected after a rainfall event (Chen et al., 2021). Furthermore, the weak positive correlation between the $\delta^{13}C_{DOC}$ and $\delta^{13}C_{POC}$ in these three rivers (Fig. 6b) indicated that DOC and POC may have been derived from the same source. We also found a strong positive correlation between $\Delta^{14}C_{DOC}$ and $\Delta^{14}C_{POC}$ in the Yinjiang River (Fig. 6c). The coupling between $\Delta^{14}C_{DOC}$ and $\Delta^{14}C_{POC}$ was an unusual relationship rarely found in other rivers (Campeau et al., 2020; Longworth et al., 2007). Campeau et al. (2020) attributed this relationship to common controls of landscape and/or hydrology on the sources of organic carbon in rivers. Yet, this correlation might have been masked by the mixing of waters from other tributaries, underscoring the combined impact of geographical factors (e.g., landscape) and anthropogenic influences (e.g., dam construction, as discussed below) on DOC sources.

Furthermore, the widespread reservoirs for irrigation and water supply would lead to the prolonged water retention time across river systems, entailing a great change in organic carbon reactivity and $CO_2$ emissions (Catalán et al., 2016; Ran et al., 2021; Yi et al., 2021). Meanwhile, reservoirs provide a favorable environment for aquatic photosynthesis and bacterial production, thereby increasing autochthonous DOM production and accumulation in rivers (Ulseth and Hall, 2015; Xenopoulos et al., 2021). In addition to influencing DOC dynamics, our earlier research demonstrated that damming and reservoirs can also significantly affect the dynamics of POC and DIC (Chen et al., 2021). Additionally, DO was significantly different between dammed rivers and undammed rivers (Fig. 8), further indicating the damming effect on the in-stream photosynthesis and, consequently, on organic carbon dynamics. Therefore, the carbon cycling in the three rivers is collectively affected by a combination of factors, including geographical controls, anthropogenic impacts, and in-stream processes. Particularly, geographical controls on DOM were mainly evidenced by carbon isotopes, while anthropogenic impacts were supported by the DOM fluorescence characters (Figs. 3 and 6). There was no significant relationship between carbon isotopes and optical properties, which is inconsistent with previous studies (Aiken et al., 2014; Butman et al., 2012; Lee et al., 2021; Sickman et al., 2010; Zhou et al., 2018). This is likely due to the potential masking effect of autochthonous DOM, as also evidenced by the decoupled relationship between $\Delta^{14}C_{DOC}$ and $SUVA_{254}$ in the St. Lawrence River (Aiken et al., 2014; Butman et al., 2012). However, it should be noticed that anthropogenic activities are spatially related to elevation, air temperature, and catchment slope (Fig. S4). Therefore, disentangling the dual influences (geographical and anthropogenic) is challenging, as they are likely to cohesively impact both DOC concentration and DOM quality in these rivers. It is evident that DOC dynamics are

complicated in the study rivers, and a comprehensive assessment of the biogeochemical processes of DOC and their multiple controlling factors will advance our understanding of carbon cycling.

Carbon isotopes of DOC and its concentration in mountainous rivers were summarized in Table 4. Global average $\Delta^{14}C_{DOC}$ is -11.5 ± 134 ‰, higher than that in the Yinjiang River (-54.7 ± 39.9 ‰; Tables 3 and 4; Marwick et al., 2015) while similar to many other mountainous rivers (e.g., the Mackenzie River; Campeau et al., 2020) and small mountainous rivers in Puerto Rico; Moyer et al., 2013). $\Delta^{14}C_{DOC}$ values for the world's mountainous streams and rivers were shown by climate (according to the Köppen–Geiger climate classification (Beck et al., 2018; Table 4) and ranged from tropical monsoon climate (Marwick

**Table 4.** Comparison of carbon isotopes of DOC in mountainous rivers worldwide.

| Rivers/Region | Sampling Date (mmyyyy) | Climate | DOC (mg L[-1]) | $\delta^{13}C_{DOC}$ (‰) | $\Delta^{14}C_{DOC}$ (‰) | References |
|---|---|---|---|---|---|---|
| The Yinjiang River (China) | 08/2018 | | 1.3 ± 0.7 | -26.6 ± 1.9 | -55 ± 38 | This study |
| Zambezi (Mozambique) | 02/2012-04/2012 | | 2.4 ± 0.6 | -21.9 ± 2.4 | 64 ± 23 | (Marwick et al., 2015) |
| Betsiboka (Madagascar) | 01/2012-02/2012 | | 1.3 ± 0.6 | -22.8 ± 2.1 | 86 ± 43 | |
| Amazon[a] | 05/1995-10/1996 | Tropical | 1.9 ± 0.7 | -26.0 ± 3.0 | 94 ± 176 | (Mayorga et al., 2005) |
| Guanica and Fajardo (Puerto Rico) | 09/2004-03/2008 | | 2.3 ± 2.1 | -26.1 ± 3.1 | -55 ± 105 | (Moyer et al., 2013) |
| North-West Australia (Australia) | 05/2010 and 06/2011 | | 1.5 ± 0.7 | -25.0 ± 1.7 | -67 ± 124 | (Fellman et al., 2014) |
| Santa Clara (USA) | 11/1997-03/1998 | | 6.2 ± 2.7 | -26.1 ± 0.9 | -148 ± 58 | (Masiello and Druffel, 2001) |
| Conwy (Wales)[b] | | Temperate | 9.2 ± 7.3 | -28.0 ± 1.8 | 105 ± 6 | (Evans et al., 2007) |
| Brocky Burn (Scotland) | 02/1998 and 06/1998 | | | -27.9 ± 0.2 | 29 ± 12 | (Palmer et al., 2001) |
| Southeast Alaska | 07/2013 | | 0.8 ± 0.2 | -27.0 ± 1.6 | -93 ± 77 | (Holt et al., 2021) |
| Gulf of Alaska | 07/2008 | | 1.2 ± 0.5 | -23.9 ± 1.1 | -207 ± 121 | (Hood et al., 2009) |
| Alaska[c] | 05/2012-10/2012 | | 3.7 ± 4.1 | -27.4 ± 0.8 | -10 ± 55 | (Behnke et al., 2020) |
| Kolyma (Russia)[d] | 01/2003-12/2003 | | | -28.5 ± 1.3 | 57 ± 51 | (Neff et al., 2006) |
| Hudson (USA)[e] | 01/2004 | | 5.9 ± 0.7 | -27.0 ± 0.0 | -26 ± 13 | (Raymond et al., 2004) |
| Central Ontario (Canada) | 1990-1992 | Continental | 6.4 ± 4.5 | | 96 ± 79 | (Schiff et al., 1997) |
| Mackenzie River Basin (Canada)[f] | 06/2018 | | 4.3 ± 1.8 | -26.9 ± 0.2 | -55 ± 72 | (Campeau et al., 2020) |
| Mulde (Germany) | 08/2008-10/2010 | | 9.8 ± 7.3 | -26.6 ± 0.5 | 7 ± 27 | (Tittel et al., 2013) |
| Fraser (Canada) | 07/2009-05/2011 | | 4.1 ± 5.6 | -26.5 ± 0.5 | 58 ± 34 | (Voss et al., 2022) |
| Yangtze River source region (China) | 02/2017-12/2017 | | 2.9 ± 1.4 | -27.9 ± 3.3 | -397 ± 185 | (Song et al., 2020) |
| Tibetan Plateau (China) | | Continental/Dry | 0.27 ± 0.0 | -23.5 ± 0.2 | -209 ± 71 | (Spencer et al., 2014) |

[a] Only rivers draining mountainous areas from the Andean Cordillera were reported. [b] Data were obtained from Marwick et al. (2015). [c] Calculated from mean values. [d] Only mountainous and upland rivers were reported. [e] Only Upper Hudson River was reported. [f] Only

tributaries sourced from Cordillera were reported.

et al., 2015), temperate oceanic climate (Evans et al., 2007), cold semi-arid climates (Spencer et al., 2014) to continental
subarctic climate (Hood et al., 2009). DOC of young age in mountainous rivers were reported across climate (Evans et al., 2007; Mayorga et al., 2005; Voss et al., 2022). While the most aged DOC was observed in the Tibetan Plateau (Song et al., 2020; Spencer et al., 2014) and the Gulf of Alaska (Hood et al., 2009). The riverine aged DOC from these regions with cold climate was mainly sourced from melting glacier with high bioavailability (Hood et al., 2009; Spencer et al., 2014) or derived from permafrost thaws in deeper soil horizons with deeper flow paths (Song et al., 2020). As global air temperature increases,
the greater input of the aged yet microbially labile DOC into rivers would lead to increasing emission of $CO_2$ and $CH_4$, which in turn intensifies global warming (Vonk and Gustafsson, 2013).

**5 Conclusions**

This study provided an insight into the DOC dynamics and their influencing factors in anthropogenically-impacted subtropical small mountainous rivers. Variations in DOC concentrations are regulated by both internal processes and external geographical
and anthropogenic disturbances. We observed a positive relationship between DOC concentrations and anthropogenic land use but a negative correlation between DOC concentration and catchment slope. Carbon isotopes variations were mainly due to changing mean catchment slope, while fluorescence properties of DOM were highly influenced by land use. Additionally, we found increased aromaticity with elevated catchment slope and reduced agricultural and urban land uses, indicating the geographical and anthropogenic controls on DOM character. We attribute these diverse DOC responses to altered water
retention time, the SOC dynamics, and water flow paths. This study highlights that the combination of dual carbon isotopes and optical properties are useful tools in tracing the origin of riverine DOC and its in-stream processes. With continued economic development and population growth, anthropogenic impacts on DOC are expected to be increasingly evident. However, anthropogenic impacts may alter various biogeochemical processes of DOC in different catchments with changing geographical features due to complicated regulating mechanisms of organic carbon cycling, which to date remains poorly
understood. Further studies are warranted to fully understand the combined effects of local geographical controls and increasing anthropogenic impacts on DOC cycling.

*Data availability.* Data are available from the corresponding author Lishan Ran upon request at lsran@hku.hk.

*Author contributions.* JZ and SL conceived and designed the study. WW, JZ, and ZY contributed to the fieldwork. SC, WW, YY and KMM contributed to the laboratory work and data analyses. SC wrote the original draft. LR, JZ and YY reviewed and edited the manuscript.

*Competing interests.* The authors declare that they have no conflict of interest.

*Financial support.* This research was funded by the Strategic Priority Research Program of the Chinese Academy of Sciences (XDB40000000), National Natural Science Foundation of China (Grant Nos. 41925002, 41422303 and 41803007), and the Research Grants Council of Hong Kong (Grant No. 17300621).

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
