# Peer review of "Geomorphologic controls and anthropogenic impacts on dissolved organic carbon from mountainous rivers: Insights from optical properties and carbon isotopes"

_Biogeosciences, 2022_

## Referee Comment (RC2)

**Summary:**

This study assessed how geographic controls (elevation, temperature, and slope) and % anthropogenic land cover (urban/agriculture) influence DOC export and DOM composition from mountainous rivers. The data presented shows that increased %urban/agriculture cover in lower reaches (and shallower gradients) of these catchments results in higher DOC concentrations, where carbon isotopic signatures (13 and 14C-DOC) of DOC are more deplete, and DOM is less aromatic. I believe these findings are of interest to a broad community. However, I have a number of concerns that I would like the authors to address and some suggestions to improve their manuscript.

**Major:**

1. The SUVA$_{254}$ values presented in this paper are typically >5 L mg C$^{-1}$m$^{-1}$. These values are extremely high when compared to blackwater riverine systems that typically have high aromaticity DOM (e.g., Holt et al., 2021; Spencer et al., 2010; Weishaar et al., 2003 and references there in). Please can you confirm how SUVA values were calculated (and include this information in text and/or DOC and absorbance data in a table). Were decadic or napierian absorbance measurements used for calculations? SUVA values should be calculated from decadic absorbance and if naperian values were used instead this may explain why the values here are so high (Hu et al., 2002; Spencer et al., 2014). Some justification of why SUVA values are seemingly so high would be useful here, and it would be good if these values were contextualized in relation to past work for mountainous rivers (and anthropogenically impacted catchments).

2. There are details missing from the methods in relation to how % land use, slope and elevation was calculated or whether this data is from the author's previous study. This information should be included in this paper since many of the figures and findings are reliant on this data. I would also recommend data from this analysis is presented and described within the site description before it is used to inform your analyses with DOM composition.

3. Findings of figure 5,6,7 are discussed without an initial description of the data/trends and thus much of the discussion comes across as a little abrupt. I would recommend you restructure and describe these figures/trends in the results section, then explain what these trends (either individually or collectively) may mean in the discussion. As it stands the discussion and results are a little brief and findings are not really discussed in detail (especially section 4.1). Given the structure it is also difficult to follow the primary reasons for the trends you observe. I wonder if you could examine trends within figures 5-7 collectively (e.g., through a PC analysis) so connections between land use and slope can be made and discussed in tandem rather than separately?

4. L189-190 it is unclear how 'enhanced' biodegradation of DOC would increase DOC concentrations. I would have thought that biodegradation would remove DOC. Also,

how can you be sure that this trend in 13C-DOC is microbially driven, rather than an increased input from aged soil/C3 plants in lower elevation stream reaches? You note in your site description that there is C3/C4 agriculture across the catchments. Is this coverage variable and could this in part drive the trend in 13C-DOC? Additionally, wouldn't removal of 12C by microbes lead to enrichment in 13C-DOC not depletion as you describe?

5. I wonder if the authors could further discuss how they arrived at the conclusion that groundwater was a significant source of DOC to these rivers, and that this groundwater played an important role in diluting DOC concentrations during base flow (1) given that statistical analysis was not performed between springs and river DOC samples; (2) that samples were taken during heavy rainfall periods (i.e., monsoon; September) and thus baseflow conditions were unlikely to have been examined; (3) optical and isotopic data was not used to inform this discussion.

6. Fluorescence properties of DOM are presented but not related to geographic or anthropogenic features of the catchments. It is unclear why this isn't considered in the manuscript, and I wonder if there are any trends observed? At least, the information gained from optical analyses should be explained and contextualized within the discussion.

7. Similarly, it is unclear in section 4.3 why anthropogenic impacts are only discussed in relation to DOC concentration. I wonder if you can draw a connection with carbon isotopes (and optical properties) and if this could help you understand the primary drivers of variability within your dataset.

**Minor comments and technical/typographical corrections:**

Abstract:
L19 'of DOC' seems a little awkward here – consider rewording
L21 POC is not defined. It is not clear how instream processing of POC is a source of DOC in this sentence. I would also make it clearer that you are using POC values from past work within the abstract and aims/objectives.
L23-24 consider making the distinction between DOC and DOM here as I think it would help with sentence flow. It is also unclear how this was 'distinct' from those catchments with lower slopes/higher temperatures.
L25 DOM is not defined
L28 I think you could make the significance more specific/explicit here in relation to your findings.

Introduction:

1. Generally, within the introduction I think it would be useful to provide more specific details in terms of DOM compositional shifts that have been noted with warming and

land use changes as well as across geographic gradients (i.e., elevation and slope). Similar to that in line 63. Much of your description only specifies that there are 'changes' but doesn't note the typical direction of change. I think this would make it easier on the reader later when reading the results/discussion as many of the changes would be somewhat familiar and would also situate your study more firmly in relation to past work.

2. I would also suggest you integrate the points made in lines 68-72 into the previous paragraphs. This section appears a little obvious and doesn't really make it clear why there is utility in using these techniques within your study.

3. Finally, within the aims and objective paragraph it would be useful to be more specific of the techniques you are using (e.g., DOM quality assessed through optical metric) and the geographic/land use parameters use are assessing against DOM quality/DOC concentration.

L33 given that your study has a large land use component, I'd suggest broadening this sentence out to encompass this.
L43 can you explain the 'difference' more specifically?
L53 this sentence is a little unclear to me. Consider rewording.
L65 consider rephrasing/reordering the sentence – 'recent pursuit' is a little awkward.
L73 please specify the geographic and anthropogenic factors you are assessing
L75 remove 'their'.
L76 add 'here'. So, it reads 'Here, we investigate…'
L76 it is unclear from this sentence how you asses autochthonous processes.
L77 I feel this hypothesis could be more specific based on past literature.
L80 seems a little obvious, can you be more specific on the insight gained?

Methods:

1. It appears from the results that 14C-DOC values only available for the Yinjiang River. This must be made clear in the aims and methods. Also, why is this the case?

L86 'of the' replace with 'in'
L89 rephrase so land use is not repeated
L104 there is a missing word here.
L108 I think more information would be useful here, despite information being published in your previous work.
L127 replace substances with fluorescence
L135 – double brackets to be replaced with ';' - please check throughout.
L140 how was proportion of different land uses/elevation/slope calculated? Was this data previously published? I would recommend adding this information as a table to the text and how this data came about in the site description.

L161 – the median value for river Y is not higher than the other rivers. Thus, it's unclear what you are referring to here.

Results/Discussion:

1. Description of optical properties in the results is extremely brief and lacks quantitative details. E.g., what are the average SUVA values for each river? What is the % each component of fluorescence is explaining? Please include these details.
2. Generally, geographic and land use parameters are not discussed in the results. However, SUVA is briefly described and then related to slope. Given the structure of the results it would make sense to wait to draw the comparison with slope. Consider including a summary figure (e.g., boxplot for SUVA) and then including SUVA v slope analysis. Similarly, why is Figure 4 a boxplot whilst other relationships with slope conducted as linear regressions?

Figure 4 – specify units of SUVA254 on axis
Figure 5 – place key at the bottom of the four panels. Dots in key maybe confused with datapoints.
Figure 5 – why are results in panel A reported as 1/DOC? Please just report as DOC since this leads to confusion in text e.g., line 188.
L166 it is unclear why you contextualize this finding but not others. Please contextualize data throughout results or move this point to the discussion.
L225 If anthropogenic activities were not the primary source of aged DOC in your catchments (as you imply). What is the primary source of aged DOC? This point should be made clearer
L230 it would be prudent to explain the two endmembers here in more detail.
L255 is this supported by your data?

Holt, A. D., Fellman, J., Hood, E., Kellerman, A. M., Raymond, P., Stubbins, A., et al. (2021). The evolution of stream dissolved organic matter composition following glacier retreat in coastal watersheds of southeast Alaska. *Biogeochemistry*, 1-18.

Hu, C., Muller-Karger, F. E., & Zepp, R. G. (2002). Absorbance, absorption coefficient, and apparent quantum yield: A comment on common ambiguity in the use of these optical concepts. *Limnology and Oceanography, 47*(4), 1261-1267.

Spencer, R. G., Guo, W., Raymond, P. A., Dittmar, T., Hood, E., Fellman, J., & Stubbins, A. (2014). Source and biolability of ancient dissolved organic matter in glacier and lake ecosystems on the Tibetan Plateau. *Geochimica et Cosmochimica Acta, 142*, 64-74.

Spencer, R. G., Hernes, P. J., Ruf, R., Baker, A., Dyda, R. Y., Stubbins, A., & Six, J. (2010). Temporal controls on dissolved organic matter and lignin biogeochemistry in a pristine tropical river, Democratic Republic of Congo. *Journal of Geophysical Research: Biogeosciences, 115*(G3).

Weishaar, J. L., Aiken, G. R., Bergamaschi, B. A., Fram, M. S., Fujii, R., & Mopper, K. (2003). Evaluation of specific ultraviolet absorbance as an indicator of the chemical composition

and reactivity of dissolved organic carbon. *Environmental science & technology, 37*(20), 4702-4708.

---

## Author Comment (AC1)

**Response to Anonymous Referee #1**

This paper evaluated how land use practices and catchment slope impact concentrations and sources of DOC in three mountainous rivers. They find that agricultural activities at lower elevations increase DOC concentrations and lead to more terrestrial ($^{13}$C-depleted) and old ($^{14}$C-depleted) organic carbon in rivers. Findings from this study address a current gap on the factors controlling DOC source (as measured by $^{13}$C, $^{14}$C, absorbance, and fluorescence) in these ecosystems because most of the literature to date has focused on POC cycling. I have three suggestions to improve the manuscript:

**Our response:** We appreciate the positive comments of the reviewer for recognizing the importance of the research. In addition, we thank the reviewer for the suggestions that has substantially improved our manuscript.

1. Many of the water chemistries measurements made in the three rivers in Figure 2 were reported in a recent paper (Chen et al. 2021). The study adds to this dataset by comparing water chemistries between the rivers and a spring water source, but there are two shortcomings. First, there is further discussion of how DOC concentrations differed between the sources, but not the other variables measured. Second, no statistical analyses were performed to determine whether the water chemistries of spring water were different from the river waters. Section 3.1 of the results should be revised to clarify when there were statistically significant differences or not between the sites sampled.

**Our response:** Thanks. We have added further discussion on how the other variables varied between the sources in the manuscript (please see the section "3 Result" for more details). For example, "*Both river water and spring water were mildly alkaline, with pH varying from 7.2 to 8.9, and pH in the Yinjiang and Shiqian rivers was higher relative to that in the spring water (Fig. 2a). The average DO presented similar values between the Yinjiang River, Shiqian River, Yuqing River, and springs with the majority of the river water samples being DO supersaturated (Fig. 2b). The Shiqian River had a higher water temperature than that in the Yinjiang River and spring water (Fig. 2c). A strong positive correlation was found between EC and $\delta^{18}O$ for the river water and spring water (Fig. S2a), and the $\delta^{18}O$ showed an increasing trend from upstream to downstream in the Yinjiang River (Fig. S2b).*" (P8 Line 196-199), "*The TP concentrations varied significantly in the study area, ranging from 0.03 to 0.27 mg L$^{-1}$, and the average TP concentration was $0.19 \pm 0.08$ mg L$^{-1}$ in the Yuqing River, considerably higher than that in the Yinjiang River ($0.10 \pm 0.03$ mg L$^{-1}$; Fig. 2f). In addition, the TP concentrations ranged between 0.11 to 0.24 mg L$^{-1}$ in the springs, averaging at $0.15 \pm 0.06$ mg L$^{-1}$. Within the rivers and springs, the water displayed similar NH$_4^+$-N concentrations with the average value at $0.04 \pm 0.03$ mg L$^{-1}$, $0.07 \pm 0.05$ mg L$^{-1}$, $0.04 \pm 0.03$ mg L$^{-1}$ and $0.03 \pm 0.04$ mg L$^{-1}$ in the Yinjiang, Shiqian, Yuqing rivers, and spring water (Fig. 2g). In springs, the average NO$_3^-$-N and TN concentrations were $1.93 \pm 0.93$ mg L$^{-1}$ and $2.88 \pm 1.30$ mg L$^{-1}$, respectively, higher than the average in the three rivers ($1.15 \pm 0.36$ mg L$^{-1}$ for NO$_3^-$-N and $1.77 \pm 0.50$ mg L$^{-1}$ for TN), though there were no significant differences for the overall NO$_3^-$-N and TN concentrations between the rivers and springs (Figs. 2h and 2i).*" (P8 Line 203-210), and "*The DOC concentrations in spring water were significantly lower than that in the surface water of the Shiqian and Yuqing rivers (Fig. 2d), and the average DOC concentration in spring water ($0.74 \pm 0.30$ mg L$^{-1}$) was also lower than the average DOC concentration in the Yinjiang River ($1.27 \pm 0.66$ mg L$^{-1}$). An analysis of Pearson's correlation coefficients revealed significant pairwise interdependencies between DOC and catchment characteristics (Fig. S3 in the Supplement).*" (P9 Line 222-226).

In addition, statistical analyses have been added to identify the differences between the sources in the manuscript as: "*Normality of the data was first examined by a Shapiro-Wilk test using SPSS 26. Normally distributed data were analyzed by one-way ANOVA with Tukey's post-test for multiple comparisons. Nonparametric data with three or more comparisons were made by Kruskal–Wallis test followed by Holm's Stepdown Bonferroni correction. The Mann–Whitney U test was used for comparison of distributions between two groups. Linear regression was applied using Origin (Pro) 2021 to quantify the relationship of DOC concentrations, DOM properties, carbon isotopes, and ion concentrations versus catchment characteristics (i.e., mean channel slope, proportion of different land uses, mean annual air temperature, and mean drainage elevation) to identify the predominant influencing factors on DOC dynamics. Moreover, the correlation among water chemistry and catchment characteristics was computed by Pearson's correlation coefficients (R) by Origin (Pro) 2021. All statistical tests were performed at 0.05 significance level. In addition, all the statistical analyses were performed again after data from site Y12 were removed to test the possible skew of findings as the sample was significantly affected by rainfall events. If not mentioned otherwise, the results from site Y12 did not skew the findings at the significance level of 0.05.*" (P7 Line 182-193).

2. The authors discuss in Section 4.2 how river reaches with shallower slopes had higher DOC concentrations and more terrestrial ($^{13}$C), less aromatic (SUVA$_{254}$), and older ($^{14}$C) DOC sources, and in Section 4.3 how the agricultural activities consistently take place at lower elevations. Because the former result could be due to agricultural activities as well as increased erosion from those activities, and not as much from shallower slopes, I suggest that the authors revise the discussion to make this point more clearly.

**Our response:** Thanks for your suggestion. We have further described that DOC concentrations, isotopes and DOM property was controlled by both slope and human land use in the revised manuscript. We have added related discussion to make it more clearly in the manuscript as follow: "*The lower $\delta^{13}C_{DOC}$ with increasing NO$_3^-$-N further indicated the greater algae or C3 plant derived DOC accumulation with a higher level of nutrients (Fig. 4d). Anthropogenic impacts on DOM characters and age have been widely proposed in the last two decades (Butman et al., 2014; Coble et al., 2022; Vidon et al., 2008; Zhou et al., 2021). There are no clear relationships between land use and $^{14}C$ ages in our study area, which may be the result of large variations in soil characteristics and limited $^{14}C$ data. However, DOM characters were found to be closely related to land use pattern (Fig. 6). Although significant relationships with urban and agricultural land uses were found for C1 and C2 (Figs. 6b and c), it remains unclear how the autochthonous contribution to riverine DOC pool varied with land use change because C1 and C2 are both likely derived from autochthonous production but exhibit opposing trends with increasing urban and agricultural land uses. Overall DOM in catchments with a higher proportion of urban and agricultural land use area was distinct from other catchments as it*

*was less aromatic (SUVA$_{254}$, Fig. 6a), less recently produced (β/α, Fig. 6d), and had a higher degree of humification (HIX, Fig. 6e). Lower DOM aromaticity in the urban and agricultural streams and rivers was consistent with previous studies (Hosen et al., 2014; Kadjeski et al., 2020), though it was not a universal phenomenon (Zhou et al., 2021). Furthermore, the less aromatic and recently produced DOM could be due to soil organic materials from deep soil profiles because of the increased soil erosion rates associated with anthropogenic activities (Inamdar et al., 2011; Stanley et al., 2012).*" (P17 Line 378-395) and "*Furthermore, the weak positive correlation between the δ$^{13}$C$_{DOC}$ and δ$^{13}$C$_{POC}$ of these three rivers (Fig. 5b) indicated that DOC and POC may derive from the same source. We also found a strong positive correlation between Δ$^{14}$C$_{DOC}$ and Δ$^{14}$C$_{POC}$ in the Yinjiang River (R$^2$ = 0.67, p < 0.01; Fig. 5c). The coupling between Δ$^{14}$C$_{DOC}$ and Δ$^{14}$C$_{POC}$ was an unusual relationship rarely found in other rivers (Campeau et al., 2020; Longworth et al., 2007). Campeau et al. (2020) have attributed this relationship to common controls of landscape and/or hydrology on the sources of organic carbon in rivers, and this correlation could be masked by the mixing of waters from other tributaries, which further support the combined geographical controls (e.g., as a driver of landscape) and anthropogenic impacts (e.g., dam construction, see discussion below) on DOC sources.*" (P18 Line 415-422).

3. I also wonder if the authors could discuss further how they arrived at the conclusion that $^{14}$C-DOC came from POC, and the implication of that finding. For example, if the authors show that DOC is more $^{14}$C-depleted in river reaches with more agricultural activity, then couldn't the DOC be coming from agricultural sources? Or would you expect wastewater and agricultural runoff have different $^{13}$C-DOC signatures compared to the POC? Chen et al. (2021) recently showed that aquatic photosynthesis was the main source of POC in these rivers. If DOC is coming from POC, then the authors should discuss how and why more $^{13}$C- and $^{14}$C-depleted POC from photosynthesis would be present at lower elevations with agricultural activity. Lastly, the discussion would benefit from a comparison of findings on $^{14}$C-DOC signatures to other papers that have measured this in mountainous rivers, including Masiello and Druffel 2001, Longworth et al. 2007, Moyer et al. 2012, Schwab et al. 2022.

**Our response:** Thanks. Combined with your and other reviewers' opinion, we have arrived at new conclusions on $^{14}$C-DOC as: "*Furthermore, aged DOC in river systems has been attributed to old soil organic matter in deeper layer input into rivers through deeper flow paths (Barnes et al., 2018; Masiello and Druffel, 2001). This also indicates that low relief regions with higher hydrologic connectivity in river network are likely the major source of riverine DOC (Connolly et al., 2018; Mzobe et al., 2020).*" (P15 Line 311-314).

[revised manuscript text omitted]

**Other minor questions:**

1. In the results, what error is being reported? Is it standard error, standard deviation, etc. and what are the sample numbers?
**Our response:** Thanks. More details on error have been added in the manuscript as follows: "*Values are presented as the mean ± standard deviation.*" (P8 Line 190). The information about samples were provided in the manuscript as follows: "*Surface water samples (n = 28) along the mainstem and major tributaries of the Yinjiang River, Shiqian River, and Yuqing River and spring water samples (n = 4) were collected in September 2018*" (P5 Line 124-125).

2. Do the statistical results shown in Figure 5 hold if Y12 is removed as an outlier?
**Our response:** There is no difference for the statistical results shown in Figure 5 if Y12 is removed. Further statistical tests have been done to avoid the likely skew of site Y12 to the results in the manuscript as follows: "*In addition, all the statistical analyses were performed again after data from site Y12 were removed to test the possible skew of findings as the sample was significantly affected by rainfall events. If not mentioned otherwise, the results from site Y12 did not skew the findings at the significance level of 0.05.*" (P8 Line 190-193).

3. How do the patterns (and statistically significances) shown between rivers and spring waters in Figures 2 and 3 change if Y12 is excluded? Do the results from Y12 skew the rest of the findings in any way?
**Our response:** Same to last question, we have performed additional statistical tests in the manuscript as follows: "*In addition, all the statistical analyses were performed again after data from site Y12 were removed to test the possible skew of findings as the*

*sample was significantly affected by rainfall events. If not mentioned otherwise, the results from site Y12 did not skew the findings at the significance level of 0.05.*" (P8 Line 190-193).

---

## Author Comment (AC2)

**Response to Anonymous Referee #2**

**Summary:**

This study assessed how geographic controls (elevation, temperature, and slope) and % anthropogenic land cover (urban/agriculture) influence DOC export and DOM composition from mountainous rivers. The data presented shows that increased %urban/agriculture cover in lower reaches (and shallower gradients) of these catchments results in higher DOC concentrations, where carbon isotopic signatures (13 and 14C-DOC) of DOC are more deplete, and DOM is less aromatic. I believe these findings are of interest to a broad community. However, I have a number of concerns that I would like the authors to address and some suggestions to improve their manuscript.

**Our response:** We thank the reviewer for the time and effort spent reviewing our paper. We also appreciate your valuable remarks and constructive suggestions, which will much improve the manuscript.

**Major:**

1. The SUVA254 values presented in this paper are typically $>5$ L mg $C^{-1}m^{-1}$. These values are extremely high when compared to blackwater riverine systems that typically have high aromaticity DOM (e.g., Holt et al., 2021; Spencer et al., 2010; Weishaar et al., 2003 and references there in). Please can you confirm how SUVA values were calculated (and include this information in text and/or DOC and absorbance data in a table). Were decadic or napierian absorbance measurements used for calculations? SUVA values should be calculated from decadic absorbance and if naperian values were used instead this may explain why the values here are so high (Hu et al., 2002; Spencer et al., 2014). Some justification of why SUVA values are seemingly so high would be useful here, and it would be good if these values were contextualized in relation to past work for mountainous rivers (and anthropogenically impacted catchments).

**Our response:** Thanks. The previous SUVA values were calculated from naperian values and thus why they are so high. We have re-calculated them from decadic absorbance. The related information has been added in the manuscript as follows: "*Decadic absorbance values were used to calculate absorption coefficients as below (Poulin et al., 2014): $a_{254} = Abs_{254}/L$. Where, $a_{254}$ is the absorption coefficients ($cm^{-1}$), $Abs_{254}$ is the absorbance at 254 nm, and L is the path length (cm). Specific UV absorbance at 254 nm ($SUVA_{254}$) was determined according to Weishaar et al. (2003; Table 1): $SUVA_{254} = a_{254}/DOC$.*" (P6 Line 159-164). The data on how SUVA values were calculated are listed in the table below:

| Site ID | DOC (mg L$^{-1}$) | Abs254 | a254 (cm$^{-1}$) | SUVA254 (L mg$^{-1}$ m$^{-1}$) |
|---------|------|--------|------|----------|
| Y1 | 1.2 | 0.041 | 0.041 | 3.4 |
| Y2 | 0.9 | 0.040 | 0.040 | 4.3 |
| Y3 | 0.9 | 0.035 | 0.035 | 3.8 |
| Y4 | 0.9 | 0.034 | 0.034 | 3.8 |
| Y5 | 0.7 | 0.031 | 0.031 | 4.4 |
| Y6 | 0.4 | 0.019 | 0.019 | 5.2 |
| Y7 | 0.6 | 0.030 | 0.030 | 4.7 |
| Y8 | 0.7 | 0.022 | 0.022 | 3.1 |
| Y9 | 1.3 | 0.037 | 0.037 | 2.8 |
| Y10 | 1.2 | 0.042 | 0.042 | 3.6 |
| Y11 | 1.5 | 0.037 | 0.037 | 2.5 |
| Y12 | 2.2 | 0.066 | 0.066 | 3.0 |
| Y13 | 2.0 | 0.040 | 0.040 | 2.0 |
| Y14 | 2.9 | 0.028 | 0.028 | 1.0 |
| Y15 | 1.1 | 0.040 | 0.040 | 3.5 |
| Y16 | 1.7 | 0.032 | 0.032 | 1.9 |
| S1 | 1.7 | n.a. | n.a. | n.a. |
| S2 | 1.1 | 0.041 | 0.041 | 3.8 |
| S3 | 1.4 | 0.101 | 0.101 | 7.0 |
| S4 | 1.9 | 0.055 | 0.055 | 3.0 |
| S5 | 1.7 | 0.047 | 0.047 | 2.7 |
| S6 | 1.4 | 0.020 | 0.020 | 1.4 |
| S7 | 1.5 | 0.041 | 0.041 | 2.8 |
| S8 | 1.5 | 0.022 | 0.022 | 1.4 |
| S9 | 1.4 | 0.040 | 0.040 | 2.8 |
| Q1 | 1.7 | 0.055 | 0.055 | 3.2 |
| Q2 | 1.7 | 0.047 | 0.047 | 2.8 |
| Q3 | 1.7 | 0.043 | 0.043 | 2.5 |

Note: n.a. means the data are not available.

In addition, these values were contextualized in relation to past work in the manuscript as follows: "*SUVA$_{254}$ values for the three study rivers were comparable with those reported in coastal glacier mountainous streams with late succession in southeast Alaska (3.4 ± 0.5 L mg$^{-1}$ m$^{-1}$, n = 5; Holt et al. 2021) and in the anthropogenic influenced downstream of the Yangtze River (3.4 ± 1.1 L mg$^{-1}$ m$^{-1}$, n = 82; Zhou et al. 2021).*" (P17 Line 388-391).

2. There are details missing from the methods in relation to how % land use, slope and elevation was calculated or whether this data is from the author's previous study. This information should be included in this paper since many of the figures and findings are reliant on this data. I would also recommend data from this analysis is presented and described within the site description before it is used to inform your analyses with DOM composition.

**Our response:** Thanks for your suggestions. We have added details on related information in "2.1 Study area" and figure about sites specific distribution of land use, slope and elevation in the SI as follows: "*Catchment characteristics (e.g., mean drainage elevation, annual air temperature, mean channel slope, and proportion of urban and agricultural land uses) for the sub-catchments were determined using ArcGIS (version 10.2). The mean drainage elevation of these three catchments ranges from 340 m to 2424 m with the lowest and highest elevation both reported in the Yinjiang River catchment, showing a great change in relief (Figs. 1a and S1a). Similar to elevation, the Yinjiang River catchment has a greater variation in mean channel slope (from 14.3° to 25.5°), while the channels in the Shiqian and Yuqing river catchments have a mean channel slope of approximately 20°, except the segment above site S8 (13.9°; Figs. 1b and S1b). Carbonate rock is widely distributed in the three catchments, accounting for a large proportion of the exposed strata (Han and Liu, 2004). The remaining areas are mainly covered by clastic rocks, igneous rocks, and low-grade metamorphic rocks. Forest, agriculture, and urban areas are the three dominant land uses in these studied catchments (Fig. 1c). Forest is generally distributed in high-elevation regions, while urban and agricultural land uses are mainly located in low-elevation regions. The proportion of urban and agricultural land uses in the Yinjiang and Shiqian river catchments is from 4.5% to 46.5% and from 9.6% to 41.3%, respectively, showing a large variation relative to the Yuqing River catchment (from 17.3% to 23.1%; Figs. 1c and S1c).*" (P4 Line 101-113).

[Figure]

**Figure S1** Distribution of (a) mean drainage elevation, (b) mean drainage slope, and (c) urban and agricultural land use in the Yinjiang (Y), Shiqian (S), and Yuqing (Q) catchments.

3. Findings of figure 5,6,7 are discussed without an initial description of the data/trends and thus much of the discussion comes across as a little abrupt. I would recommend you restructure and describe these figures/trends in the results section, then explain what these trends (either individually or collectively) may mean in the discussion. As it stands the discussion and results are a little brief and findings are not really discussed in detail (especially section 4.1). Given the structure it is also difficult to follow the primary reasons for the trends you observe. I wonder if you could examine trends within figures 5-7 collectively (e.g., through a PC analysis) so connections between land use and slope can be made and discussed in tandem rather than separately?

**Our response:** Thanks. We have restructured and described these figures/trends in the results section as you suggested. Details are in the section "**3 result**" of the revised manuscript (please see section 3.2 as an example shown below). We have tried to examine the trends through a PCA analysis, but it failed due to the limited number of $^{14}$C data. Instead, we examine the trends in a correlation plot (Fig. S3) as shown below to facilitate the further discussion on the trends.

**3.2. Riverine DOM Optical Properties**

The average SUVA$_{254}$ were 3.3 ± 1.1, 3.1 ± 1.8, and 2.8 ± 0.3 L mg$^{-1}$ m$^{-1}$ in the Yinjiang, Shiqian, and Yuqing rivers, respectively. Although no spatial differences in SUVA$_{254}$ were found across the three rivers (Fig. S4a), SUVA$_{254}$ showed an increasing trend with increasing mean channel slope (Fig. 3e), which indicated that DOM in the lower reaches with a gentle channel slope was less aromatic. Furthermore, there was a significant negative correlation between SUVA$_{254}$ and proportion of urban and agricultural land uses (Fig. 6a).

Two humic-like fluorescence components (C1 and C3) and one protein-like fluorescence component (C2) were identified by PARAFAC model in these three rivers (Fig. 7; Table S1). Component C1 is similar to traditionally defined peak M and sourced form microbial processes or autochthonous production (Kim et al., 2022; Li et al., 2016; Walker et al., 2009). Component C2 was previously related to recent biological production (DeFrancesco and Guéguen, 2021; Du et al., 2019; Lambert et al., 2017). C3 was the most widely found component in previous research among three fluorescence components, and was identified as traditional fulvic-like peaks A and C, representing terrestrial delivered OM or authochthonous microbial sourced OM (Amaral et al., 2016; Ryan et al., 2022; Shutova et al., 2014). Although C1 and C2 varied more widely in the Yinjiang River compared with the Shiqian and Yuqing rivers, the two fluorescence components did not show a statistical difference among the three rivers (Figs.

7a and b). However, a greater proportion of C3 was found in the Shiqian and Yuqing rivers, exhibiting a distinctive signature compared with the Yinjiang River (Fig. 7c). The fluorescence components did not exhibit any significant variations with changing channel slope (Fig. S3), but C1 and C2 were positively or negatively related to proportion of urban and agricultural land uses (Figs. 6b and c).

For the fluorescence indexes, the overall fluorescence property did not vary significantly among the three rivers (Figs. S4b, c, and d). FI varied in a narrow range compared with $\beta/\alpha$ and HIX. FI of DOM ranged from 1.66 to 1.94, averaging 1.78 (Fig. S4d), indicating a mixture of DOM of terrestrial and microbial origins. In comparison, $\beta/\alpha$ varied from 0.70 to 1.22 (Fig. S4b) and HIX varied from 0.33 to 0.65 (Fig. S4c), with greater variability among the three rivers. Although no significant correlation was observed between channel slope and the fluorescence indexes, they (except for FI) were found to be closely related to land use pattern (Figs. 6d, e, f, and S3). The less recently produced DOM ($\beta/\alpha$) in the urban and agricultural streams was also characterized by a higher proportion of C1 and lower proportion of C2 (Fig. S3).

[Figure]

**Figure S3** Correlation plot of the selected water chemistry and catchment characteristics. The colors represent the degree of pairwise correlation regarding Pearson's correlation coefficient. $\delta^{13}C_{DOC}$ and $\Delta^{14}C_{DOC}$ at site Y12 was excluded from analyses as the sample was collected after a rainfall event. In addition, $SUVA_{254}$ at site S3 was excluded from analyses as the sample was strongly influenced by the road construction, which was evidenced by high POC and TSM concentration (Chen et al., 2021). Human land use used here represents proportion of urban and agricultural land use. Elevation and Annual $T_{air}$ represent mean drainage elevation and annual air temperature, respectively.

4. L189-190 it is unclear how 'enhanced' biodegradation of DOC would increase DOC concentrations. I would have thought that biodegradation would remove DOC. Also, how can you be sure that this trend in 13C-DOC is microbially driven, rather than an increased input from aged soil/C3 plants in lower elevation stream reaches? You note in your site description that there is C3/C4 agriculture across the catchments. Is this coverage variable and could this in part drive the trend in 13C-DOC? Additionally, wouldn't removal of 12C by microbes lead to enrichment in 13C-DOC not depletion as you describe?

**Our response:** Thanks. Here, the enhanced biodegradation of DOC with the preferential removal of $^{12}C$ is the process happen in the soil layers or spring water. Thus, the remaining DOC would be $^{13}C$-enriched and input into the rivers. We have rewritten this part in the manuscript as follow:"*Previous studies have reported a decreasing $\delta^{13}C_{DOC}$ with a corresponding increase in DOC concentrations (Fig. 5a) in spring water (Nkoue Ndondo et al., 2020) and for TOC in soil profiles (Lloret et al., 2016; Nkoue Ndondo et al., 2020). This can be explained by the lateral transport of DOC from microbial active soil horizons into rivers (Lambert et al., 2011), resulting in the enhanced biodegradation of DOC with the preferential removal of $^{12}C$. As a result, the remaining DOC of lower concentrations is typically characterized by a heavier $\delta^{13}C_{DOC}$ (Nkoue Ndondo et al., 2020; Opsahl and Zepp, 2001).*" (P15 Line 329-334).

5. I wonder if the authors could further discuss how they arrived at the conclusion that groundwater was a significant source of DOC to these rivers, and that this groundwater played an important role in diluting DOC concentrations during base flow (1)

given that statistical analysis was not performed between springs and river DOC samples; (2) that samples were taken during heavy rainfall periods (i.e., monsoon; September) and thus baseflow conditions were unlikely to have been examined; (3) optical and isotopic data was not used to inform this discussion.

**Our response:** Thanks. We have further discussed this point and arrived at the new conclusion that groundwater is an important but not a crucial source of riverine DOC.

(1) Statistical analysis was added in the "2.4 Statistical analysis" and in the results as follow: "*The DOC concentrations in spring water were significantly lower than that in the surface water of the Shiqian and Yuqing rivers (Fig. 2d), and the average DOC concentration in spring water (0.74 ± 0.30 mg L$^{-1}$) was also lower than the average DOC concentration in the Yinjiang River (1.27 ± 0.66 mg L$^{-1}$), indicating there must be other sources of DOC besides groundwater.*" (P9 Line 222-225).

For question (2) and (3), we have rewritten this part and added information on $\delta^{18}O$ as evidence to further support our conclusion in the manuscript as follow: "*Groundwater with large SOC inputs due to highly active microbial activities has long been recognized as a significant source of DOC, especially under warm and wet conditions (McDonough et al., 2020; Shen et al., 2014). Several studies have reported increased groundwater contributions with distance downstream at the watershed scale (Asano et al., 2020; Cowie et al., 2017; Iwasaki et al., 2021), suggesting the important role of groundwater in mountainous rivers, though this is not a general phenomenon due to great differences in catchment characteristics (e.g., bedrock and topography) and climate (e.g., precipitation; Somers and McKenzie, 2020). The positive relationship between conductivity and $\delta^{18}O$ are due largely to the mixing of two end-members for river water (Fig. S2a), though it may also indicate the evaporation processes in the catchment (Zhong et al., 2020). The trend showed that one of the end-members was characterized by high-conductivity with $^{18}O$-enriched (groundwater), and the other was characterized by low-conductivity with $\delta^{18}O$-depleted headstream water. Similar relationships were also identified in other mountainous rivers (Lambs, 2004). In addition, $\delta^{18}O$ values increased progressively from upstream to downstream (Fig. S2b), which also validates the two sources (i.e., headstream water, and groundwater) of downstream river water, indicating that groundwater was likely an important contributor to downstream river water. Despite that groundwater inflow seemed to be of great importance to river water, groundwater DOC was likely not the most important source of riverine DOC due to the relatively low groundwater DOC concentrations as compared with riverine DOC concentrations (Fig. 2d). Moreover, the groundwater contribution was probably much less significant in the wet season, even in catchments where DOC is mainly derived from groundwater (Lloret et al., 2016). Thus, we infer that groundwater is an important but not a crucial source of riverine DOC in the three study rivers.*" (P15-16 Line 335-351).

[Figure]

**Figure S2** Implications for water from deeper flow paths. (a) Relationship between EC and $\delta^{18}O$ for the river water and spring water. This relationship represents the mixing of the two reference waters: groundwater and upstream river water. (b) Variations of $\delta^{18}O$ for water in the mainstream of the Yinjiang River. The $\delta^{18}O$ value used in panel b for site Y12 is from the sample we collected before rainfall. The $\delta^{18}O$ value for site Y15 was influenced by rainfall. The statistical test used a significance level of 0.05.

6. Fluorescence properties of DOM are presented but not related to geographic or anthropogenic features of the catchments. It is unclear why this isn't considered in the manuscript, and I wonder if there are any trends observed? At least, the information gained from optical analyses should be explained and contextualized within the discussion.

**Our response:** Thanks for your suggestions. Fluorescence properties of DOM were related to anthropogenic features of the catchments in the SI of our original manuscript. We have restructured the related information and added it in the manuscript as shown in the Figure 3, 6 and S3. In addition, the information has been explained and contextualized in the manuscript as follow: "*For the fluorescence indexes, the overall fluorescence property did not vary significantly among the three rivers (Figs. S4b, c, and d). FI varied in a narrow range compared with β/α and HIX. FI of DOM ranged from 1.66 to 1.94, averaging 1.78 (Fig. S4d), indicating a mixture of DOM of terrestrial and microbial origins. In comparison, β/α varied from 0.70 to 1.22 (Fig. S4b) and HIX varied from 0.33 to 0.65 (Fig. S4c), with greater variability among the three rivers. Although no significant correlation was observed between channel slope and the fluorescence indexes, they (except for FI) were found to be closely related to land use pattern (Figs. 6d, e, f, and S3). The less recently produced DOM (β/α) in the urban and agricultural streams was also characterized by a higher proportion of C1 and lower proportion of C2 (Fig. S3).*" (P14 Line 296-302) and "*However, DOM characters were found to be closely related to land use pattern (Fig. 6). Although significant relationships with urban and*

*agricultural land uses were found for C1 and C2 (Figs. 6b and c), it remains unclear how the autochthonous contribution to riverine DOC pool varied with land use change because C1 and C2 are both likely derived from autochthonous production but exhibit opposing trends with increasing urban and agricultural land uses. Overall DOM in catchments with a higher proportion of urban and agricultural land use area was distinct from other catchments as it was less aromatic (SUVA$_{254}$, Fig. 6a), less recently produced (β/α, Fig. 6d), and had a higher degree of humification (HIX, Fig. 6e). SUVA$_{254}$ values for the three study rivers were comparable with those reported in coastal glacier mountainous streams with late succession in southeast Alaska (3.4 ± 0.5 L mg$^{-1}$ m$^{-1}$, n = 5; Holt et al. 2021) and in the anthropogenic influenced downstream of the Yangtze River (3.4 ± 1.1 L mg$^{-1}$ m$^{-1}$, n = 82; Zhou et al. 2021). Lower DOM aromaticity in the urban and agricultural streams and rivers was consistent with previous studies (Hosen et al., 2014; Kadjeski et al., 2020), though it was not a universal phenomenon (Zhou et al., 2021). Furthermore, the less aromatic and recently produced DOM could be due to soil organic materials from deep soil profiles most aged DOC because of the increased soil erosion rates associated with anthropogenic activities (Inamdar et al., 2011; Stanley et al., 2012)."* (P17 Line 382-395).

[Figure]

**Figure 6**. Land use pattern impacts on DOM character. (a) SUVA254, (c) C2, and (d) freshness index (β/α) decreased with increasing proportion of urban and agricultural land use area in the studied catchments. Outlier (site S3) was excluded from analyses in panel a as the sample was strongly influenced by the road construction, which was evidenced by high POC and TSM concentration (Chen et al., 2021). (b) C1 and (e) humification index (HIX) positively related to the increasing proportion of urban and agricultural land use area.

[Figure]

**Figure S3** Correlation plot of the selected water chemistry and catchment characteristics. $\delta^{13}C_{DOC}$ and $\Delta^{14}C_{DOC}$ at site Y12 was excluded from analyses as the sample was collected after a rainfall event. In addition, $SUVA_{254}$ at site S3 was excluded from analyses as the sample was strongly influenced by the road construction, which was evidenced by high POC and TSM concentration (Chen et al., 2021). Human land use used here represents proportion of urban and agricultural land use. Elevation and Annual $T_{air}$ represent mean drainage elevation and annual air temperature, respectively.

7. Similarly, it is unclear in section 4.3 why anthropogenic impacts are only discussed in relation to DOC concentration. I wonder if you can draw a connection with carbon isotopes (and optical properties) and if this could help you understand the primary drivers of variability within your dataset.

**Our response:** Same to last question, anthropogenic impacts on carbon isotopes and optical properties have been provided in Figure 6 and S3 (see above) and related discussion in "4.2 anthropogenic impacts on DOC" (please see the details in the revised manuscript or last question).

[Figure]

Figure 6. Land use pattern impacts on DOM character. (a) SUVA254, (c) C2, and (d) freshness index (β/α) decreased with increasing proportion of urban and agricultural land use area in the studied catchments. Outlier (site S3) was excluded from analyses in panel a as the sample was strongly influenced by the road construction, which was evidenced by high POC and TSM concentration (Chen et al., 2021). (b) C1 and (e) humification index (HIX) positively related to the increasing proportion of urban and agricultural land use area.

**Minor comments and technical/typographical corrections:**

Abstract:
L19 'of DOC' seems a little awkward here – consider rewording
**Our response:** Thanks. We have reworded it as: "*Water chemistry, stable and radioactive carbon isotopes of DOC ($\delta^{13}C_{DOC}$ and $\Delta^{14}C_{DOC}$) and optical properties (UV absorbance and fluorescence spectra) were employed to assess the biogeochemical processes and controlling factors on riverine DOC.*" (P1 Line 17-19).

L21 POC is not defined. It is not clear how instream processing of POC is a source of DOC in this sentence. I would also make it clearer that you are using POC values from past work within the abstract and aims/objectives.
**Our response:** Thanks. POC has been defined in the manuscript as follow: "Our prior observations from these catchments showed that particulate organic carbon (POC) dynamics were highly affected by in-stream photosynthesis." (P3 Line 86-88). The point that "instream processing of POC is an important source of DOC" has been discarded in the new manuscript as you and other reviewers discussed. Therefore, we have removed this sentence.

L23-24 consider making the distinction between DOC and DOM here as I think it would help with sentence flow. It is also unclear how this was 'distinct' from those catchments with lower slopes/higher temperatures.
**Our response:** Thanks. The sentence is rephrased in the revised manuscript as: "*Catchments with higher channel slope gradients and lower annual air temperature were characterized by lower DOC concentrations, enriched $\delta^{13}C_{DOC}$ and $\Delta^{14}C_{DOC}$, and more aromatic dissolved organic matter (DOM), which were opposite to those with gentle channel slopes and higher temperature.*" (P1 Line 20-22).

L25 DOM is not defined
**Our response:** Thanks. We have added definition of DOM (dissolved organic matter) in the revised manuscript (P1 Line 21).
L28 I think you could make the significance more specific/explicit here in relation to your findings.
**Our response:** Thanks. We have modified this sentence to make the significance more specific in the revised manuscript as: "*This research highlights the significance of incorporating geographical controls on DOC sources and anthropogenic impacts on DOM composition into the understanding of DOC dynamics and quality of DOM in mountainous rivers which are globally abundant.*" (P1 Line 25-27).

Introduction:

1. Generally, within the introduction I think it would be useful to provide more specific details in terms of DOM compositional shifts that have been noted with warming and land use changes as well as across geographic gradients (i.e., elevation and slope). Similar to that in line 63. Much of your description only specifies that there are 'changes' but doesn't note the typical direction of change. I think this would make it easier on the reader later when reading the results/discussion as many of the changes would be somewhat familiar and would also situate your study more firmly in relation to past work.
**Our response:** Thanks. We have provided more specific details on DOM compositional shifts with warming and land use changes as well as across geographic gradients in the manuscript as follow: "*A recent global study found that increasing elevation was associated with greater protein-like fluorescence DOM and lower specific ultraviolet absorbance at 254 nm (SUVA$_{254}$), which indicate the effect of enhanced UV radiation and accumulation of autochthonous DOM at higher elevation areas with low temperatures (Zhou et al., 2018).*" (P2 Line 51-53) and "*On the scale of decades to centuries, however, anthropogenic impacts would shift natural DOM to forms of low-molecular weight, enhanced redox state with potentially increased lability, or increased aromaticity due to warmer climate and altered hydrology (Stanley et al., 2012; Xenopoulos et al., 2021). In addition, a warmer climate can enhance microbial activity and in-stream production of DOM (He et al., 2016), and may simultaneously increase the microbial degradation rate of soil DOM, thus reducing the potential input of DOM into rivers (Voss et al., 2015). Consequently, this will increase the relative contribution of autochthonous DOM. Nevertheless, elevated temperature can also enhance photo-chemical degradation and reduce autochthonous microbial humic-like DOM (Henderson et al., 2009; Zhou et al., 2018), thus potentially limiting the accumulation of autochthonous DOM in inland waters. These two seemingly contradictory findings are due to the complex effect of temperature on organic matter.*" (P3 Line 73-81).
In addition, we have specified the typical direction of many changes in the manuscript, such as: "*For example, elevated temperature has a dominant effect on DOC concentration and dissolved organic matter composition by enhancing decomposition and photochemical degradation rates of DOM (Zhou et al., 2018), contributing to significant $CO_2$ emissions from inland waters (Raymond et al., 2013).*" (P2 Line 33-35) and "*Compared with high-relief catchments, low relief regions with longer water residence time, stronger hydrologic connectivity to rivers, and greater development of wetlands are typically characterized by greater releases of DOC (Harms et al., 2016; McGuire et al., 2005).*" (P2 Line 48-50).

2. I would also suggest you integrate the points made in lines 68-72 into the previous paragraphs. This section appears a little obvious and doesn't really make it clear why there is utility in using these techniques within your study.
**Our response:** Thanks. We have integrated the points into the previous paragraphs in the manuscript as you suggested: "*Recent advances in spectroscopic techniques, especially the UV-visible spectrophotometry and fluorescence spectroscopy, and widespread application of stable and radiocarbon isotopes on bulk DOC have provided insights into the composition, source, and age of DOM in freshwater ecosystems (Coble, 1996; Fellman et al., 2010; Marwick et al., 2015; Minor et al., 2014; Sanderman et al., 2009). These new techniques have led to significant improvements in our understanding of the biogeochemical processes of DOC in river systems*, which will continue to be effective tools for researchers to gain deeper insights into the riverine carbon

cycle" (P1 Line 40-46).

3. Finally, within the aims and objective paragraph it would be useful to be more specific of the techniques you are using (e.g., DOM quality assessed through optical metric) and the geographic/land use parameters use are assessing against DOM quality/DOC concentration.
**Our response:** Thanks. We have added related information to make it more specific in the revised manuscript as: "*In this study, we evaluated how geographical controls (i.e., mean channel slope, mean drainage elevation, and annual air temperature) and anthropogenic impacts (i.e., land use patterns) affect the DOC dynamics and DOM characteristics in three subtropical catchments that contain many small to medium mountainous rivers in southwest China.*"(P3 Line 84-86) and "*Relationships of DOC concentrations, stable and radiocarbon isotopic values of DOC (radiocarbon isotopes were only available for nine sampling sites in the Yinjiang River), DOM quality assessed through optical metric, nutrient concentrations, and land use patterns versus geographical characteristics (i.e., mean channel slope, mean drainage elevation, and annual air temperature) were examined.*" (P3-4 Line 91-95).

L33 given that your study has a large land use component, I'd suggest broadening this sentence out to encompass this.
**Our response:** Thanks. We have broadened this sentence in the manuscript as: "*Owing to continued climate warming and rapid land use changes, it is important to gain a better understanding of the spatial and temporal dynamics of DOC transport in river systems (Butman et al., 2014; Fasching et al., 2016; Zhong et al., 2021).*" (P1 Line 31-33).

L43 can you explain the 'difference' more specifically?
**Our response:** Thanks. We have modified this sentence as: "*Compared with high-relief catchments, low relief regions with longer water residence time, stronger hydrologic connectivity to rivers, and greater development of wetlands are typically characterized by greater releases of DOC (Harms et al., 2016; McGuire et al., 2005).*" (P2 Line 48-50).

L53 this sentence is a little unclear to me. Consider rewording.
**Our response:** Thanks. We have reworded this sentence as: "*Subtropical small mountainous rivers are characterized by steep channel slopes, high erosion rates, frequent rainfall events in wet seasons, and rapid change in hydrology during these rainfall events (Lee et al., 2019; Leithold et al., 2006; Qiao et al., 2019), however, the DOC dynamics in these rivers remain poorly studied.*" (P1 Line 60-62).

L65 consider rephrasing/reordering the sentence – 'recent pursuit' is a little awkward.
**Our response:** Thanks. We have rewritten this sentence as: "*These two seemingly contradictory findings are due to the complex effect of temperature on organic matter. Clearly, it remains largely unknown how these impacts have regulated riverine DOC dynamics due to their complex regulating mechanisms and changing influencing factors.*" (P3 Line 80-83).

L73 please specify the geographic and anthropogenic factors you are assessing
**Our response:** Thanks. The sentence was modified to specify the geographic and anthropogenic factors in the manuscript as follow: "*In this study, we evaluated how geographical controls (i.e., mean channel slope, mean drainage elevation, and annual air temperature) and anthropogenic impacts (i.e., land use patterns) affect the DOC dynamics and DOM characteristics in three subtropical catchments that contain many small to medium mountainous rivers in southwest China.*" (P3 Line 84-86).

L75 remove 'their'.
**Our response:** Thanks. We have removed 'their' in this sentence from the revised manuscript: "*Our prior observations from these catchments showed that particulate organic carbon (POC) dynamics were highly affected by in-stream photosynthesis.*" (P3 Line 86-88).

L76 add 'here'. So, it reads 'Here, we investigate…'
**Our response:** Thanks. We have deleted this sentence as we did not take autochthonous processes as a significant source of DOC in the revised manuscript.

L76 it is unclear from this sentence how you asses autochthonous processes.
**Our response:** Thanks. Same to the last question, we have deleted this sentence as autochthonous processes were not a significant source of DOC in the revised manuscript.

L77 I feel this hypothesis could be more specific based on past literature.
**Our response:** Thanks. The hypothesis was modified to be more specific in the revised manuscript as: "*We hypothesize that catchments with a higher proportion of agricultural and urban land use, more gentle channel slope, and lower elevation would exhibit higher riverine DOC concentrations. We further hypothesize that there will be a large difference on DOM quality and carbon isotopes between these catchments and those with less influences by agricultural and urban land uses but steeper channel slopes and higher elevation.*" (P3 Line 88-91).

L80 seems a little obvious, can you be more specific on the insight gained?
**Our response:** Thanks. We have modified this sentence as: "*This study allows us to gain a deeper insight into the geographical controls and anthropogenic impacts on the DOC dynamics and DOM quality in the subtropical, anthropogenically influenced small mountainous rivers.*" (P4 Line 95-96).

Methods:

1. It appears from the results that 14C-DOC values only available for the Yinjiang River. This must be made clear in the aims and methods. Also, why is this the case?

**Our response:** Thanks. We have added related information in the aims and methods in the manuscript as follow: "*Relationships of DOC concentrations, stable and radiocarbon isotopic values of DOC (radiocarbon isotopes were only available for nine sampling sites in the Yinjiang River), DOM quality assessed through optical metric, nutrient concentrations, and land use patterns versus geographical characteristics (i.e., mean channel slope, mean drainage elevation, and annual air temperature) were examined.*" (P3-4 Line 91-95) and "*In this study, nine water samples collected from the Yinjiang River were selected for $\Delta^{14}C_{DOC}$ analysis as the Yinjiang River catchment has the greatest change in geographical characteristics (i.e., elevation and channel slope) and the highest proportion of agricultural and urban land uses among the three catchments.*" (P6 Line 150-152). "*The Yinjiang River catchment has the greatest change in geographical characteristics (i.e., elevation and channel slope) and the highest proportion of agricultural and urban land uses among the three catchments*" is one of the reasons we only collect 14C-DOC data in the Yinjiang River. The other reason is the expensive analytical cost (McNichol and Aluwihare, 2007), which is about $500 per sample.

Reference: *McNichol A. P. and Aluwihare, L. I.: The power of radiocarbon in biogeochemical studies of the marine carbon cycle: Insights from studies of dissolved and particulate organic carbon (DOC and POC), Chem. rev., 107, 443-466, doi:10.1002/chin.200724246, 2007.*

L86 'of the' replace with 'in'

**Our response:** Thanks. 'Of the' has been replaced with 'in' in the manuscript as follow: "*The mean drainage elevation of these three catchments ranges from 340 m to 2424 m with the lowest and highest elevation both reported in the Yinjiang River catchment, showing a great change in relief.*" (P4 Line 103-104).

L89 rephrase so land use is not repeated

**Our response:** Thanks. We have rephrased the sentence as: "*Forest, agriculture, and urban areas are the three dominant land uses in these studied catchments.*" (P4 Line 109-110)

L104 there is a missing word here.

**Our response:** Thanks. We have reworded this sentence as: "*water samples were filtered through 0.45 μm cellulose acetate membranes.*" (P5 Line 131).

L108 I think more information would be useful here, despite information being published in your previous work.

**Our response:** Thanks. We have added more details in the revised manuscript as: "*Water samples were also filtered for determining dissolved inorganic carbon (DIC) through titration with hydrochloric acid and analyzing POC using retained suspended particles on the filter membranes. Moreover, the water samples filtered through 0.22 μm cellulose-acetate filter membranes were used to determine water isotopes ($\delta^{18}O$ and $\delta D$). Detailed information on sampling methods was provided in Chen et al. (2021) and Zhong et al. (2020).*" (P5 Line 135-138).

L127 replace substances with fluorescence

**Our response:** Thanks.We have replaced it as you suggested: "*The fate of humic-like fluorescences may be self-assembly particles or be adsorbed onto minerals, while protein-like fluorescences are tightly associated with biological processes, and biodegraded into inorganic matter*" (P7 Line 167-169).

L135 – double brackets to be replaced with ';' - please check throughout.

**Our response:** Thanks. We have replaced double brackets with ';' throughout the manuscript. Here are two examples: "*Several common indices of DOM composition were determined from EEMs, including fluorescence index (FI; McKnight et al., 2001), humification index (HIX; Ohno, 2002), and freshness index (β/α; Parlanti et al., 2000; Table 1).*" (P7 Line 175-177) and "*Global average $\Delta^{14}C_{DOC}$ is -11.5±134‰, higher than that in the Yinjiang River (-54.7±39.9‰; Tables 2 and 3; Marwick et al., 2015) while similar to many other mountainous rivers (e.g., the Mackenzie River; Campeau et al., 2020) and small mountainous rivers in Puerto Rico; Moyer et al., 2013). $\Delta^{14}C_{DOC}$ values for the world's mountainous streams and rivers were shown by climate (according to the Köppen–Geiger climate classification (Beck et al., 2018; Table 3) and ranged from tropical monsoon climate (Marwick et al., 2015.*" (P19 Line 434-439).

L140 how was proportion of different land uses/elevation/slope calculated? Was this data previously published? I would recommend adding this information as a table to the text and how this data came about in the site description.

**Our response:** Thanks. We have added details on related information in "2.1 Study area" in the manuscript as: "*Catchment characteristics (e.g., mean drainage elevation, annual air temperature, mean channel slope, and proportion of urban and agricultural land uses) for the sub-catchments were determined using ArcGIS (version 10.2). The mean drainage elevation of these three catchments ranges from 340 m to 2424 m with the lowest and highest elevation both reported in the Yinjiang River catchment, showing a great change in relief (Figs. 1a and S1a). Similar to elevation, the Yinjiang River catchment has a greater variation in mean channel slope (from 14.3°to 25.5°), while the channels in the Shiqian and Yuqing river catchments have a mean channel slope of approximately 20°, except the segment above site S8 (13.9°; Figs. 1b and S1b). Carbonate rock is widely distributed in the three catchments, accounting for a large proportion of the exposed strata (Han and Liu, 2004). The remaining*

*areas are mainly covered by clastic rocks, igneous rocks, and low-grade metamorphic rocks. Forest, agriculture, and urban areas are the three dominant land uses in these studied catchments (Fig. 1c). Forest is generally distributed in high-elevation regions, while urban and agricultural land uses are mainly located in low-elevation regions. The proportion of urban and agricultural land uses in the Yinjiang and Shiqian river catchments is from 4.5% to 46.5% and from 9.6% to 41.3%, respectively, showing a large variation relative to the Yuqing River catchment (from 17.3% to 23.1%; Figs. 1c and S1c).*" (P4 Line 101-113) and figure about sites specific distribution of land use, slope and elevation in the SI (please see the figure S1).

L161 – the median value for river Y is not higher than the other rivers. Thus, it's unclear what you are referring to here.
**Our response:** Thanks. Here, "*DOC concentrations in the three study rivers varied from 0.36 to 2.85 mg L$^{-1}$ with the highest average concentrations in the Yuqing River (1.70 ± 0.04 mg L$^{-1}$; Fig. 2d)*" (P9 Line 221-222) means the average concentrations in the Yuqing River (Q) were the highest, not the Yinjiang River (Y).

Results/Discussion:

1. Description of optical properties in the results is extremely brief and lacks quantitative details. E.g., what are the average SUVA values for each river? What is the % each component of fluorescence is explaining? Please include these details.
**Our response:** Thanks for your suggestions. We have added these details in the manuscript as follow: "*The average SUVA$_{254}$ were 3.3 ± 1.1, 3.1 ± 1.8, and 2.8 ± 0.3 L mg$^{-1}$ m$^{-1}$ in the Yinjiang, Shiqian, and Yuqing rivers, respectively. Although no spatial differences in SUVA$_{254}$ were found across the three rivers (Fig. S4a), SUVA$_{254}$ showed an increasing trend with increasing mean channel slope (Fig. 3e), which indicated that DOM in the lower reaches with a gentle channel slope was less aromatic. Furthermore, there was a significant negative correlation between SUVA$_{254}$ and proportion of urban and agricultural land uses (Fig. 6a).*" (P12 Line 264-268) and "*Two humic-like fluorescence components (C1 and C3) and one protein-like fluorescence component (C2) were identified by PARAFAC model in these three rivers (Fig. 7; Table S1). Component C1 is similar to traditionally defined peak M and sourced form microbial processes or autochthonous production (Kim et al., 2022; Li et al., 2016; Walker et al., 2009). Component C2 was previously related to recent biological production (DeFrancesco and Guéguen, 2021; Du et al., 2019; Lambert et al., 2017). C3 was the most widely found component in previous research among three fluorescence components, and was identified as traditional fulvic-like peaks A and C, representing terrestrial delivered OM or authochthonous microbial sourced OM (Amaral et al., 2016; Ryan et al., 2022; Shutova et al., 2014). Although C1 and C2 varied more widely in the Yinjiang River compared with the Shiqian and Yuqing rivers, the two fluorescence components did not show a statistical difference among the three rivers (Figs. 7a and b). However, a greater proportion of C3 was found in the Shiqian and Yuqing rivers, exhibiting a distinctive signature comprred with the Yinjiang River (Fig. 7c). The fluorescence components did not exhibit any significant variations with changing channel slope (Fig. S3), but C1 and C2 were positively or negatively related to proportion of urban and agricultural land uses (Figs. 6b and c).*" (P13 Line 277-288).

2. Generally, geographic and land use parameters are not discussed in the results. However, SUVA is briefly described and then related to slope. Given the structure of the results it would make sense to wait to draw the comparison with slope. Consider including a summary figure (e.g., boxplot for SUVA) and then including SUVA v slope analysis. Similarly, why is Figure 4 a boxplot whilst other relationships with slope conducted as linear regressions?
**Our response:** Thanks. We have added more details on geographic and land use parameters in the manuscript as follow: "*Catchment characteristics (e.g., mean drainage elevation, annual air temperature, mean channel slope, and proportion of urban and agricultural land uses) for the sub-catchments were determined using ArcGIS (version 10.2). The mean drainage elevation of these three catchments ranges from 340 m to 2424 m with the lowest and highest elevation both reported in the Yinjiang River catchment, showing a great change in relief (Figs. 1a and S1a). Similar to elevation, the Yinjiang River catchment has a greater variation in mean channel slope (from 14.3° to 25.5°), while the channels in the Shiqian and Yuqing river catchments have a mean channel slope of approximately 20°, except the segment above site S8 (13.9°; Figs. 1b and S1b).*" (P4 Line 101-107) and "*The proportion of urban and agricultural land uses in the Yinjiang and Shiqian river catchments is from 4.5% to 46.5% and from 9.6% to 41.3%, respectively, showing a large variation relative to the Yuqing River catchment (from 17.3% to 23.1%; Figs. 1c and S1c).*" (P4 Line 111-113)
In addition, "SUVA vs slope" has been included in Figure 3 (see below) in the revised manuscript and all the DOM property data were plot as a boxplot (Figure S4; see below).

[Figure]

**Figure 3.** Mean drainage slope (°) controls on (**a**) DOC concentrations, (**b**) stable carbon isotopes of DOC ($\delta^{13}C_{DOC}$), (**c**) radiocarbon isotope of DOC ($\Delta^{14}C_{DOC}$), (**d**) radiocarbon isotope of POC ($\Delta^{14}C_{POC}$) and (**e**) SUVA$_{254}$. The $\Delta^{14}C_{DOC}$ and $\Delta^{14}C_{POC}$ are only available for the Yinjiang River. Outliers in orange were excluded from analyses as they were samples at site Y12 collected after a rainfall event in panels (**c**) and (**d**) and sample collected at site S3 due to the highly influence by the road construction, which was evidenced by high POC and TSM concentration (Chen et al., 2021). The statistical test used a significance level of 0.05.

[Figure]

**Figure S4** Spatial variations in DOM property in the Yinjiang (Y), Shiqian (S), and Yuqing (Q) catchments. (a) SUVA$_{254}$, (b) freshness index (β/α), (c) HIX, and (d) fluorescence index. In each box plot, the end of the box represents the 25th and 75th percentiles, the blue solid dot represents the average, the horizontal red line represents the median, and whiskers represent 1.5 IQR. The magenta solid dot represents the outlier, which is outside of the 1.5 interquartile ranges. Different lowercase letters above the boxes denote significant differences across rivers based on statistical analysis with $p<0.05$.

Figure 4 – specify units of SUVA254 on axis
**Our response:** Thanks. We have added the unit of SUVA$_{254}$ on axis (please see the above Figure 3 and S4).

Figure 5 – place key at the bottom of the four panels. Dots in key maybe confused with datapoints.

**Our response:** Thanks. To avoid the confusion, we have added a box for the key in Figure 5 (see below).

[Figure]

**Figure 5.** Scatter plot showing (**a**) $\delta^{13}C_{DOC}$ versus DOC in river water, (**b**) $\delta^{13}C_{DOC}$ against $\delta^{13}C_{POC}$ in the Yinjiang River (Y), Shiqian River (S), and Yuqing River (Q), and (**c**) relationship between $\Delta^{14}C_{POC}$ and $\Delta^{14}C_{DOC}$ in the Yinjiang River. For panel (**c**) the DOC with modern age at site Y12 was shown in the top-right corner. The statistical test used a significance level of 0.05.

Figure 5 – why are results in panel A reported as 1/DOC? Please just report as DOC since this leads to confusion in text e.g., line 188.
**Our response:** Thanks. We have modified it to be reported as DOC in Figure 5 (please see figure above). The reason for using 1/DOC in the original manuscript is because previous studies have reported similar relationships (i.e., $\delta^{13}C_{DOC}$ vs 1/DOC) in *"spring water (Nkoue Ndondo et al., 2020) and for TOC in soil profiles (Lloret et al., 2016; Nkoue Ndondo et al., 2020)"* (P15 Line 329-332) to explain the lateral transportation of DOC from microbial active soil horizons into rivers (Lambert et al., 2011)."

L166 it is unclear why you contextualize this finding but not others. Please contextualize data throughout results or move this point to the discussion.
**Our response:** Thanks. We have moved this point to the discussion (P19 Line 434).

L225 If anthropogenic activities were not the primary source of aged DOC in your catchments (as you imply). What is the primary source of aged DOC? This point should be made clearer
**Our response:** Thanks. We have added related information in the revised manuscript as follow: *"In comparison, when relatively $^{14}$C-depleted DIC and $CO_2$ (aq) derived from carbonate weathering is incorporated into primary production in low relief regions, it would also produce aged organic carbon (Fig. 3d; Chen et al., 2021). Furthermore, aged DOC in river systems has been attributed to old soil organic matter in deeper layer input into rivers through deeper flow paths (Barnes et al., 2018; Masiello and Druffel, 2001). This also indicates that low relief regions with higher hydrologic connectivity in river network are likely the major source of riverine DOC (Connolly et al., 2018; Mzobe et al., 2020)."* (P15 Line 309-314) and *"Furthermore, the weak positive correlation between the $\delta^{13}C_{DOC}$ and $\delta^{13}C_{POC}$ of these three rivers (Fig. 5b) indicated that DOC and POC may derive from the same source. We also found a strong positive correlation between $\Delta^{14}C_{DOC}$ and $\Delta^{14}C_{POC}$ in the Yinjiang River ($R^2 = 0.67$, $p < 0.01$; Fig. 5c). The coupling between $\Delta^{14}C_{DOC}$ and $\Delta^{14}C_{POC}$ was an unusual relationship rarely found in other rivers (Campeau et al., 2020; Longworth et al., 2007). Campeau et al. (2020) have attributed this relationship to common controls of landscape and/or hydrology on the sources of organic carbon in rivers, and this correlation could be masked by the mixing of waters from other tributaries, which further support the combined geographical controls (e.g., as a driver of landscape) and anthropogenic impacts (e.g., dam construction, see discussion below) on DOC sources."* (P18 Line 415-422).

L230 it would be prudent to explain the two endmembers here in more detail.
**Our response:** Thanks. As we have modified the original view (two endmembers mixing) due to further discussion. We have deleted the figure which showed relationships between $\delta^{13}C_{DOC}$ and $\Delta^{14}C_{DOC}$, and take it as common controls of landscape and/or hydrology on the sources of organic carbon. Please see related discussion in the revised manuscript as: *"Furthermore, the weak positive correlation between the $\delta^{13}C_{DOC}$ and $\delta^{13}C_{POC}$ of these three rivers (Fig. 5b) indicated that DOC and POC may derive from the same source. We also found a strong positive correlation between $\Delta^{14}C_{DOC}$ and $\Delta^{14}C_{POC}$ in the Yinjiang River ($R^2 = 0.67$, $p < 0.01$; Fig. 5c). The coupling between $\Delta^{14}C_{DOC}$ and $\Delta^{14}C_{POC}$ was an unusual relationship rarely found in other rivers (Campeau et al., 2020; Longworth et al., 2007). Campeau et al. (2020) have attributed this relationship to common controls of landscape and/or hydrology on the sources of organic carbon in rivers, and this correlation could be masked by the mixing of waters from other tributaries, which further support the combined geographical controls (e.g., as a driver of landscape) and anthropogenic impacts (e.g., dam construction, see discussion below) on DOC sources."* (P18 Line 415-422)

L255 is this supported by your data?
**Our response:** Thanks. We just compare it with our study, rather than showing the views from our data. In order to avoid the misunderstanding of the sentence, we have modified it as: *"Channel slope in a number of Arctic watersheds has also been found to be positively associated with the FI, which was possibly due to the differences in soil characteristics (e.g., volumetric water content and soil temperature) and the resulting changing extent of microbial processing (Harms et al., 2016). Moreover, channel slope was negatively associated with terrestrial humic-like organic material due to the effects of climate and organic layer*

*thickness (Harms et al., 2016). However, there were no similar correlation between channel slope and fluorescence components/indexes in this study, demonstrating the likely complicated mechanisms (e.g., soil property, catchment characteristics, and anthropogenic activities) in regulating DOM. DOC in low relief regions was characterized by more $^{13}C$ depleted values, which may be due to the greater inputs of C3-derived organic carbon (e.g., from rice).*" (P15 Line 319-327).

---

## Author Comment (AC3)

**Response to Anonymous Referee #3**

**General Comments:**

This study characterizes the geomorphological controls and anthropogenic effects on DOM dynamics in small mountainous rivers. There are some interesting results and discussion that would make this manuscript a good contribution to the literature. However, there are important methodological details that need to be included or rectified before publication. Further, the discussion is lacking a clear structure, and I think the manuscript would be much improved by the removal of tangential and speculative discussion.

Allochthonous and autochthonous are incorrectly defined here. Autochthonous does not mean "microbial," but rather, produced within the river (e.g., from primary producers within the aquatic system). Allochthonous means organic C produced outside the river, but can also be "microbial" in composition (for example, if the DOM is derived from soils that have undergone significant microbial degradation).

**Our response:** We appreciate the reviewer for the time and effort spent reviewing our paper. We also thank to the reviewer for the affirmation of results and discussion. We have modified the manuscript as you suggested (see details on our response in specific comments). After revision, we believe the manuscript was much improved.

**Specific Comments:**

Methods: There are many important details missing in the methodology, particularly in the laboratory analysis section. This makes it challenging to assess whether the methodology in this study is sound. See specific comments below.

**Please include methodology on how land-use and slope are calculated.**

**Our response:** Thanks. We have added details on how land-use and slope are calculated in the manuscript as: *"Catchment characteristics (e.g., mean drainage elevation, annual air temperature, mean channel slope, and proportion of urban and agricultural land uses) for the sub-catchments were determined using ArcGIS (version 10.2)." (P4 Line 101-102).*

**Ln 100: What were samples collected with? Acid-washed or baked equipment?**

**Our response:** Thanks. Details on sample collection have been added in the manuscript as follow: "*The filtered water* was stored in a Milli-Q water pre-washed low-density polyethylene container at low temperature (4°C) in the dark before optical properties analysis and acidified by phosphoric acid to pH=2 for DOC analysis." (P5 Line 133-135).

Ln 106: How long were samples stored before optical measurement? 0.7 µm pore size allows microbes into filtrate, so long storage times (even at 4°C) can be problematic.

**Our response:** Thanks. Refrigerated water samples for determine DOM absorbance and fluorescence were analyzed within one week, which is a suggested period for DOM optical measurement (Coble et al., 2014). We have included the information in the manuscript as: "*Refrigerated water samples for DOM absorbance and fluorescence were analyzed within one week after sampling*." (P6 Line 166-167).

Reference: Coble P. G., Lead, J., Baker, A., Reynolds, D. M. and Spencer, R. G.: Aquatic organic matter fluorescence, Cambridge University Press, 2014.

Ln 117-123: Did the authors measure blanks or standards for radiocarbon measurement to test and correct for contamination in the radiocarbon processing setup? Please describe whether these checks and corrections were performed, and if so, what the amount of error from contamination was, as contamination can be quite significant when processing samples for radiocarbon.

**Our response:** Thanks. We did have measured blanks and standards for radiocarbon measurement. The  ${}^{14}C/{}^{12}C$  background ratio was better than  $2 \times 10^{-15}$ . Detailed information for the radiocarbon processing is available at Dong et al. (2018).

Reference: Dong K., Lang, Y., Hu, N., Zhong, J., Xu, S., Hauser, T.-M. and Gan, R.: The new AMS facility at Tianjin University, Radiation Detection Technology and Methods, 2, doi:10.1007/s41605-018-0064-0, 2018.

Ln 124: Were blanks measured between absorption analyses and were the blanks subtracted from the measured samples? How did the authors calculate the absorption coefficient (which is needed to calculate SUVA), or are the values presented here raw absorbance values? The SUVA values presented in Figure 4a are much too high for the typical range of surface waters, so maybe SUVA was miscalculated by using the raw absorbance value rather than the decadic absorption coefficient as in Weishaar et al. 2003 (doi: 10.1021/es030360x)?

**Our response:** Thanks. Blanks have been measured and subtracted (please see the revised manuscript: "*The UV-visible spectrophotometer was blanked with Milli-Q water prior to data collection*"; P6 Line 158-159). Our previous SUVA

values were calculated from naperian values and thus why they are so high. We have re-calculated them from decadic absorbance. The related information has been added in the manuscript as follows: "Decadic absorbance values were used to calculate absorption coefficients as below (Poulin et al., 2014):  $a_{254} = Abs_{254}/L$ . Where,  $a_{254}$  is the absorption coefficients (cm-1),  $Abs_{254}$  is the absorbance at 254 nm, and L is the path length (cm). Specific UV absorbance at 254 nm (SUVA254) was determined according to Weishaar et al. (2003; Table 1): SUVA254 =  $a_{254}/DOC$ ." (P6 Line 159-164).

Ln 126: Absorbance and therefore SUVA are highly affected by the presence of iron in filtrate (see Poulin et al. 2014, doi: 1021/es502670r). Do you happen to know the iron concentrations or the iron:DOC ratio in this system? Iron interference could explain why the SUVA values are much higher than typical surface water values (typically  $< 5.0 \text{ L mg}^{-1} \text{ m}^{-1}$ , see Poulin et al. (2014) and references therein), though the SUVA values presented here seem too high for just an iron interference issue.

**Our response:** Thanks. We have recalculated the SUVA. The reason why SUVA values are much higher than typical surface water values was because previous SUVA values were calculated from naperian values. The new SUVA values are not much high now. In addition, there are no available references to show that the iron concentrations were high.

Ln 130: How often were blanks measured? Was inner-filter correction performed (see Kothawala et al. 2013, doi: 10.4319/lom.2013.11.616)? Inner-filter correction is essential when working with fluorescence data. Additionally, PARAFAC results should be compared previous work (for example, using OpenFluor) for context.

**Our response:** Thanks. We measured three blanks (before, during and after the sample measurement). We have added the information in the manuscript as: "*Blanks were measured daily with the same settings to correct excitation-emission matrices (EEMs)*." (P7 Line 171). Inner-filter correction was not performed, this is because the inner filter effects could be ignored as the absorbance at 254 nm was lower than  $0.1 \text{ m}^{-1}$ (Kothawala et al., 2013). Additionally, all the absorbance at 254 nm of the water samples in this study was lower than  $0.1 \text{ m}^{-1}$ .

Reference: Kothawala D. N., Murphy, K. R., Stedmon, C. A., Weyhenmeyer, G. A. and Tranvik, L. J.: Inner filter correction of dissolved organic matter fluorescence, Limnol. Oceanogr. Methods, 11, 616-630, doi:10.4319/lom.2013.11.616, 2013.

Discussion: Many results are presented in the discussion that are not first presented in the results. I suggest the authors move Figures 3-7 to the Results and add text describing these findings. Further, I think the number of figures can be greatly reduced and the discussion streamlined.

**Our response:** Thanks for your suggestions. We have moved these figures to the "Results" and added related description in the "Results". Also, the discussion has been modified accordingly. Please see details in the manuscript throughout the "Results" and "Discussion".

Figure 3: Unclear what the purpose of this figure or PARAFAC analysis is as these results are never discussed. This figure can be removed or discussion of PARAFAC results (after contextualizing the components based on OpenFluor, for example) should be added.

**Our response:** Thanks. We have added discussion of PARAFAC results after contextualizing the components based on OpenFluor in the manuscript: "*Two humic-like fluorescence components (C1 and C3) and one protein-like fluorescence component (C2) were identified by PARAFAC model in these three rivers (Fig. 7; Table S1). Component C1 is similar to traditionally defined peak M and sourced form microbial processes or autochthonous production (Kim et al., 2022; Li et al., 2016; Walker et al., 2009). Component C2 was previously related to recent biological production (DeFrancesco and Guéguen, 2021; Du et al., 2019; Lambert et al., 2017). C3 was the most widely found component in previous research among three fluorescence components, and was identified as traditional fulvic-like peaks A and C, representing terrestrial delivered OM or authochthonous microbial sourced OM (Amaral et al., 2016; Ryan et al., 2022; Shutova et al., 2014). Although C1 and C2 varied more widely in the Yinjiang River compared with the Shiqian and Yuqing rivers, the two fluorescence components did not show a statistical difference among the three rivers (Figs. 7a and b). However, a greater proportion of C3 was found in the Shiqian and Yuqing rivers, exhibiting a distinctive signature compred with the Yinjiang River (Fig. 7c). The fluorescence components did not exhibit any significant variations with changing channel slope (Fig. S3), but C1 and C2 were positively or negatively related to proportion of urban and agricultural land uses (Figs. 6b and c)." (P13 Line 277-288).*

| Component | Description and likely structure                                                                                                                                                           | Number of matches in Openfluor | Previous studies                                                                             |
|-----------|--------------------------------------------------------------------------------------------------------------------------------------------------------------------------------------------|--------------------------------|----------------------------------------------------------------------------------------------|
| C1        | Similar to traditionally defined peak M,
marine humic-like component, the products
from microbial processes or autochthonous
production.                                          | 6                              | C6 (Walker et al., 2009);
C4 (Kim et al., 2022);
C4 (Li et al., 2016)                  |
| C2        | Protein-like (Tryptophan-like) components,
commonly found in anthropogenically
affected rivers, associated with recent
biological production and breakdown
products of lignin. | 30                             | C3 (DeFrancesco and
Guéguen, 2021);
C7 (Lambert et al., 2017);
C2 (Du et al., 2019) |
| C3        | Traditional fulvic-like peaks A and C,
humic-like and terrestrial delivered OM,
authochthonous, or microbial source                                                                  | 70                             | C1 (Amaral et al., 2016);
C1 (Ryan et al., 2022);
C1 (Shutova et al., 2014)            |

**Table S1** Description of the three components identified by PARAFAC and comparison with previous studies from the OpenFluor database with a minimum similarity score of 0.95 (Murphy et al., 2014).

Figure 4a: SUVA values are uncharacteristically high (see comment above in the methods). Additionally, this figure (once the high SUVA values are addressed or corrected) would be more effective if is discussed in the section below when discussing geomorphology controls on DOM. Why not include it in Figure 6 when trends with slope are shown? **Our response:** Thanks. We have added it into the figure as shown below:

**Figure 3.** Mean channel slope (°) controls on (**a**) DOC concentrations, (**b**) stable carbon isotopes of DOC ( $\delta^{13}C_{DOC}$ ), (**c**) radiocarbon isotope of DOC ( $\Delta^{14}C_{DOC}$ ), (**d**) radiocarbon isotope of POC ( $\Delta^{14}C_{POC}$ ), and (**e**) SUVA254. The  $\Delta^{14}C_{DOC}$  and  $\Delta^{14}C_{POC}$  are only available for the Yinjiang River. Outliers in orange were excluded from analyses as they were samples at site Y12 collected after a rainfall event in panels (**c**) and (**d**) and sample collected at site S3 due to the high influence by road construction, which was evidenced by high POC and TSM concentration (Chen et al., 2021). The statistical test used a significance level of 0.05.

Figure 4b: This figure is confusing as we do not know much about the site numbers. I suggest the authors present this with changing slope or land-use instead. Additionally, I think this figure would be more effective if the three indices were split into three different plots. The small changes in each parameter are minimized by the large scale of the plot, and the indices shouldn't be compared directly, so there is no need to include them all on the same axis.

**Our response:** Thanks. We have added these indexes into the Figure 3, 6, S3 and S4 (please see above or below) or modified the figures as you suggested.

---

## Referee Report (RR1)

This study assessed how geographic controls (elevation, temperature, and slope) and % anthropogenic land cover (urban/agriculture) influence DOC export and DOM composition from mountainous rivers. The data presented shows that increased %urban/agriculture cover in lower reaches (and shallower gradients) of these catchments results in higher DOC concentrations, where carbon isotopic signatures (13 and 14C-DOC) of DOC are more deplete, and DOM is less aromatic. I believe these findings are of interest to a broad community. I believe the manuscript would be suitable for publication once the comments below are addressed.

General Comments:
1. The final paragraph of the introduction (aim/objectives) should be revised, as it is a little unclear as it stands. Specifically, I suggest combining the two hypotheses sentences into one to make the hypothesis clearer (i.e., by dealing with DOC concentration and quality at the same time). Lines 93-5 is a list of metrics/relationships examined but the sentence is long and could do with being broken up. I would also mention analysis of 14C-DOC samples in the Yinjiang river as a separate point.
2. Section 3.1 – there is a tendency to only reference 2 out of 3 of the rivers in this section. I understand why this is the case in some places as you are looking at where there are the largest differences, but this could be made clearer.
3. Results section: many of the figures are discussed out of order in text, and there is a lot of jumping between boxplots figures and then linear regressions. I suggest you restructure the results to that there is an initial description of metrics across sites (i.e., geochemical, DOC, isotopes, optical), followed by an additional section on the regressions with slope and land use. I think this will make the results easier to follow. Because of this, it would be good to include a section briefly describing optical parameters before diving into covariance with slope/land use. Consider including Figure S4 as a main text figure to aid initial description of optical metrics. This figure could be combined with Figure 7 and moved earlier in text. I would also suggest you discuss optical indices (e.g., SUVA, FI, HIX) before describing PARAFAC results.
4. $SUVA_{254}$ values have been corrected (i.e., in rebuttal table, and figure 3e), but not consistently (i.e., Figure 6a). Please update the figure and confirm that there is still a correlation with urban/agricultural land use.
5. 434 – 447 seems out of place. I think it would be better to contextualizes values more briefly and earlier in the discussion or within the results section. The importance of table 3 is also unclear to me.

Line Specific:
37: this sentence seems to be missing an 'and' or 'as well as'
53-55:  this is a little unclear consider rewording.
58: consider revising to "…, the underlying mechanisms that regulate DOC dynamics in small mountainous rivers remain poorly understood'.
63: SOC has not been defined. Please check that abbreviations are always defined in text (same applies to OC on line 144, and TP)
64: 'effectiveness' seems like the wrong word choice here.
71-70: suggest making the difference in DOM quality between the two stream types clearer in this sentence.
100: consider abbreviating or including abbreviation throughout text.
104: 'greatest' rather than 'great'.

111: revise sentence. Perhaps… "The proportion of urban and agricultural land uses in the Yuqing River catchment is from 17.3% to 23.1% (Figs. 1c and S1c). This catchment has typically higher % urban/ agriculture land use than other studied catchments, and less variability in land use compared to Yinjiang and Shiqian river catchments (4.5% to 46.5% and from 9.6% to 41.3%, respectively)."

113: remove "typically".

117: 'respectively' needs moving earlier in the sentence, just after values.

125: where were springs samples taken?

126-128: consider revising this sentence – it is a little unclear.

133: were bottles acid-washed before use?

135: what filter size was used for DIC? This should be specified.

136: remove 'moreover'.

146: which analysis does the deviation relate to?

155: remove 'same methods as 14C-POC'. The POC data was previously published? This should be made clearer in aims/objectives (i.e., after line 88), and perhaps reference should be made in figure captions.

157: I suggest stating that optical analyses were only conducted on River samples

199: can you contextualize which values represent supersaturation?

199: Q river also had higher water temperature than Y and springs?

224: add 'although not significantly different.'

225: sentence seems incomplete – perhaps add '…besides groundwater to the rivers'

226: remove 'in the supplement.'

285-286: it is unclear how a greater proportion of C3 in S and Q make them 'distinctive' to Y, especially when this is only significant for Y-S.

301: it would be useful to specify the direction of trends here.

298: your data does not support that this OC was 'fresher', as you did not find a significant correlation between freshness index and slope (lines 300)

316: there is more not 'less' aromaticity in steeper catchments.

317: there seems to be words missing from this sentence. Aromatic content tends to decline with what?

325: change to '….in regulating DOM composition.'

331: add '…increase in DOC concentration with microbial degradation in spring water…'

334: I think it would be useful to relate this back to the trends seen in your data.

352: replace 'it is worth noting that' with 'furthermore.'

365: I agree, but I can't see this trend in figure S3. There doesn't seem to be a correlation between elevation and land use or Cl- plotted.

379: I am unclear as to how you concluded that lower del13C-DOC with nitrate indicates greater algae contributions when no isotopic values for algae are presented. Some literature shows that algae has very enriched del13C values. Please expand and use isotopic data from the literature for endmembers to evaluate this discussion point.

399:'we further discussed' is confusing here.

408: it would be useful to include values from your study here for context.

421: consider rephrasing

423: if there are significant damming/reservoirs on these rivers I think it would be useful to reference this in the site description

431: this sentence could be made clearer. I think you are point to the fact that it is difficult to disentangle the two influences and they likely affect DOC concentration and DOM quality in tandem in these rivers.

454: 'deeper' is not needed

463-464: this seems like repetition.

Figure 2: please include Cl- concentrations as a panel.

Figure 2 / 7: 'average' should be replaced with 'mean' in the caption.
Figure 6: if PARAFAC components identified in the 3 rivers then why is there only one rivers data shown in panels b and C and/or why has the coloring changed. why is C3 not plotted in figure 6?
Figure 7: why are axes on EEMS not labelled. Please label axes and remove from caption.
Figures: boxes around legends are not always present – see comment from first review.

---

## Author Response (AR2)

**Response to Editor and Reviewers**

Dear Professor Shen,

We appreciate the opportunity to revise the manuscript and resubmit our paper after addressing all the comments of editors and reviewers on our previously submitted manuscript (No. bg-2022-217) once again. We also greatly appreciate the very helpful review from all reviewers. We have revised the manuscript taking into account all the comments we got.

Changes in the modified manuscript and revised supplementary document (in PDF) are marked by red and blue colors Please find blow our point-by-point responses (in bright blue) to the comments. All changes have also been highlighted in the revised version of the manuscript. The mentioned page number and line number below refer to the pages and lines in the clean version. We hope the revised version of the manuscript will be acceptable for publication in the journal Biogeosciences. We look forward to hearing from you.

Thank you very much for your kind consideration.

Sincerely,

Lishan Ran

On behalf of all the authors

**Response to Anonymous Referee #1**

Thank you for addressing many of technical comments and concerns from myself and the other reviewers. While the methods and statistical significance of the study results are clearer in this latest draft, the major contributions of this study remain difficult to follow because of the multitude of figure panels through the main text, the dense discussion, and the unnecessary inclusion of some previously published data. I suggest that the authors streamline the discussion text to focus on how the measurements made in the study can be used to build evidence in support of channel slope or land use as the major driver of DOC chemistry in the mountainous rivers, which is the main goal of this study. These types of revisions could help the authors identify which results to highlight in the discussion and which figure panels could be moved to the SI, which would improve the readability of the manuscript.

**Our response:** Thank you for your very helpful review of our manuscript once again and gave us very useful comments. We have already depleted some of unnecessary figure panels and previously published data in the modified manuscript, such as Figs. 1 and 4. We also streamlined the discussion text, such as the discussion on groundwater contribution on riverine DOC. Details are provided in the specific response below.

One aspect of the paper that remains feeble is the back and forth between channel slope and land use as potential major drivers of DOC cycling in the mountainous rivers in this study. Both of these factors could be driving the changes in DOC chemistry that were measured, but a discussion of which one might be more important is lacking. Likewise, if the authors cannot rule out one or the other, it is not admissible to say that both factors are driving DOC cycling because it is still possible that one factor is more important than the other. I suggest that the authors start the discussion by walking through the lines of evidence in support of one or the other. Without this type of critical analysis, it is difficult to believe that channel slope alone drives DOC concentration, which is the focus at the start of the discussion, and it is confusing to then read later in the discussion that land use is also a major driver. If there isn't strong enough evidence to support one explanation over the other, then the authors should discuss why they cannot be teased apart rather than saying that both channel slope and land use drive DOC chemistry.

For example, the authors report that agricultural activities become more prominent going downstream, while at the same time channel slope generally decreases. This type of result makes me wonder whether there is a significant negative correlation between the % of urban and agriculture land use and the channel slope. Including a figure showing the % of urban and agriculture land use plotted versus the channel slope (even adding it as a panel to Fig. S1) would help readers to understand how correlation may not mean causation for this dataset, but that additional chemistry measurements (like the ones the author made) are needed to tease apart the major controls.

**Our response:** Previous studies have shown that catchment slope (Harms et al., 2016; Lee et al., 2019) can be a single significant factor controlling the DOC/DOM dynamics. In these studies, the slope is linked to the hydrologic response of the catchment, rather than the human land use pattern. Although several studies also demonstrate that slope is closely associated with wetlands (Connolly et al., 2018; Inamdar and Mitchell, 2006; Winn et al., 2009), where DOM processing is complicated (Zhou et al., 2022), they also highlight the important role of controlling catchment hydrological connectivity. Additionally, we emphasize the anthropogenic influence (i.e., agriculture and urban land use) rather than the impact of wetlands in this study. In conclusion, the slope is an important factor related to hydrologic connectivity, while land use is more important for controlling the source of DOM. On the other hand, $R^2$ in Fig. 3a and Fig. 4a are 0.5 and 0.47, respectively. This means they are almost equally important to DOC export in the studied rivers. Thus, there isn't strong enough evidence to support one explanation over the other.

We have added a figure to show the % of urban and agricultural land use plotted versus the slope in Fig. S1 (please see figure below). Although there is a negative correlation between urban and agricultural land use and mean catchment slope, it does not mean causation here. In this study, carbon isotopes of DOC and POC were shown to be regulated by slope and showed no significant correlation with urban and agricultural land use (Fig. S4). However, fluorescence indexes and components are closely related to urban and agricultural land use rather than the mean catchment slope. These correlations indicated that urban and agricultural land use and mean catchment slope are two separate factors influencing the DOC dynamic and DOM composition.

[Figure]

**Figure S1** Distribution of (a) mean drainage elevation, (b) mean channel slope, (c) urban and agricultural land uses in the Yinjiang (Y), Shiqian (S), and Yuqing (Q) catchments, and (d) percentage of urban and agricultural land uses versus the mean catchment slope.

[Figure]

**Figure S4** Correlation plot of the selected water chemistry and catchment characteristics. The colors represent the degree of pairwise correlation regarding Pearson's correlation coefficient. $\delta^{13}C_{DOC}$ and $\Delta^{14}C_{DOC}$ at site Y12 was excluded from analyses as the sample was collected after a rainfall event. In addition, $SUVA_{254}$ at site S3 was excluded from analyses as the sample was strongly influenced by the road construction, which was evidenced by high POC and TSM concentration (Chen et al., 2021). Human land use used here represents proportion of urban and agricultural land use. Elevation and Annual $T_{air}$ represent mean drainage elevation and annual air temperature, respectively.

References:

Connolly C. T., Khosh, M. S., Burkart, G. A., Douglas, T. A., Holmes, R. M., Jacobson, A. D., Tank, S. E. and McClelland, J. W.: Watershed slope as a predictor of fluvial dissolved organic matter and nitrate concentrations across geographical space and catchment size in the Arctic, Environ. Res. Lett., 13, 104015, doi:10.1088/1748-9326/aae35d, 2018.

Harms T. K., Edmonds, J. W., Genet, H., Creed, I. F., Aldred, D., Balser, A. and Jones, J. B.: Catchment influence on nitrate and dissolved organic matter in Alaskan streams across a latitudinal gradient, J. Geophys. Res.: Biogeo., 121, 350-369, doi:10.1002/2015jg003201, 2016.

Inamdar S. P. and Mitchell, M. J.: Hydrologic and topographic controls on storm-event exports of dissolved organic carbon (DOC) and nitrate across catchment scales, Water Resour. Res., 42, doi:10.1029/2005wr004212, 2006.

Lee L.-C., Hsu, T.-C., Lee, T.-Y., Shih, Y.-T., Lin, C.-Y., Jien, S.-H., Hein, T., Zehetner, F., Shiah, F.-K. and Huang, J.-C.: Unusual roles of discharge, slope and soc in doc transport in small mountainous rivers, Taiwan, Sci. Rep., 9, 41422303, doi:10.1038/s41598-018-38276-x, 2019.

Winn N., Williamson, C. E., Abbitt, R., Rose, K., Renwick, W., Henry, M. and Saros, J.: Modeling dissolved organic carbon in subalpine and alpine lakes with GIS and remote sensing, Landscape Ecol., 24, 807-816, doi:10.1007/s10980-009-9359-3, 2009.

Zhou X., Johnston, S. E. and Bogard, M. J.: Organic matter cycling in a model restored wetland receiving complex effluent, Biogeochemistry, doi:10.1007/s10533-022-01002-x, 2022.

The authors include many measurements of water and DOC chemistry in the manuscript, but not all are needed for the authors to arrive at the main conclusions of the study. Including all the results obtained in the study in the main text makes it difficult to follow which figures best show the main results of the study and which are distracting to the reader. For example, past work by the authors in Chen et al. (2021) showcase only the figures needed to arrive at the main conclusions of the paper and it is much easier to follow. As the discussion reads right now, it is difficult to identify which chemistry measurements are being used to support the main findings of the paper and follow why the water chemistry previously published in Chen et al. (2021) are needed to support the main findings.

**Our response:** According to your suggestion, we have removed some unnecessary figures related to water chemistry, such as pH, WT, TN, and TP (Fig. 2, please see below). On the other hand, we need to provide relevant information on the distribution of nutrients, DOC concentration, and their isotopes in spring water and rivers to give readers a basic understanding of the distribution of these parameters, which would provide a background for further in-depth discussions. In addition, the remaining water chemistry were discussed in the following discussion sections.

[Figure]

**Figure 2.** Spatial variations in water chemistry in the Yinjiang (Y), Shiqian (S), Yuqing (Q) rivers, and springs. (**a**) DO, (**b**) Cl⁻, (**c**) $NH_4^+$-N, (**d**) $NO_3^-$-N, (**e**) DOC, and (**f**) $\delta^{13}C_{DOC}$. In each box plot, the end of the box represents the 25th and 75th percentiles, the blue solid dot represents the mean, the horizontal line inside the box represents the median, and whiskers represent 1.5 times the upper and lower interquartile ranges (IQR). The magenta solid dot represents the outlier (data points outside of the 1.5 interquartile ranges). Letters above the boxes represent significant differences between the grouping of river and/or spring water based on statistical analyses at the significance level of 0.05 (e.g., Y-Spring above panel (b) indicates that the Cl⁻ in river water of the Yinjiang River was significantly different from that in the spring water).

Other comments:

1. Given that DOC concentrations decrease with distance downstream, similar to what the authors previously found for DIC and POC in Chen et al. (2021), wouldn't it be more likely that DOC concentrations are also driven by groundwater or erosion rather than agricultural activities?

**Our response:** We agree with you that DOC concentrations increase with distance downstream is likely driven by groundwater or erosion. We attributed the DIC and POC concentrations increase with distance downstream to the carbonate weathering and in-stream photosynthesis in our previous study (Chen et al., 2021). Groundwater contribution to riverine DOC concentrations has been discussed in the manuscript (P16 Line 373-385). However, "*groundwater was likely not the primary source of riverine DOC*

*due to the relatively low groundwater DOC concentrations as compared with riverine DOC concentrations." (P16 Line 381).* Groundwater DOC is lower than that in the river. Therefore, it can be the reason for the increase of DOC towards downstream. We have already discussed the erosion controls on DOC concentrations, and it is related to urban and agricultural land use: *"Conversion of native forest and pasture to row crop agriculture may lead to substantial losses of SOC stores as a result of greatly accelerated erosion rates and decomposition rates (Guo and Gifford, 2002; Montgomery, 2007; Stanley et al., 2012). In comparison, natural vegetation could greatly reduce SOC input into rivers by effectively reducing soil erosion through the consolidation effect of roots on soil and the interception of rainfall by stems and leaves (Zhang et al., 2019). Anthropogenic activities are closely related to mean drainage elevation (Fig. S4). Agricultural activities mainly occur in low-elevation areas (Fig. S4), which tend to liberate SOC through erosion over longer timescales and cause an elevated DOC export into rivers (Fig. 5a), although DOC of urban origin can also make a huge contribution to the riverine DOC pool (Sickman et al., 2007)." (P17 Line 394-401).*

2. It seems that comments about incorrect SUVA254 values have not been addressed in the current paper. The range of SUVA254 values on the y-axis of Fig. 6a should be the same as those reported in the y-axis of Fig. 3e, but they are not. The range of values showed in Fig. 3e is consistent with what has been previously reported in the literature, which makes me think that there was an error in the calculation for SUVA254 in Fig. 6a. The authors should check their spreadsheets to make sure that SUVA254 was calculated correctly in the files used to generate these figures. The authors could also report in the methods which final units for SUVA254 were obtained. They should be in units of L mg-C-1 m-1 using the decadic absorption coefficient at 254 nm.
**Our response:** Thanks for. We have redrawn Fig. 6a with the correct SUVA254 data (Fig. 7a). We also added the units for SUVA254: *"Specific UV absorbance at 254 nm (SUVA$_{254}$; reported in units of L mg C$^{-1}$ m$^{-1}$) was determined according to Weishaar et al."* (P7 Line 191).

[Figure]

**Figure 7.** Land use pattern impacts on DOM character. (**a**) SUVA$_{254}$, (**b**) freshness index (β/α), and (**f**) C2 decreased with the increasing proportion of urban and agricultural land uses. Outlier (site S3) was excluded from analysis in panel (a) as the sample was strongly influenced by road construction, which was evidenced by high POC and TSM concentration (Chen et al., 2021). (**c**) humification index (HIX) and (**e**) C1 were positively related to the increasing proportion of urban and agricultural land uses. However, there was no significant correlation between (**d**) fluorescence index and the proportion of urban and agricultural land uses.

3. Line 101: How was annual air temperature obtained from ArcGIS? It seems to me that this should be reported separately from

the physical catchment characteristics.

**Our response:** The information on annual air temperature was provided in our previous study: The data of land use types and air temperature in 2015 were retrieved from the Resource and Environment Data Cloud Platform of the Chinese Academy of Sciences (http://www.resdc.cn/ ). We also added the related information in the revised manuscript as follows: "*Data on land use types and air temperature in 2015, as well as a 90 m digital elevation model (Shuttle Radar Topography Mission, SRTM) were obtained from the Resource and Environment Data Cloud Platform of the Chinese Academy of Sciences (http://www.resdc.cn/).*". (P4 Line 119-121).

4. Adding details about how the catchment slopes were estimated has improved the manuscript, but it would still be helpful to include a visual in the main text to show how the slopes vary within and between the rivers studied. Could you make another panel in Fig. 1 that shows channel slope on the y-axis versus distance in meters downstream? It is difficult to use the gradients shown in Fig. 1b and dots in Fig. 1a to piece this together myself, so having a figure that clearly shows the gradients would be helpful, especially when readers interpret the results shown in Fig. 3.

**Our response:** Thank you. We have added another panel in Fig. 1 (please see below) that shows channel slope on the y-axis versus distance in meters downstream.

[Figure]

**Figure 1.** Map of the study area. (**a**) Overview of the sampling sites and elevation characteristics in the three study catchments, including the Yinjiang, Shiqian, and Yuqing catchments, (**b**) correlation between mean catchment slope and the distance from the river mouth (i.e., the Yinjiang, Shiqian, and Yuqing rivers) to the sampling site, and (**c**) spatial variation in land-use patterns.

5. If discussing differences in water chemistry between the spring water and rivers does not help support or refute the roles of land use or channel slope as drivers, then does this need to be included in the main text?

**Our response:** According to your suggestion, we have removed some unnecessary figures related to water chemistry, such as pH, WT, TN, and TP (Fig. 2). On the other hand, we need to provide relevant information on the distribution of nutrients, DOC concentration, and their isotopes in spring water and rivers to give readers a basic understanding of the distribution of these

parameters, which would provide a background for further in-depth discussions.

[Figure]

**Figure 2.** Spatial variations in water chemistry in the Yinjiang (Y), Shiqian (S), Yuqing (Q) rivers, and springs. (**a**) DO, (**b**) Cl⁻, (**c**) $NH_4^+$-N, (**d**) $NO_3^-$-N, (**e**) DOC, and (**f**) $\delta^{13}C_{DOC}$. In each box plot, the end of the box represents the 25th and 75th percentiles, the blue solid dot represents the mean, the horizontal line inside the box represents the median, and whiskers represent 1.5 times the upper and lower interquartile ranges (IQR). The magenta solid dot represents the outlier (data points outside of the 1.5 interquartile ranges). Letters above the boxes represent significant differences between the grouping of river and/or spring water based on statistical analyses at the significance level of 0.05 (e.g., Y-Spring above panel (b) indicates that the Cl⁻ in river water of the Yinjiang River was significantly different from that in the spring water).

8. Some of the information reported in Fig. 4 could be inferred or grasped easily from what is shown in Fig. 3 – do you need all of these figures in the main text? Which are needed to arrive at the main points in the paper?

**Our response:** Thank you, some of the panels in Figs. 3 and 4 have been moved to supplement. The new figures are shown below as Figs. 4 and 5 in the revised manuscript. We kept some panels. Although it is easy to infer the possible relationships between DOC and isotopes of DOC, however, we can not get information on the significance between these variables. Thus, we moved some panels to the supplement as they are not so important (e.g., slope vs $\Delta^{14}C_{POC}$, $NH_4^+$-N vs DOC, and $NO_3^-$-N vs $\delta^{13}C_{DOC}$).

[Figure]

**Figure 4.** Mean catchment slope (°) controls on (**a**) DOC concentrations, (**b**) stable carbon isotopes of DOC ($\delta^{13}C_{DOC}$), (**c**) radiocarbon isotope of DOC ($\Delta^{14}C_{DOC}$), and (**d**) SUVA$_{254}$. The $\Delta^{14}C_{DOC}$ is only available for the Yinjiang River. Outliers in orange were excluded from analyses as they were samples at site Y12 (Fig. 1a) collected after a rainfall event in panel (**c**) and the samples collected at site S3 (Fig. 1a) in panel (**d**) due to the high influence by road construction, which was evidenced by high POC and TSM concentration (Chen et al., 2021). The statistical test used a significance level of 0.05.

[Figure]

**Figure 5.** Land use pattern and anthropogenic impacts on DOC concentrations, indicated by relationships between DOC and (**a**) proportion of urban and agricultural land uses, and (**b**) anthropogenic Cl⁻ concentration (i.e., Cl⁻$_{anthro}$, calculated as the total Cl⁻ concentration minus atmospheric contributed Cl⁻ concentration, which is the lowest Cl⁻ concentration at site Y5 in the Yinjiang River). The statistical test used a significance level of 0.05.

9. Figure 7 does not seem to contribute to the main points of the paper, so this could be moved to the SI.

**Our response:** We have moved it to the SI.

**Response to Anonymous Referee #2**

This study assessed how geographic controls (elevation, temperature, and slope) and % anthropogenic land cover (urban/agriculture) influence DOC export and DOM composition from mountainous rivers. The data presented shows that increased %urban/agriculture cover in lower reaches (and shallower gradients) of these catchments results in higher DOC concentrations, where carbon isotopic signatures (13 and 14C-DOC) of DOC are more depleted, and DOM is less aromatic. I believe these findings are of interest to a broad community. I believe the manuscript would be suitable for publication once the comments below are addressed.

**Our response:** Thank you very much for providing so many valuable opinions and suggestions. The details of our modifications are provided in the response below.

**General Comments:**

1. The final paragraph of the introduction (aim/objectives) should be revised, as it is a little unclear as it stands. Specifically, I suggest combining the two hypotheses sentences into one to make the hypothesis clearer (i.e., by dealing with DOC concentration and quality at the same time). Lines 93-5 is a list of metrics/relationships examined, but the sentence is long and could do with being broken up. I would also mention analysis of 14C-DOC samples in the Yinjiang river as a separate point.

**Our response:** Thanks. We have revised these sentences as: "*We hypothesize that catchments with a higher proportion of agricultural and urban land uses, more gentle catchment slope, and lower elevation would exhibit higher riverine DOC concentrations and more autochthonous microbial humic-like DOM than steeper catchments at high elevations with less influences by agricultural and urban land uses. Relationships of DOC concentrations, stable isotopic values of DOC, DOM quality assessed through optical metric, nutrient concentrations, and land use patterns versus geographical characteristics (i.e., mean catchment slope, mean drainage elevation, and annual air temperature) were examined. We also examined relationships between geographical characteristics and radiocarbon for nine sampling sites in the Yinjiang River.*" (P4 Line 106-112).

2. Section 3.1 – there is a tendency to only reference 2 out of 3 of the rivers in this section. I understand why this is the case in some places as you are looking at where there are the largest differences, but this could be made clearer.

**Our response:** We have modified the results section to make it clearer by adding relative information for all three rivers. Please see "3 Results" for details (P9-15).

3. Results section: many of the figures are discussed out of order in text, and there is a lot of jumping between boxplots figures and then linear regressions. I suggest you restructure the results to that there is an initial description of metrics across sites (i.e., geochemical, DOC, isotopes, optical), followed by an additional section on the regressions with slope and land use. I think this will make the results easier to follow. Because of this, it would be good to include a section briefly describing optical parameters before diving into covariance with slope/land use. Consider including Figure S4 as a main text figure to aid initial description of optical metrics. This figure could be combined with Figure 7 and moved earlier in text. I would also suggest you discuss optical indices (e.g., SUVA, FI, HIX) before describing PARAFAC results.

**Our response:** We have restructured the results following the order you suggested (P9-15). We first gave the basic information of metrics across sites following the order of geochemical, DOC, isotopes, and optical results. Then we added an additional section on the regressions between the water chemistry and slope and land use. In addition, we have moved Figure S4 to the main text (P12), and it was moved earlier with Figure 7. In the end, we also discussed optical indices earlier than the PARAFAC results (P11).

4. SUVA254 values have been corrected (i.e., in rebuttal table, and figure 3e), but not consistently (i.e., Figure 6a). Please update the figure and confirm that there is still a correlation with urban/agricultural land use.

**Our response:** We have updated Fig. 6a with the correct SUVA254 data (P15; please see below, as shown in Fig. 7a).

[Figure]

**Figure 7.** Land use pattern impacts on DOM character. (**a**) SUVA$_{254}$, (**b**) freshness index (β/α), and (**f**) C2 decreased with the increasing proportion of urban and agricultural land uses. Outlier (site S3) was excluded from analysis in panel (a) as the sample was strongly influenced by road construction, which was evidenced by high POC and TSM concentration (Chen et al., 2021). (**c**) humification index (HIX) and (**e**) C1 were positively related to the increasing proportion of urban and agricultural land uses. However, there was no significant correlation between (**d**) fluorescence index and the proportion of urban and agricultural land uses.

5. 434 – 447 seems out of place. I think it would be better to contextualizes values more briefly and earlier in the discussion or within the results section. The importance of table 3 is also unclear to me.

**Our response:** The reason we included this part and Table 3 at the end of the Discussion is because of a suggestion from one of the reviewers last time. This section can help us to understand the distribution of carbon isotopes, especially 14C, in other mountainous rivers and the reasons for such distribution. However, since this is not the core issue of our study, we only briefly described it. But considering describing the age of DOC from so many mountainous rivers and explaining its reasons, it is not appropriate to make it too short. Therefore, we chose to keep this paragraph. In addition, we also explained the purpose of adding this paragraph in the title of section 4.3 (Combined effects of geomorphologic and anthropogenic controls on DOC and comparison of $\Delta^{14}C_{DOC}$ in mountainous rivers). (P20-21).

**Line Specific**:

37: this sentence seems to be missing an 'and' or 'as well as'

**Our response:** We have modified this sentence as "*Riverine DOC can also restrict in-stream primary production by reducing light penetration and lowering temperature in the water column, thereby serving as an important determinant in shaping the ecological and biogeochemical processes in aquatic environments*" (P2 Line 37-39).

53-55: this is a little unclear consider rewording.

**Our response:** The sentence was reworded as: "*More specifically, DOC supply is likely regulated by the amount of stored soil organic carbon (SOC) in a catchment (Lee et al., 2019; Rawlins et al., 2021). However, this supply is limited by shallow soil depth and high water flow velocity (Lee et al., 2019).*" (P2-3 Line 62-64).

58: consider revising to "…, the underlying mechanisms that regulate DOC dynamics in small mountainous rivers remain poorly understood'.

**Our response:** We have revised it as you advised. (P3 Line 70).

63: SOC has not been defined. Please check that abbreviations are always defined in text (same applies to OC on line 144, and TP)

**Our response:** SOC has been defined as "*More specifically, DOC supply is likely regulated by the amount of stored soil organic carbon (SOC) in a catchment* (P2 Line 62-63)". We have revised the OC on line 144 as organic carbon: "*DOC concentrations were determined with a total organic carbon analyser*" (P7 Line 172) and deleted the relative description on TP as another reviewer's suggestion.

64: 'Effectiveness' seems like the wrong word choice here.

**Our response:** We have reworded this sentence as: "*Yet, the extent to which these factors, along with land use patterns, effectively regulate the DOC dynamic is still far from well-understood*" (P3 Line 76-77).

71-70: suggest making the difference in DOM quality between the two stream types clearer in this sentence.

**Our response:** We have revised this sentence as: "*Agricultural streams and rivers are dominated by microbial-derived, protein-like DOM, while urban freshwater ecosystems are characterized by microbial, humic-like or protein-like, and autochthonous DOM (Hosen et al., 2014; Williams et al., 2016; Xenopoulos et al., 2021). Agricultural and urban land uses tend to increase nutrient loading in streams, resulting in enhanced bacterial production and DOM decomposition (Quinton et al., 2010; Williams et al., 2010). Therefore, microbial-derived DOM plays a crucial role in agricultural and urban rivers. In addition, DOM tends to have a more reduced redox state and is likely more labile and accessible to the microbial community in agricultural streams when compared to the DOM found in natural streams (Williams et al., 2010). Although the DOM in urban rivers shares some similarities with agricultural rivers (such as microbial origins), the sources of DOM in urban rivers are much more complex, which may originate from urban point-source inputs (e.g., wastewater treatment facilities) and nonpoint source inputs (e.g., household sewage and petroleum-based hydrocarbons) (Hosen et al., 2014).*" (P3 Line 80-89).

100: consider abbreviating or including abbreviation throughout text.

**Our response:** Thanks. Since the names of these rivers are not very long, and abbreviating them with single letters (e.g., Y for the Yinjiang River) could cause ambiguity, we did not abbreviate them throughout the entire text. However, where abbreviations are used in the figures, we also provide the full name in the caption. Therefore, we did not add additional abbreviations in the full text because it seems redundant.

104: 'greatest' rather than 'great'.

**Our response:** Thanks. We have replaced 'great' with 'greatest'. (P5 Line 133).

111: revise sentence. Perhaps… "The proportion of urban and agricultural land uses in the Yuqing River catchment is from 17.3% to 23.1% (Figs. 1c and S1c). This catchment has typically higher % urban/agriculture land use than other studied catchments, and less variability in land use compared to Yinjiang and Shiqian river catchments (4.5% to 46.5% and from 9.6% to 41.3%, respectively)."

**Our response:** Thank you. We have modified this sentence as you suggested. (P5 Line 133).

113: remove "typically".

**Our response:** "typical," was removed.

117: 'respectively' needs moving earlier in the sentence, just after values.

**Our response:** We have moved it earlier: "*This study area is highly affected by monsoon-influenced humid subtropical climate*

*with April to October being the rainy season, and the average annual precipitation and discharge are 1100 mm and 14.4 m³/s, respectively, in the Yinjiang River catchment.*" (P6 Line 146-148).

125: where were springs samples taken?

**Our response:** Thank you for your reminder. We have added the locations in Fig. 1a (please see below).

[Figure]

**Figure 2.** Map of the study area. (**a**) Overview of the sampling sites and elevation characteristics in the three study catchments, including the Yinjiang, Shiqian, and Yuqing catchments, (**b**) correlation between mean catchment slope and the distance from the river mouth (i.e., the Yinjiang, Shiqian, and Yuqing rivers) to the sampling site, and (**c**) spatial variation in land-use patterns.

126-128: consider revising this sentence – it is a little unclear.

**Our response:** The sentence was revised as: "*Unless stated otherwise, the data used in this study from site Y12 are based on the sample collected after rainfall event due to the availability of carbon isotopes.*" (P6 Line 154-155).

133: were bottles acid-washed before use?

**Our response:** The bottles were not acid-washed before use. However, these bottles are brand new and have not been used before. We not only washed them with Milli-Q water in the laboratory before going out for the field trip but also rinsed them with the sampling water three times or more before collecting the water samples. Therefore, we believe that any residue in the sampling bottles will not affect the experimental analysis. We have added more details in this sentence: "*The filtered water was stored in a Milli-Q water and sampling water pre-washed brand new low-density polyethylene container at low temperature (4℃) in the dark before optical properties analysis and acidified by phosphoric acid to pH = 2 for DOC analysis.*" (P6 Line 160-162).

135: what filter size was used for DIC? This should be specified.

**Our response:** We have added information on filter size in the revised manuscript: "*Water samples were also filtered for*

*determining dissolved inorganic carbon (DIC; through 0.45 µm cellulose acetate membranes) through titration with hydrochloric acid and analyzing POC using retained suspended particles on the filter membranes.*" (P6 Line 162-164).

136: remove 'moreover'.

**Our response:** We have removed 'moreover' in this sentence.

146: which analysis does the deviation relate to?

**Our response:** To make it clear, we modify this paragraph as: "*The concentrations of $NH_4^+$-N were analyzed using an automatic flow analyzer (Skalar Sans Plus Systems), and the relative deviations of the results of $NH_4^+$-N and were less than 5%. DOC concentrations were determined with a total organic carbon analyser (OI Analytical, Aurora 1030W, USA) with duplicates (±1.5%, analytical error) and a detection limit at 0.01 mg $L^{-1}$. Water isotopes were measured by a Liquid Water Isotope Analyzer (Picarro L2140-i, USA) with measurement precisions at ± 0.3 ‰ for $\delta^{18}O$. The above analyses were carried out at the Institute of Surface Earth System Science, Tianjin University*." (P6-7 Line 170-175).

155: remove 'same methods as 14C-POC'. The POC data was previously published? This should be made clearer in aims/objectives (i.e., after line 88), and perhaps reference should be made in figure captions.

**Our response:** We have made it clearer in the revised manuscript: "*Our prior observations from these catchments showed that particulate organic carbon (POC) and dissolved inorganic carbon (DIC) dynamics were highly affected by in-stream photosynthesis, as evidenced by stable carbon isotope and radioactive carbon isotope of POC and DIC (Chen et al., 2021).*" (P4 Line 103-106). In addition, the reference has been made in figure captions (e.g., Fig. 6; please see below).

[Figure]

**Figure 6.** Scatter plot showing (**a**) $\delta^{13}C_{DOC}$ versus DOC in river water, (**b**) $\delta^{13}C_{DOC}$ against $\delta^{13}C_{POC}$ in the Yinjiang River (Y), Shiqian River (S), and Yuqing River (Q), and (**c**) relationship between $\Delta^{14}C_{POC}$ and $\Delta^{14}C_{DOC}$ in the Yinjiang River. For panel (**c**), the DOC with modern age at site Y12 was shown in the top-right corner. The statistical test used a significance level of 0.05. Details on $\delta^{13}C_{POC}$ and $\Delta^{14}C_{POC}$ are available in our earlier work (Chen et al., 2021).

157: I suggest stating that optical analyses were only conducted on River samples

**Our response:** We have added this information in the manuscript as the first sentence in this paragraph: "*Optical analyses on DOM were conducted on river samples.*" (P7 Line 185).

199: can you contextualize which values represent supersaturation?

**Our response:** We have revised this sentence as: "*The average DO presented similar values between the Yinjiang River, Shiqian River, Yuqing River, and springs with the majority of the river water samples being DO supersaturated (i.e., higher than 100%)*" (P9 Line 229-230).

199: Q river also had higher water temperature than Y and springs?

**Our response:** The related description has been depleted according to another reviewer's suggestion.

224: add 'although not significantly different.'

**Our response:** We have revised this sentence in the revised manuscript: "*The DOC concentrations in spring water were significantly lower than that in the surface water of the Shiqian and Yuqing rivers ($p < 0.05$; Fig. 2e), and the average DOC concentration in spring water ($0.74 \pm 0.30$ mg L-1) was also lower than the average DOC concentration in the Yinjiang River, indicating there must be other sources of DOC besides groundwater.*" (P10 Line 252-254).

225: sentence seems incomplete – perhaps add '…besides groundwater to the rivers.'

**Our response:** Thanks. We have added it as shown in the last question.

226: remove 'in the supplement.'

**Our response:** 'In the supplement' was removed in the revised manuscript.

285-286: it is unclear how a greater proportion of C3 in S and Q make them 'distinctive' to Y, especially when this is only significant for Y-S.

**Our response:** We have revised this sentence as: "*However, a greater proportion of C3 was found in the Shiqian River, exhibiting a distinctive signature compared with the Yinjiang River (Fig. S3c). The proportion of C3 did not show any significant difference between the Yuqing River and the other two rivers (the Yinjiang and Shiqian rivers).*" (P11 Line 272-275).

301: it would be useful to specify the direction of trends here.

**Our response:** we have added the direction of trends here: "*Although no significant correlation was observed between the fluorescence indexes and catchment slope, they (except for FI) were found to be closely related to land use pattern ($p < 0.05$; Figs. 7b, c, d, and S4). For example, HIX had a positive correlation with urban and agricultural land uses ($p < 0.001$; Fig.7e ), while $\beta/\alpha$ had a negative correlation with urban and agricultural land uses ($p < 0.01$; Fig.7d)*" (P13-14 Line 313-316).

308: your data does not support that this OC was 'fresher', as you did not find a significant correlation between freshness index and slope (lines 300)

**Our response:** Thanks. We have deleted 'fresher' in this sentence.

316: there is more not 'less' aromaticity in steeper catchments.

**Our response:** We have revised this sentence as: "*The correlation of SUVA$_{254}$ with mean catchment slope suggests that steeper catchments tend to export DOC with more aromaticity (Fig. 4d)*" (P15 Line 352-353).

317: there seems to be words missing from this sentence. Aromatic content tends to decline with what?

**Our response:** We have revised this sentence as: "*Previous research has reported that the aromatic content of DOM tends to decline when DOM is derived from deeper soil profiles*" (P16 Line 354-355).

325: change to '….in regulating DOM composition.'

**Our response:** We have changed it as you suggested. "*However, there were no similar correlation between catchment slope and fluorescence components/indexes in this study, demonstrating the likely complicated mechanisms (e.g., soil property, catchment characteristics, and anthropogenic activities) in regulating DOM composition.*" (P16 Line 360-362).

331: add '…increase in DOC concentration with microbial degradation in spring water…'

**Our response:** We have added it in the revised manuscript as: "*Previous studies have reported a decreasing $\delta^{13}C_{DOC}$ with a corresponding increase in DOC concentrations (Fig. 5a) with microbial degradation in spring water (Nkoue Ndondo et al., 2020) and for TOC in soil profiles*" (P16 Line 366-368).

334: I think it would be useful to relate this back to the trends seen in your data.

**Our response:** We have added relate this to Fig. 5 in the revised manuscript: "*As a result, the remaining DOC of lower concentrations is typically characterized by a heavier $\delta^{13}C_{DOC}$ (Fig. 5a; Nkoue Ndondo et al., 2020; Opsahl and Zepp, 2001)*." (P16 Line 370-372).

352: replace 'it is worth noting that' with 'furthermore.'

**Our response:** Thanks. We have replaced it: "*Furthermore, the decrease in DOC concentration with increasing mean catchment slope (Fig. 3a) may also be controlled by annual air temperature and land use pattern*" (P17 Line 386-387).

365: I agree, but I can't see this trend in figure S3. There doesn't seem to be a correlation between elevation and land use or Cl- plotted.

**Our response:** Thanks. The correlation between elevation and land use or Cl- can be found in figure S4.

[Figure]

**Figure S4** Correlation plot of the selected water chemistry and catchment characteristics. The colors represent the degree of pairwise correlation regarding Pearson's correlation coefficient. $\delta^{13}C_{DOC}$ and $\Delta^{14}C_{DOC}$ at site Y12 were excluded from analyses as the sample was collected after a rainfall event. In addition, $SUVA_{254}$ at site S3 was excluded from analyses as the sample was strongly influenced by the road construction, which was evidenced by high POC and TSM concentration (Chen et al., 2021). Human land use used here represents the proportion of urban and agricultural land use. Elevation and Annual $T_{air}$ represent mean drainage elevation and annual air temperature, respectively.

379: I am unclear as to how you concluded that lower del13C-DOC with nitrate indicates greater algae contributions when no isotopic values for algae are presented. Some literature shows that algae has very enriched del13C values. Please expand and use isotopic data from the literature for endmembers to evaluate this discussion point.

**Our response:** We didn't find direct $\delta^{13}C_{DOC}$ data for algae, but we found $^{13}C$ data for POC that is mainly contributed by phytoplankton in adjacent rivers. Since this POC is also an important source of DOC, we can use this as an endmember for the $^{13}C$

signature of DOC derived from phytoplankton. We have added this information in the revised manuscript: "*A recent study conducted in the Longtan Reservoir in the Xijiang River basin (China) with widespread karst landscape found that a majority of its POC was intercepted or degraded within the reservoir, with the POC primarily originating from phytoplankton (Yi et al., 2022). Its carbon isotope composition of POC ($\delta^{13}C_{POC}$) ranged from -35‰ to -30‰, which is relatively depleted, and the POC was found to be a significant contributor to the reservoir's DOC (Yi et al., 2022). Thus, the lower $\delta^{13}C_{DOC}$ with increasing $NO_3^-$-N further indicated the greater algae or C3 plant derived DOC accumulation with a higher level of nutrients (Fig. S5b).*" (P17-18 Line 411-416).

399:'we further discussed' is confusing here.

**Our response:** We have revised this sentence as: "*Here, we further examine how these two factors regulate riverine DOC.*" (P18 Line 435).

408: it would be useful to include values from your study here for context.

**Our response:** We have added the values in the revised manuscript: "*The DOC ages of the Yinjiang River (548 ± 195 yr BP; the modern sample at site Y12 was excluded) were younger than that of DOC reported in agricultural rivers (Moyer et al., 2013; Sickman et al., 2010) or treated wastewater (Griffith et al., 2009), which is typically more than 1000 years old.*" (P18-19 Line 443-445).

421: consider rephrasing

**Our response:** We have rephrased this sentence as: "*Campeau et al. (2020) attributed this relationship to common controls of landscape and/or hydrology on the sources of organic carbon in rivers. Yet, this correlation might have been masked by the mixing of waters from other tributaries, underscoring the combined impact of geographical factors (e.g., landscape) and anthropogenic influences (e.g., dam construction, as discussed below) on DOC sources.*" (P19 Line 455-458).

423: if there are significant damming/reservoirs on these rivers I think it would be useful to reference this in the site description

**Our response:** Thanks. We have added more information in the revised manuscript: "*Dams and reservoirs are widely distributed in three study catchments, and these dams are mainly used for agricultural irrigation and power generation (Fig. 1a). (P6 Line 145-146)*" and "*Furthermore, the widespread reservoirs for irrigation and water supply would lead to the prolonged water retention time across river systems, entailing a great change in organic carbon reactivity and $CO_2$ emissions (Catalán et al., 2016; Ran et al., 2021; Yi et al., 2021). Meanwhile, reservoirs provide a favorable environment for aquatic photosynthesis and bacterial production, thereby increasing autochthonous DOM production and accumulation in rivers (Ulseth and Hall, 2015; Xenopoulos et al., 2021). In addition to influencing DOC dynamics, our earlier research demonstrated that damming and reservoirs can also significantly affect the dynamics of POC and DIC (Chen et al., 2021). Additionally, DO was significantly different between dammed rivers and undammed rivers (Fig. 8), further indicating the damming effect on the in-stream photosynthesis and, consequently, on organic carbon dynamics.*" (P19 Line 459-466).

431: this sentence could be made clearer. I think you are point to the fact that it is difficult to disentangle the two influences and they likely affect DOC concentration and DOM quality in tandem in these rivers.

**Our response:** We have revised this sentence to make it clearer: "*However, it should be noticed that anthropogenic activities are spatially related to elevation, air temperature, and catchment slope (Fig. S4). Therefore, disentangling the dual influences (geographical and anthropogenic) is challenging, as they are likely to cohesively impact both DOC concentration and DOM quality in these rivers.*" (P19 Line 473-475).

454: 'deeper' is not needed

**Our response:** We have deleted 'deeper'.

463-464: this seems like repetition.

**Our response:** We have deleted this sentence.

Figure 2: please include Cl- concentrations as a panel.
**Our response:** We have added Cl- concentrations in the new figure.

[Figure]

**Figure 2.** Spatial variations in water chemistry in the Yinjiang (Y), Shiqian (S), and Yuqing (Q) rivers and springs. (**a**) DO, (**b**) Cl⁻, (**c**) $NH_4^+$-N, (**d**) $NO_3^-$-N, (**e**) DOC, and (**f**) $\delta^{13}C_{DOC}$. In each box plot, the end of the box represents the 25th and 75th percentiles, the blue solid dot represents the mean, the horizontal line inside the box represents the median, and whiskers represent 1.5 times the upper and lower interquartile ranges (IQR). The magenta solid dot represents the outlier (data points outside of the 1.5 interquartile ranges). Letters above the boxes represent significant differences between the grouping of river and/or spring water based on statistical analyses at the significance level of 0.05 (e.g., Y-Spring above panel (b) indicates that the Cl⁻ in river water of the Yinjiang River was significantly different from that in the spring water).

Figure 2 / 7: 'average' should be replaced with 'mean' in the caption.
**Our response:** We have replaced 'average' with 'mean' in the caption.

[Figure]

**Figure S3.** Fluorescence components identified by the PARAFAC model for the three rivers. Fluorescence peaks are C1 (295/402), C2 (275/338), and C3 (325/440), with wavelengths (nm) for excitation and emission, respectively. In each box plot, the end of the box represents the 25th and 75th percentiles, the blue solid dot represents the mean, the horizontal red line represents the median, and whiskers represent 1.5 IQR. The magenta solid dot represents the outlier, which is outside of the 1.5 interquartile ranges. Letters above the boxes represent significant differences between the grouping of rivers based on statistical analysis with $p < 0.05$.

Figure 6: if PARAFAC components identified in the 3 rivers then why is there only one rivers data shown in panels b and C and/or why has the coloring changed. why is C3 not plotted in figure 6?

**Our response:** Actually, there were data on three rivers shown in panels b and c. We have revised the color to show the data correctly. C3 was not significantly correlated with urban and agricultural land use, so it was not plotted here. We have added a related description in the revised manuscript: "*However, unlike C1 and C2, C3 was not significantly correlated with urban and agricultural land uses (p > 0.05; Fig. S4).*" (P14 Line 320).

[Figure]

**Figure S4** Correlation plot of the selected water chemistry and catchment characteristics. The colors represent the degree of pairwise correlation regarding Pearson's correlation coefficient. $\delta^{13}C_{DOC}$ and $\Delta^{14}C_{DOC}$ at site Y12 were excluded from analyses as the sample was collected after a rainfall event. In addition, $SUVA_{254}$ at site S3 was excluded from analyses as the sample was strongly influenced by the road construction, which was evidenced by high POC and TSM concentration (Chen et al., 2021). Human land use used here represents the proportion of urban and agricultural land use. Elevation and Annual $T_{air}$ represent mean drainage elevation and annual air temperature, respectively.

Figure 7: why are axes on EEMS not labeled. Please label axes and remove from caption.
**Our response:** We have labeled axes and removed it from the caption as you suggested.

[Figure]

**Figure S3.** Fluorescence components identified by the PARAFAC model for the three rivers. Fluorescence peaks are C1 (295/402), C2 (275/338), and C3 (325/440), with wavelengths (nm) for excitation and emission, respectively. In each box plot, the end of the box represents the 25th and 75th percentiles, the blue solid dot represents the mean, the horizontal red line represents the median, and whiskers represent 1.5 IQR. The magenta solid dot represents the outlier, which is outside of the 1.5 interquartile ranges. Letters above the boxes represent significant differences between the grouping of rivers based on statistical analysis with $p < 0.05$.

Figures: boxes around legends are not always present – see comment from first review.
**Our response:** We have added boxes around legends in figures where necessary (please see the manuscript).

**Response to Anonymous Referee #3**

I appreciate the authors' time and effort revising the manuscript substantially and addressing reviewer comments. While some of my concerns have been addressed, some of the methodological details provided are problematic and many details are still lacking. I believe that the authors need to provide significantly more details on their analyses and potentially revise their analyses/measurements to match standard protocol. Some issues (i.e., the lack of any bottle acid-washing or pre-combustion for DOC) I am not sure can be rectified at this point. Finally, there are some inconsistencies in the interpretation of trends in the discussion (see specific comments below).

**Our response:** Thank you very much for reviewing our manuscript again and providing valuable feedback. According to your suggestions, we have provided more methodological details, and we have also modified some interpretations of trends in the discussion. Please see the following responses for specific modifications.

**Methodological concerns**:

1- Sample bottles in which DOM are stored prior to analysis should be acid-washed (for plastic) or pre-combusted at 450°C for at least six hours for borosilicate glass (Mannino et al., 2019; Chow et al., 2022) to remove any organic carbon on bottle walls that can contaminate samples, especially when concentrations are as low as they are in this system. It is not acceptable to milli-Q rinse sample bottles for DOM without any acid-washing or baking, and I have never seen DOM work that does not pre-clean their sample bottles using acid or combustion. This is especially true when conducting isotopic analysis including radiocarbon.

**Our response:** Thank you for providing such specific comments and references. Although we did not acid-wash our sample bottles, we used brand-new never-used bottles and washed them with Milli-Q water before sampling. We also rinsed the samples at least three times prior to sampling. Therefore, we believe that organic contamination from the sample bottles can be ignored. However, we will use the acid-wash method you mentioned in future sampling to minimize the effects of organic contamination. In our study, for the subsequent isotope analysis (both $^{13}$C and $^{14}$C), we used borosilicate glass that had been pre-combusted at 450°C for at least six hours. We have added more details in the manuscript: "*The filtered water was stored in a Milli-Q water and sampling water pre-washed brand new low-density polyethylene container at low temperature (4°C) in the dark before optical properties analysis and acidified by phosphoric acid to pH = 2 for DOC analysis.*" (P6 Line 160-162).

2- The authors have provided no information on how catchment characteristics were calculated, only stating that it was calculated in ArcGIS. What data products were used (e.g., for land-use and for watershed boundaries)? What specific methods? Did you delineate sub-basin boundaries above each riverine sampling point and use that to calculate land-use? What reach length was used to calculate slope? This is the level of detail that is needed.

**Our response:** We have added more details here: "*Data on land use types and air temperature in 2015, as well as a 90 m digital elevation model (Shuttle Radar Topography Mission, SRTM) were obtained from the Resource and Environment Data Cloud Platform of the Chinese Academy of Sciences (http://www.resdc.cn/). Information on dams was retrieved from Wang et al. (2022), and their location was identified by Google Earth. Furthermore, the distance from the river mouth (i.e., the Yinjiang, Shiqian, and Yuqing rivers) to the sampling sites was also estimated using Google Earth. We further delineated the sub-catchments, which constitute the contributing area upstream of sampling sites, by spatial analyst tools of ArcGIS (version 10.2). The mean catchment slope (degrees; 3D analysis tools) and elevation for sub-catchments were extracted from the digital elevation model using ArcGIS. Annual air temperature, catchment slope, and proportion of urban and agricultural land uses for these sub-catchments were also determined using ArcGIS.*" (P4-5 Line 119-132).

3- I am not sure where the a254 threshold value of 0.1 cm-1 for inner-filter correction cited by the authors comes from, as it is not from Kothawala et al. (2013). According to Kothawala et al. (2013), inner-filter correction should be conducted when the sum of the absorbance of the excitation and emission wavelengths is greater than 0.05 cm-1. This was not tested by the authors to determine whether or not they needed to perform inner-filter correction. However, based on the presented a254 values, which averaged around 0.04 cm-1 (for a single wavelength), it seems likely that inner-filter correction would need to be performed once

accounting for the sum of both excitation and emission wavelengths. Substantial changes to fluorescence data, and therefore, PARAFAC results, can occur if not correcting for absorbance (see Kothawala et al., 2013, Fig. 2), which can be especially problematic if absorbance varies across samples (as it does in this study), because differences in absorbance could be driving observed differences in fluorescence.

**Our response:** Sorry for providing the wrong literature last time. We did not perform inner-filter correction because the inner filter effects could be ignored as the absorbance at 254 nm was lower than 0.3 $m^{-1}$ (Ohno, 2002). According to Ohno (2002), "An exact correction using explicit correction factors for both primary and secondary inner filtration effects was shown to give humification index values that are concentration invariant when absorbance of the solution at 254 nm was less than approximately 0.3 unit." All the absorbance at 254 nm of the water samples in this study was lower than 0.3 $m^{-1}$. We also tried to perform inner-filter correction to see the differences if inner-filter effect was not corrected. The differences (Absolute value of (uncorrected value - corrected value)/ corrected value *1000‰) were shown in the figure below. Most of the differences are lower than 10‰, indicating inner filtration effects can be ignored in this study.

[Figure]

Figures show the differences (Absolute value of (uncorrected value - corrected value)/ corrected value *1000‰) between inner-filter effect uncorrected and corrected values.

Reference:

Ohno T.: Fluorescence inner-filtering correction for determining the humification index of dissolved organic matter, Environ. Sci. Technol., 36, 742-746, doi:10.1021/es0155276, 2002.

4- The radiocarbon description here and in the cited paper with more detailed methods contains no information on how the

samples for radiocarbon measurement were stored and for how long, and how they were processed following measurement (for example, were Δ14C values corrected for fractionation using 13C?) I appreciate the authors additional information around the radiocarbon measurement 14C/12C background ratio, and I think the Dong et al. 2018 citation should be included when measurement on the AMS is mentioned (Ln 155-156): "…and measured by an accelerator mass spectrometry (AMS) system with an analytical error of ±3‰." However, I was referring to whether procedural blanks were conducted when processing the samples using wet chemical oxidation – a good example of this is Xu et al. (2021) (for UV oxidation), and I would argue this is necessary and should be standard practice whenever processing samples for radiocarbon using wet chemical oxidation. Please see Haghipour et al. (2019) for an example of how to use procedural blanks or standards to correct for contamination when processing samples using wet chemical oxidation.

**Our response:** After the samples were transformed into graphite, they were measured directly within 24 hours without long-term storage. The measured $\Delta^{14}$C values were corrected for fractionation using 13C. We have added the Dong et al. 2018 citation in this sentence: "*and measured by an accelerator mass spectrometry (AMS) system with an analytical error of ±3 ‰ (Dong et al., 2018)*" (P7 Line 184). As for the procedural blanks, five potassium acid phthalate (KPH) solutions as standard samples were processed and measured following the same method we analyzed radiocarbon for DOC samples. The results are shown below.

Table. Radiocarbon measurement results for standard samples (i.e., KPH).

| Sample ID | $\delta^{13}$C (‰) | %M ($^{14}$C/$^{13}$C) | σ (%M) | Age ($^{14}$C/$^{13}$C) | σ (Age) | $\Delta^{14}$C (‰) | Uncertainty |
|---|---|---|---|---|---|---|---|
| g81461 | -28.80 | 0.39 | 0.02 | 44666.00 | 328.00 | -996.15 | 39.85 |
| g81462 | -29.41 | 0.40 | 0.02 | 44438.00 | 338.00 | -996.04 | 41.00 |
| g81463 | -28.70 | 0.28 | 0.01 | 47240.00 | 439.00 | -997.21 | 53.26 |
| g81464 | -28.70 | 0.32 | 0.02 | 46270.00 | 435.00 | -996.85 | 52.53 |
| g81465 | -28.60 | 0.36 | 0.02 | 45205.00 | 356.00 | -996.40 | 43.18 |

In general, please include as much information as possible on sample handling, storage conditions, etc. for all measurements performed.

**Other comments**:

1. Autochthonous DOM introduced in the introduction but not defined.
**Our response:** We have added related information in the revised manuscript: "*Riverine DOM has both internal and external origins, namely autochthonous and allochthonous DOM. Autochthonous DOM is a pool of dead and living microbial and algal biomass that is derived within the aquatic ecosystem (Devesa-Rey and Barral, 2011).*" (P2 Line 48-50).

2. The organization of the introduction is very confusing – it needs a cohesive structure. I suggest the authors first define DOM and explain why it's important (this is mostly already done in the first paragraph). Then explain DOM source (allochthonous vs. autochthonous) and explain how source relates to DOM composition generally. Then explain how and why DOM source and composition might vary by geographic controls (elevation, slope, temperature) and land-use (agriculture, urbanization). End with your hypotheses/predictions based on what you have laid out in the earlier paragraphs.
**Our response:** Thank you. We have restructured the introduction and added related information, as you suggested. Please see "1 Introduction" (P1-4).

3. Ln 89-91: "We further hypothesize that there will be a large difference on DOM quality and carbon isotopes between these catchments and those with less influences by agricultural and urban land uses but steeper channel slopes and higher elevation" What kind of differences?

**Our response:** We have revised this sentence as: "*We hypothesize that catchments with a higher proportion of agricultural and urban land use, more gentle catchment slope, and lower elevation would exhibit higher riverine DOC concentrations and more autochthonous microbial humic-like DOM than steeper catchments at high elevations with fewer influences by agricultural and urban land uses.*" (P4 Line 106-108).

4. Table S1: I would suggest moving this to the main text in the methods so that it is easy to see what the different components represent.
**Our response:** Thanks. Table S1 has already been moved to the main text. (P8).

5. Figure 6a: SUVA values are the previous incorrectly-calculated values
**Our response:** Thank you. We have redrawn the figure using the correctly-calculated values.

[Figure]

**Figure 7.** Land use pattern impacts on DOM character. (**a**) SUVA$_{254}$, (**b**) freshness index (β/α), and (**f**) C2 decreased with the increasing proportion of urban and agricultural land uses. Outlier (site S3) was excluded from analysis in panel (a) as the sample was strongly influenced by road construction, which was evidenced by high POC and TSM concentration (Chen et al., 2021). (**c**) humification index (HIX) and (**e**) C1 were positively related to the increasing proportion of urban and agricultural land uses. However, there was no significant correlation between (**d**) fluorescence index and the proportion of urban and agricultural land uses.

6. Figure 2: Why are Δ14C -DOC and -POC not included, given that spatial variations are discussed?
**Our response:** Δ14C-DOC and Δ14C-POC are only available in the Yinjiang River. However, the water chemistry data in Figure 2 are derived from three rivers and spring water. Therefore, we present the 14CDOC data separately in Table 1. In addition, the 14CPOC data have already been reported in our previous study (Chen et al., 2021), so they are not presented here.

7. Figures 4-6: I don't really understand the organization of these figures. For example, why was 13C-DOC vs. NO3-N included in Figure 4, which seemingly seems to be all related to DOC amount and anthropogenic impact? There are a lot of indices/relationships to keep track of, and I wonder if a principal component analysis would visually be easier to follow (and driver wise easier to make sense of)?
**Our response:** Thanks. We have reorganized these figures. Some of the panels were moved to the supplement. As we responded

last time, we have tried to examine the trends through a PCA analysis, but it failed due to the limited number of $^{14}C$ data. Therefore, we have instead examined the trends in a correlation plot (Fig. S4), as shown below, to facilitate further discussion on the trends.

[Figure]

**Figure S4** Correlation plot of the selected water chemistry and catchment characteristics. The colors represent the degree of pairwise correlation regarding Pearson's correlation coefficient. $\delta^{13}C_{DOC}$ and $\Delta^{14}C_{DOC}$ at site Y12 were excluded from analyses as the sample was collected after a rainfall event. In addition, $SUVA_{254}$ at site S3 was excluded from analyses as the sample was strongly influenced by the road construction, which was evidenced by high POC and TSM concentration (Chen et al., 2021). Human land use used here represents the proportion of urban and agricultural land use. Elevation and Annual $T_{air}$ represent mean drainage elevation and annual air temperature, respectively.

8. Ln 234: "Unlike DOC, a strong negative correlation with mean channel slope was found for δ13CDOC (Fig. 3b)" There was a strong negative correlation for DOC concentration, and a positive correlation for 13C-DOC – is this a typo?
**Our response:** Yes. "negative" was replaced with "positive" in this sentence. (P13 Line 296).

9. Ln 283: "Authochthonous" spelling typo
**Our response:** "Authochthonous" has been replaced with "Autochthonous". (P11 Line 270).

10. Ln 287: Specify that you are referring to C1 and C2 percentage, not amount
**Our response:** Thanks. We have added the percentage information in this sentence as: "*In addition, the fluorescence components did not exhibit significant variations with changing catchment slope (p > 0.05; Fig. S4), but the percentage of C1 and C2 were positively (p < 0.05; Fig. 7b) or negatively (p < 0.01; Fig. 7c) related to the proportion of urban and agricultural land uses.*" (P14 Line 316-318).

11. Ln 311-312: "Furthermore, aged DOC in river systems has been attributed to old soil organic matter in deeper layer input into

rivers through deeper flow paths (Barnes et al., 2018; Masiello and Druffel, 2001)." Revise this sentence for clarity

**Our response:** We have revised it as: "*Furthermore, the aged DOC in river systems has been attributed to deeper, older soil organic matter inputs through deeper flow paths (Barnes et al., 2018; Masiello and Druffel, 2001).*" (P15 Line 348-350).

12. Ln 315: "The correlation of SUVA254 with mean channel slope suggests that steeper catchments tend to export DOC with less aromaticity (Fig. 3e), indicating the geomorphologic effects on DOM characteristics (Harms et al., 2016)." This is not consistent with what your data are showing, which is that SUVA is higher in high relief areas (so more aromatic). How would you interpret this?

**Our response:** Thanks. This is a typo. We have revised it as: "*The correlation of $SUVA_{254}$ with mean catchment slope suggests that steeper catchments tend to export DOC with more aromaticity (Fig. 4d), indicating the geomorphologic effects on DOM characteristics (Harms et al., 2016).*" (P15-16 Line 352-354).

13. Ln 328-334: It is unclear how this relates to your results.

**Our response:** We have added information related to our results: "*As a result, the remaining DOC with lower concentrations is typically characterized by a heavier $\delta^{13}C_{DOC}$ (Fig. 5a; Nkoue Ndondo et al., 2020; Opsahl and Zepp, 2001), which further indicates that the low-concentration DOC in the three rivers is the result of substantial microbial degradation.*" (P16 Line 370-372).

14. Ln 348: Mention that the groundwater is "spring" in Fig 2d

**Our response:** We have added this information in this sentence: "*However, groundwater was likely not the primary source of riverine DOC due to the relatively low groundwater DOC concentrations as compared with riverine DOC concentrations (Fig. 2e; groundwater is shown as "spring").*" (P16 Line 381-382).

15. Ln 335-351: I appreciate the thorough analysis regarding the contributions of groundwater, but I think you can shorten this paragraph a bit given that it turns out the groundwater is not an important source of DOC

**Our response:** Thanks. We have shortened this paragraph to: "*Groundwater with large SOC inputs due to highly active microbial activities has long been recognized as a significant source of DOC (McDonough et al., 2020; Shen et al., 2014). Several studies have reported increased groundwater contributions with distance downstream at the watershed scale (Asano et al., 2020; Cowie et al., 2017; Iwasaki et al., 2021). The positive relationship between conductivity and $\delta^{18}O$ is largely due to the mixing of two end-members (i.e., high-conductivity with $^{18}O$-enriched groundwater and low-conductivity with $\delta^{18}O$-depleted headstream water) for river water (Lambs, 2004), though it may also indicate the impact of evaporation in the catchment (Zhong et al., 2020). In addition, the $\delta^{18}O$ values increased progressively from upstream to downstream (Fig. S2b), which also validates the two sources (i.e., headstream water and groundwater) of downstream river water, indicating that groundwater was likely an important contributor to downstream river water. However, groundwater was likely not the primary source of riverine DOC due to the relatively low groundwater DOC concentrations as compared with riverine DOC concentrations (Fig. 2e; groundwater is shown as "spring"). Moreover, the groundwater contribution was probably much less significant in the wet season, even in catchments where DOC is mainly derived from groundwater (Lloret et al., 2016). Thus, we infer that groundwater is an important but not a primary source of riverine DOC in the three study rivers.*" (P16-17 Line 373-385).

16. Ln 352-358: I agree, however, your results do not really point to greater contributions of terrestrial DOM in the low relief areas – there is an increase in C1 the microbial component downstream, but a decrease in SUVA, and relatively little change in the freshness index or HIX. Maybe comparing the absolute value of the PARAFAC components (i.e., in Raman Units rather than percent) you could see if C3 increases in low relief compared to high relief (suggesting that the amount of terrestrial DOC input increases)?

**Our response:** We have compared the absolute value of the PARAFAC components in Raman Units. However, there is also no significant relationship found between the mean catchment slope and C3. We did not point to greater contributions of terrestrial

DOM in the low-relief areas in this study. In Ln 352-358, our aim is to point out other possible factors that may affect riverine DOC, which is related to the slope. Factors such as higher temperature can both increase terrestrial DOC input and enhance microbial decomposition, resulting in a decrease in DOC. Therefore, we do not draw a conclusion here but only provide possible impacts on DOC.

[Figure]

The figure shows the correlation between the mean catchment slope and C3 (Raman unit).

17. Ln 426: "This may have significantly influenced the organic carbon dynamics in the study rivers (Chen et al., 2021)." Can the authors elaborate on damming and reservoirs in this system? This seems like an extremely important point that was not addressed here.

**Our response:** We have added more details to elaborate on damming and reservoirs in this system: "*Furthermore, the widespread reservoirs for irrigation and water supply would lead to the prolonged water retention time across river systems, entailing a great change in organic carbon reactivity and $CO_2$ emissions (Catalán et al., 2016; Ran et al., 2021; Yi et al., 2021). Meanwhile, reservoirs provide a favorable environment for aquatic photosynthesis and bacterial production, thereby increasing autochthonous DOM production and accumulation in rivers (Ulseth and Hall, 2015; Xenopoulos et al., 2021). In addition to influencing DOC dynamics, our earlier research demonstrated that damming and reservoirs can also significantly affect the dynamics of POC and DIC (Chen et al., 2021). Additionally, DO was significantly different between dammed rivers and undammed rivers (Fig. 8), further indicating the damming effect on the in-stream photosynthesis and, consequently, on organic carbon dynamics.* (P19 Line 459-466)*"* We also added dam information in Figure 1a. Additionally, we compared DO in damming and undamming rivers to verify the damming effects on the river biogeochemical processes.

[Figure]

**Figure S7.** DO levels in dammed and undammed rivers. Dammed rivers refer to the rivers located immediately downstream of the dam in Fig. 1a, with no other sampling sites or tributary influences between the dam and the river. Undammed rivers are the rivers absent from dam impacts. The lowercase letters a and b above the boxes denote significant differences across rivers based on statistical analyses with $p < 0.05$.

18. Conclusions: Could you provide some explanation for why DOC age and optical properties are showing such different drivers? Other work would suggest that these should be highly related (Butman et al., 2012), and it seems odd that agriculture would influence the optics but not radiocarbon and the geomorphology vice versa.

**Our response:** Previous studies usually link Δ14C-DOC to SUVA254 (Aiken et al., 2014; Butman et al., 2012; Lee et al., 2021; Sickman et al., 2010). Although DOC age and SUVA254 are positively correlated in several studies (Aiken et al., 2014; Butman et al., 2012), there are also insignificant relationships found in data collected from Lee et al. (2021) (see figure below). For example, the major rivers in Korea and agricultural drains in California (separate data for wet winter months and other months) seem to have no significant correlation between Δ14C-DOC and SUVA254. Aiken et al. (2014) have attributed this relationship to "the influence of organic matter derived from leaf litter and upper soil horizons on the age of DOC in these rivers." However, Aiken et al. (2014) have reminded us that "caution is warranted in generalizing the use of DOM optical data to infer DOC age for systems that have not been adequately characterized." They provided an illustrative example to further demonstrate the potential masking effect of this relationship, "*For example, the St. Lawrence River, sampled approximately 115 km downriver of Lake Ontario, consistently contained low SUVA254, modern DOC due to the autochthonous production of DOM within the Great Lakes.*" Data in studied rivers is likely the same as this example that autochthonous DOM has masked the relationships between DOC age and optical properties. We have added this information in the revised manuscript: "*There was no significant relationship between carbon isotopes and optical properties, which is inconsistent with previous studies (Aiken et al., 2014; Butman et al., 2012; Lee et al., 2021; Sickman et al., 2010; Zhou et al., 2018). This is likely due to the potential masking effect of autochthonous DOM, as also evidenced by the decoupled relationship between $\Delta^{14}C_{DOC}$ and $SUVA_{254}$ in the St. Lawrence River (Aiken et al., 2014; Butman et al., 2012).*" (P19 Line 469-472).

[Figure]

**Fig. 5.** A relationship between $\Delta^{14}$C-DOC and SUVA$_{254}$ of composite stream water of each storm in the forested (W1) and the most agricultural (W4) watershed. The precipitation during the storms are 71.0 mm (storm 1), 87.5 mm (storm 2), 47.5 mm (storm 3), and 62.5 mm (storm 4). The empty circles are major rivers in the USA (Butman et al., 2012), and the rectangles are agricultural drains in California, USA (Sickman et al., 2010). The black rectangles are for wet winter months (Dec., 2003−Mar., 2004) and grey rectangles are for the other months (Sickman et al., 2010). The asterisks are the five largest rivers in South Korea, including the Han River, with blue asterisks for summer only (Lee et al., in review). (For interpretation of the references to color in this figure legend, the reader is referred to the Web version of this article.)

References:

Aiken G. R., Spencer, R. G. M., Striegl, R. G., Schuster, P. F. and Raymond, P. A.: Influences of glacier melt and permafrost thaw on the age of dissolved organic carbon in the Yukon River basin, Global Biogeochem. Cy., 28, 525-537, doi:10.1002/2013gb004764, 2014.

Butman D., Raymond, P. A., Butler, K. and Aiken, G.: Relationships between Delta C-14 and the molecular quality of dissolved organic carbon in rivers draining to the coast from the conterminous United States, Global Biogeochem. Cy., 26, GB4014, doi:10.1029/2012GB004361, 2012.

Lee S. C., Shin, Y., Jeon, Y. J., Lee, E. J., Eom, J. S., Kim, B. and Oh, N. H.: Optical properties and (14)C ages of stream DOM from agricultural and forest watersheds during storms, Environ. Pollut., 272, 116412, doi:10.1016/j.envpol.2020.116412, 2021.

Sickman J. O., DiGiorgio, C. L., Davisson, M. L., Lucero, D. M. and Bergamaschi, B.: Identifying sources of dissolved organic carbon in agriculturally dominated rivers using radiocarbon age dating: Sacramento-San Joaquín River Basin, California, Biogeochemistry, 99, 79-96, doi:10.1007/s10533-009-9391-z, 2010.

Zhou X., Johnston, S. E. and Bogard, M. J.: Organic matter cycling in a model restored wetland receiving complex effluent, Biogeochemistry, doi:10.1007/s10533-022-01002-x, 2022.

---

## Author Response (AR3)

**Response to Editor and Reviewers**

Dear Professor Shen,

Thank you for the thoughtful review of our manuscript (No. bg-2022-217) and the opportunity to submit a revision. We also greatly appreciate the very helpful review from all reviewers. We have revised the manuscript, taking into account all the comments we got.

All changes have also been highlighted in yellow in the revised version of the manuscript and the Supplementary Information. We hope the revised version of the manuscript will be acceptable for publication in the journal Biogeosciences. We look forward to hearing from you.

Thank you very much for your kind consideration.

Sincerely,

Lishan Ran

On behalf of all the authors

**Response to Anonymous Referee #1**

- There are some contradicting statements of how DOM chemistry change with elevation between lines 59-62 and lines 66-69. Do you use these examples to show that there are differing results in past studies?

**Our response:** Thanks. We have revised the first part (lines 59-62) in the revised manuscript to avoid misleading: "*A recent global study on lakes and rivers found that increasing elevation is associated with greater protein-like fluorescent DOM and lower specific ultraviolet absorbance at 254 nm ($SUVA_{254}$), which indicates the effect of enhanced UV radiation and accumulation of autochthonous DOM in higher elevation areas (Zhou et al., 2018).*" (P2 Line 62-64)

- Line 86: There are other papers you can cite here as well on DOM lability in agricultural streams and under high nutrient loads

**Our response:** We have added another paper here as you suggested: "*In addition, DOM tends to have a more reduced redox state and is likely more labile and accessible to the microbial community in agricultural streams when compared to the DOM found in natural streams (Fasching et al., 2019; Williams et al., 2010).*" (P3 Line 87-89)

- Line 417: I'm guessing this should say DOM characteristics

**Our response:** Thanks. We have replaced it with "*DOM characteristics" in the revised manuscript: "Anthropogenic impacts on DOM characteristics and age have been widely proposed in the last two decades (Butman et al., 2014; Coble et al., 2022; Vidon et al., 2008; Zhou et al., 2021).*" (P18 Line 423)

**Response to Anonymous Referee #4**

This manuscript describes relationships between riverine organic carbon concentration and quality in three mountainous catchments. The authors highlight the importance of catchment slope and anthropogenic land use as important factors affecting these organic carbon properties, and cite the literature extensively to support their hypotheses. There is a lot of interesting data collected and summarized, which is useful for comparisons with mountain stream and river ecosystems globally.

**Our response:** Thank you very much for your valuable opinions and suggestions. Details are provided in the specific response below.

Three major issues with the manuscript in its current form:

1) Even after several comments from previous reviewers, there remains some concerns about the methods employed and a lack of description in some cases. Additionally, with such multivariate data, it is concerning that the authors rely solely on univariate statistics. The complicated and sometimes overwhelming interpretation of linear regressions could be simplified with multivariate approaches.

**Our response:** Thanks. We have performed a stepwise multiple linear regression (MLR) modeling and the partial least squares path model (PLS-PM) as multivariate approaches to simplify the previous interpretation of linear regressions. Details are provided in the specific response below and the revised manuscript.

**Some specific issues in the Methods:**

Line 131: "Annual air temperature…[was] determined using ArcGIS." Perhaps this is in reference to a specific data layer, but as written it is unclear.

**Our response:** Thanks. We have showed the related information in the manuscript that the data on annual air temperature were obtained from a cloud platform: "*Data on land use types and air temperature in 2015, as well as a 90 m digital elevation model (Shuttle Radar Topography Mission, SRTM) were obtained from the Resource and Environment Data Cloud Platform of the Chinese Academy of Sciences (http://www.resdc.cn/).*" (P4 Line 112-114)

Line 136: Description of carbonate rocks suggests weathering rates could be high, but the impact of this on carbon ages is not thoroughly discussed.

**Our response:** Thanks. Yes, we have discussed the role of carbonate rocks in controlling chemical weathering and carbon isotopes of DIC in previous manuscript: "As discussed above, lithology is the underlying factor controlling the spatial distribution of DIC in these rivers (Fig. 3a). However, the $\delta^{13}C_{DIC}$ and $\Delta^{14}C_{DIC}$ did not show significant variations with increasing carbonate weathering intensity (Fig. 3b), implying that riverine carbon isotopes of DIC could be influenced by multiple biogeochemical processes. (please see (Chen et al., 2021))". This study was mainly focused on DOM and DOC dynamics. Thus, we did not include the above discussion in the revised manuscript. Further details are provided in (Chen et al., 2021).

Line 139: What is the spatial distribution of soil organic carbon or soil depth? No data on soil properties is described despite likely being the most important source of organic carbon to the streams.

**Our response:** Thanks. We have added spatial distribution of soil organic carbon as you suggested in the revised manuscript: "*The topsoil SOC exhibited a spatial distribution that resembled elevation, as areas with greater elevation displayed higher SOC content (Fig. S2).*" (P5 Line 128-129) We also added the figure in the Supplement, as shown below. Additionally, the role of SOC in controlling DOC dynamics was considered in the revised manuscript. For instance, SOC was included in the MLR model and the PLS-PM model, as shown in Table 4 and Figure 6 shown below.

[Figure]

Figure S2 Spatial distribution in SOC content in the surface layer (0–5 cm).

Table 4. Multiple stepwise linear regression models of catchment attributes and water chemistry on DOC concentrations and DOM properties.

| Dependent variables | Predictors | Model equation | $n$ | Adj $R^2$ | Significance level |
|---|---|---|---|---|---|
| DOC[a] | slope, $NH_4^+$-N | $= -0.109*slope + 4.295*NH_4^+\text{-}N + 3.375$ | 28 | 0.50 | $p < 0.001$ |
| DOC | SOC, POC | $= -0.006*SOC + 0.384*POC + 4.145$ | 28 | 0.59 | $p < 0.001$ |
| $SUVA_{254}$ | urban and agricultural land use, slope | $= -5.461*\text{urban and agricultural land use} + 0.145*slope + 1.318$ | 26 | 0.77 | $p < 0.001$ |
| HIX | urban and agricultural land use | $= 0.433*\text{urban and agricultural land use} + 0.438$ | 27 | 0.34 | $p < 0.001$ |
| FI | | No variables were entered into the equation. | 27 | | |
| $\beta/\alpha$ | pH | $= -0.195*pH + 2.476$ | 27 | 0.41 | $p < 0.001$ |
| C1 | DO, TP, urban and agricultural land use | $= 7.713*DO - 220.846*TP + 90.905*\text{urban and agricultural land use} - 36.005$ | 27 | 0.46 | $p < 0.001$ |
| C2 | urban and agricultural land use, DO | $= -48.748*\text{urban and agricultural land use} - 2.515*DO + 58.255$ | 27 | 0.36 | $p = 0.002$ |
| C3 | $NO_3^-$-N, POC | $= 4.181*NO_3^-\text{-}N + 3.738*POC + 3.826$ | 27 | 0.34 | $p = 0.003$ |

[a] SOC was not included as predictors in this model to examine the impacts of human activities and geomorphology, rather than the direct influence of SOC on DOC concentrations.

[Figure]

**Figure 6.** The most parsimonious PLS-PM model showing the direct and indirect effects of geomorphology and anthropogenic activities on DOC concentrations. (a) Path coefficients are shown as arrows with blue and red to represent positive and negative effects, respectively. The solid and dotted lines indicate the direct and indirect influence pathways of environmental drivers on DOC concentrations, respectively. The indicators (e.g., TN) of latent variables (e.g., nutrient) are shown at the beginning of the grey arrows. The numbers in the parentheses are the loading scores. GOF denotes the goodness of fit of the entire model. $R^2$ indicates the amount of variance in DOC concentrations explained by its independent latent variables. The standardized path coefficients that are significantly different from zero are indicated by $*p = < 0.05$, $** p = < 0.01$, $*** p = < 0.001$, $† p = 0.06$, $†† p = 0.07$. (b) Standardized direct and indirect mean effects of environmental drivers on DOC concentrations derived from the PLS-PM analysis.

Line 147: Runoff units would be helpful, either instead or additionally.

**Our response:** Thanks. We have added runoff in the revised manuscript as an additional information: "*This study area is highly affected by monsoon-influenced humid subtropical climate with April to October being the rainy season, and the average annual precipitation, runoff, and discharge are 1100 mm, 1004 mm/yr and 14.4 m³/s, respectively, in the Yinjiang River catchment.*" (P6 Line 141-143)

Line 152: Time of year of sampling is likely very important, but is not discussed later.

**Our response:** Thanks. Our sampling time (September 2018) is during the wet season (April–September). We have added it in the discussion as you suggested: "*Moreover, the groundwater contribution was probably much less significant in the wet season (e.g., September in the study area), even in catchments where DOC is mainly derived from groundwater (Lloret et al., 2016).*" (P17 Line 393-394)

Line 161: How long were samples stored?

**Our response:** Thanks. The period for sample storage was previously reported in 2.3 Laboratory analysis as "Refrigerated water samples for DOM absorbance and fluorescence were analyzed within one week after sampling." We now add this information earlier in the revised manuscript: "The filtered water was stored in a Milli-Q water and sampling water pre-washed brand-new low-density polyethylene container at low temperature (4°C) in the dark within one week before optical properties analysis and acidified by phosphoric acid to pH = 2 for DOC analysis." (P6 Line 156-158)

Line 170: Is this true for all samples?

**Our response:** Thanks. Yes, it is true. The normalized inorganic charge balance ranged between 0.02 % to 4.52 % for all samples. (P Line)

Line 172: Relative deviation of what? Did you take replicate field samples? Lab samples? This a concern for several

analyses, i.e., if duplicates were taken, and if so, what kind of duplicate was measured.

**Our response:** Thanks. Here, relative deviation means the relative deviation of lab samples. Replicate field samples were taken for several analyses (e.g., DOC, stable and radiocarbon isotopes) and measured when necessary.

Line 178: Which 9 samples? Is that all of the samples from the Yinjiang River? If not, why were they chosen?

**Our response:** Thanks. Yes, all of the [14]C samples are from the Yinjiang River. "*The Yinjiang River catchment has the greatest change in geomorphologic characteristics (i.e., elevation and channel slope) and the highest proportion of agricultural and urban land uses among the three catchments*" is one of the reasons we only collect 14C-DOC data in the Yinjiang River. The other reason is the expensive analytical cost (McNichol and Aluwihare, 2007), which is about $500 per sample.

Reference: *McNichol A. P. and Aluwihare, L. I.: The power of radiocarbon in biogeochemical studies of the marine carbon cycle:Insights from studies of dissolved and particulate organic carbon (DOC and POC), Chem. rev., 107, 443-466, doi:10.1002/chin.200724246, 2007.*

Line 186: At what interval?

**Our response:** Thanks. We have added this information in the revised manuscript as: "*DOM absorbance of river water samples was measured from 250 to 750 nm at 1 nm intervals using a UV (ultraviolet)-visible spectrophotometer (UV-2700, Shimadzu) with a 1 cm quartz cuvette.*" (P7 Line 181-183)

Line 196: Is there any concern about DOM properties changing in 1 week?

**Our response:** Thanks. DOM absorbance and fluorescence analyzed within one week was a suggested period for DOM optical measurement (Coble et al., 2014).

Reference: *Coble P. G., Lead, J., Baker, A., Reynolds, D. M. and Spencer, R. G.: Aquatic organic matter fluorescence, Cambridge University Press, 2014.*

Line 215: If you are doing all of these correlations, how do you determine what is "predominant?" Anything significant? Do you have concerns about doing so many univariate analyses?

**Our response:** Thanks. Following your suggestions above, we have performed stepwise multiple linear regression (MLR) modeling and the partial least squares path model (PLS-PM) as multivariate approaches to identify the dominant drivers of DOC dynamics. Details are provided in "2.4. Statistical analysis", "Results" and "Discussion" of the revised manuscript.

Line 220: A mean is not necessarily representative if you are working with small sample sizes and non-normal data.

**Our response:** Thanks. We have compared the mean, median and their differences (i.e., (Mean-Median)/Mean) of the major dataset we reported in the table below. The results showed that the differences are mostly lower than 10%, which indicated the mean is also an appropriate value to represent our data. In addition, for the data we reported mean value in the manuscript, we also provided box plots (e.g., Figs. 2 and 3) for these data, which is useful for the reader to have a better understanding of the distribution of data.

| | DOC | $\delta^{13}C_{DOC}$ | $\Delta^{14}C_{DOC}$ | SUVA$_{254}$ | FI | HIX | $\beta/\alpha$ | Cl | NH4-N | NO3-N | pH |
|---|---|---|---|---|---|---|---|---|---|---|---|
| Mean | 1.5 | -26.8 | -54.7 | 3.2 | 1.78 | 0.55 | 0.78 | 3.1 | 0.057 | 1.1 | 8.29 |
| Median | 1.33 | -25.7 | -52.2 | 3.0 | 1.76 | 0.59 | 0.74 | 3.3 | 0.06 | 0.83 | 8.21 |
| (Mean-Median)/Mean | 11.3 | 4.2 | 4.6 | 6.8 | 1.1 | -7.3 | 5.1 | -5.1 | -5.3 | 24.5 | 1.0 |

Line 221: Sometimes removing and sometimes including a data point seems difficult to defend without a specific metric.

**Our response:** Thanks. The reason why we did not remove Y12 in the whole study was because radiocarbon isotopes of the Y12 were measured, and these data can provide additional information on carbon dynamics under the influence of heavy rainfall events. In addition, we only exclude Y12 in Figures 4 and S4, where we add details on the data selection in figure captions. So the reader can easily distinguish the impact of an outlier on the research results and the reason why we did not

include outlier in the result analysis.

Line 302: Anthropogenically derived chloride is not defined in the text.

**Our response:** Thanks. Anthropogenically derived chloride was discussed in the caption of Fig. 5 in the previous manuscript. We have moved it earlier in the revised manuscript as: "The environmental factors used in the model were categorized into seven latent variables, including geomorphology (elevation and slope), anthropogenic activities (e.g., urban and agricultural land uses and anthropogenically derived $Cl^-$ ($Cl^-_{anthro}$, calculated as the total $Cl^-$ concentration minus atmospheric contributed $Cl^-$ concentration, which is the lowest $Cl^-$ concentration at site Y5 in the Yinjiang River; Gaillardet et al., 1997; Meybeck, 1983))". (P Line)

2) The Results are exhaustive, and as a result, quite unfocused. It is unclear which results are important to the central message of the manuscript. This is indicative of a lack of clear focus within the entire manuscript. As previous reviews have pointed out, there are a lot of figures and tables. I would agree and suggest that most of these figures are relatively simple and do little else than to show correlation. Some higher-level analysis or results would be greatly beneficial, including some multivariate approaches.

**Our response:** Thanks. We have deleted many figures to avoid exhaustive, especially those figures showing the regression correlation. For example, previous Figs. 4-7 were removed in the revised manuscript. Then, we applied multivariate approaches in this study following your suggestions, such as stepwise multiple linear regression (MLR) modeling and the partial least squares path model (PLS-PM). Details are provided in the "2.4. Statistical analysis" of the revised manuscript: "*We performed a stepwise multiple linear regression (MLR) modeling to identify significant environmental factors of DOC concentrations and DOM properties using SPSS 26. All environmental factors were included in the models except for SOC, because we aim at examining the impacts of human activities and geomorphology rather than the direct influence of SOC on DOC concentrations and DOM properties. The objective model with the highest adjusted $R^2$ value was used to infer the DOC concentrations and DOM properties. In addition to the MLR and Pearson correlation analyses to explore the relationships between environmental factors and DOC, we further performed the partial least squares path model (PLS-PM) to infer direct and indirect effects of multiple factors (e.g., geomorphologic and anthropogenic impacts) on DOC concentrations. The PLS-PM analysis was performed using the R package "plspm" (Sanchez, 2013). Because PLS-PM offers the advantage of not imposing any distributional assumptions on the data, which enhances its broad applicability (Sanchez, 2013), and allows for the exploration of complex cause-effect relationships involving latent variables, it is a suitable technique for multivariate analyses . Each latent variable consists of one or more manifest variables (e.g., geomorphology, including elevation and slope). The environmental factors used in the model were categorized into seven latent variables, including geomorphology (elevation and slope), anthropogenic activities (e.g., urban and agricultural land uses and anthropogenically derived $Cl^-$ ($Cl^-_{anthro}$, calculated as the total $Cl^-$ concentration minus atmospheric contributed $Cl^-$ concentration, which is the lowest $Cl^-$ concentration at site Y5 in the Yinjiang River; (Gaillardet et al., 1997; Meybeck, 1983))), climate ($T_{air}$), SOC (SOC content), water chemistry (pH), POC (POC concentrations) and nutrient ($NH_4^+$-N and TN). The environmental factors and their manifest variables included in the model were the most critical variables identified based on the Pearson correlation results. These variables were selected after reducing the full models (initial models with more variables) to meet the requirements of the PLS-PM analysis (Du et al., 2023; Sanchez, 2013; Tian et al., 2019). In addition, the structure of the model was simplified to focus on the major effect of environmental factors on DOC concentrations rather than to explore the effects on other factors (e.g., the geomorphologic controls on POC were ignored). The significance of the path coefficients was determined through a nonparametric bootstrap resampling of 1000 times.*" (P8-9 Line 216-239)

Further results of the MLR and PLS-PM are shown in Table 4 and Figs. 6, S4, and S5. Please see the revised manuscript for more details.

**Table 4.** Multiple stepwise linear regression models of catchment attributes and water chemistry on DOC concentrations and DOM properties.

| Dependent variables | Predictors | Model equation | $n$ | Adj $R^2$ | Significance level |
|---|---|---|---|---|---|
| DOC[a] | slope, $NH_4^+$-N | $= -0.109*slope + 4.295*NH_4^+$-$N+ 3.375$ | 28 | 0.50 | $p < 0.001$ |
| DOC | SOC, POC | $= -0.006*SOC + 0.384*POC + 4.145$ | 28 | 0.59 | $p < 0.001$ |
| SUVA$_{254}$ | urban and agricultural land use, slope | $= -5.461*$urban and agricultural land use $+$ $0.145*slope+1.318$ | 26 | 0.77 | $p < 0.001$ |
| HIX | urban and agricultural land use | $= 0.433*$urban and agricultural land use $+$ $0.438$ | 27 | 0.34 | $p < 0.001$ |
| FI | | No variables were entered into the equation. | 27 | | |
| $\beta/\alpha$ | pH | $= -0.195*pH + 2.476$ | 27 | 0.41 | $p < 0.001$ |
| C1 | DO, TP, urban and agricultural land use | $= 7.713*DO - 220.846*TP +90.905*$urban and agricultural land use $- 36.005$ | 27 | 0.46 | $p < 0.001$ |
| C2 | urban and agricultural land use, DO | $= -48.748*$urban and agricultural land use $- 2.515*DO + 58.255$ | 27 | 0.36 | $p = 0.002$ |
| C3 | $NO_3^-$-N, POC | $= 4.181*NO_3^-$-$N + 3.738*POC + 3.826$ | 27 | 0.34 | $p = 0.003$ |

[a] SOC was not included as predictors in this model to examine the impacts of human activities and geomorphology, rather than the direct influence of SOC on DOC concentrations.

[Figure]

**Figure 6**. The most parsimonious PLS-PM model showing the direct and indirect effects of geomorphology and anthropogenic activities on DOC concentrations. (a) Path coefficients are shown as arrows with blue and red to represent positive and negative effects, respectively. The solid and dotted lines indicate the direct and indirect influence pathways of environmental drivers on DOC concentrations, respectively. The indicators (e.g., TN) of latent variables (e.g., nutrient) are shown at the beginning of the grey arrows. The numbers in the parentheses are the loading scores. GOF denotes the goodness of fit of the entire model. $R^2$ indicates the amount of variance in DOC concentrations explained by its independent latent variables. The standardized path coefficients that are significantly different from zero are indicated by $*p = < 0.05$, $** p = < 0.01$, $*** p = < 0.001$, $\dagger p = 0.06$, $\dagger\dagger p = 0.07$. (b) Standardized direct and indirect mean effects of environmental drivers on DOC concentrations derived from the PLS-PM analysis.

[Figure]

**Figure S4** The most parsimonious PLS-PM showing the direct and indirect effects of geomorphology and anthropogenic activities on fluorescent components. (a) Path coefficients are shown as arrows with blue and red to represent positive and negative effects, respectively. The solid and dotted lines indicate the direct and indirect influence pathways of environmental drivers on fluorescent components, respectively. The indicators (e.g., TN) of latent variables (e.g., nutrient) are shown at the beginning of the grey arrows. The numbers in the parentheses are the loading scores. GOF denotes the goodness of fit of the entire model. $R^2$ indicates the amount of variance in fluorescent components explained by its independent latent variables. The standardized path coefficients that are significantly different from zero are indicated by $*p = < 0.05$, $** p = < 0.01$, $*** p = < 0.001$. (b) Standardized direct and indirect mean effects of environmental drivers on fluorescent components derived from the PLS-PM analysis. C1 was initially included in the model but had to be removed to fulfill the requirements of the model analysis.

[Figure]

**Figure S5** The most parsimonious PLS-PM showing the direct and indirect effects of geomorphology and anthropogenic activities on DOM optical parameters. (a) Path coefficients are shown as arrows with blue and red to represent positive and negative effects, respectively. The solid and dotted lines indicate the direct and indirect influence pathways of environmental drivers on DOM optical parameters, respectively. The indicators (e.g., TN) of latent variables (e.g., nutrient) are shown at the beginning of the grey arrows. The numbers in the parentheses are the loading scores. GOF denotes the goodness of fit of the entire model. $R^2$ indicates the amount of variance in DOM optical parameters explained by its independent latent variables. The standardized path coefficients that are significantly different from zero are indicated by $*p = < 0.05$, $** p = < 0.01$, $*** p = < 0.001$. (b) Standardized direct and indirect mean effects of environmental drivers on DOM optical parameters derived from the PLS-PM analysis. FI was initially included in the model but had to be removed to meet the requirements of the model analysis.

3) The Discussion section is written in a way that does not clearly differentiate what are interpretations of this study and what are supporting ideas from other studies. Perhaps this is because the interpretation is complex, but the authors do not clearly establish what are their main conclusions or interpretations. Line 474 is telling, "disentangling the dual influences…is challenging." The results do not accomplish this, the discussion does not do much to clarify either. But the analysis is perhaps what is most limiting, in which univariate statistics are repeatedly used to approach a complex and multivariate problem. The repeated interpretation of correlation as causation is also problematic. Additionally, while the authors mention how land use and slope are not independent variables towards the end of the discussion, this should be a major caveat and approached more thoroughly early in the manuscript.

**Our response:** Thanks. We have restructured the discussion by removing many paragraphs/sentences that are not so clear or insignificant based on our new analysis. As showed in the last question, we have performed a stepwise multiple linear regression (MLR) modeling and the partial least squares path model (PLS-PM) to deal with the complex and multivariate problem. Using these tools, we made significant changes to the manuscript, especially the results and discussion. We believe that our new results and discussion are clear enough to demonstrate the dual influences of geomorphologic characteristics and anthropogenic activities on DOM. For example, PLS-PM analysis has clearly indicated the direct and indirect effects of environmental factors on DOC concentrations: "*The PLS-PM analysis showed that 67% of the variance in DOC concentrations could be explained by our constructed seven environmental factors ($R^2 = 0.67$, Fig. 6a). The total effect on DOC concentrations is strongest from geomorphology (-0.65), followed by SOC (-0.45), anthropogenic activities (0.39), climate (0.38), POC (0.27), nutrient (0.21), and water chemistry (0.10) (Fig. 6b). The results indicated that geomorphology was the most significant factor in controlling DOC concentrations, primarily through indirect regulation on SOC content, which was directly influenced by annual catchment temperature and anthropogenic activities (Figs. 6a and b). In comparison, anthropogenic activities not only indirectly regulated riverine DOC concentrations through SOC, but also had a significant indirect impact through the regulation of nutrient levels. Similar to DOC concentrations, geomorphology (-0.53) exhibited the most pronounced effects on fluorescent components (Fig. S4). However, anthropogenic activities (0.49) demonstrated a comparable effect on fluorescent components, primarily through a direct pathway (0.37; Fig. S4b). Anthropogenic activities (-0.84) were the strongest driver for DOM optical parameters, although geomorphology (0.59) played a significant role in indirectly influencing DOM optical parameters (Fig. S5).*" The PLS-PM analysis also clearly shows us the correlation between land use and slope (Fig. 6). (P14 Line 332-342)

**Response to Anonymous Referee #5**

The manuscript here explores how geographical and anthropogenic factors impact DOM abundance and composition. This topic is important in global carbon cycling, especially under the scenario of climate change and intensified anthropogenic activities.

**Our response:** Thank you for your very helpful review of our manuscript once again and for giving us very useful comments. We have already depleted some of the unnecessary figure panels and previously published data in the modified manuscript, such as Figs. 1 and 4. We also streamlined the discussion text, such as the discussion on groundwater contribution on riverine DOC. Details are provided in the specific response below.

However, here are some specific comments which should be addressed before acceptance.

**Specific comments**:

L56-57 The authors should check this sentence for grammar and logic issues: "Recent studies have shown that geographical (e.g., elevation and catchment slope) controls on DOC export may also be important for riverine carbon cycling (Connolly et al., 2018; Li Yung Lung et al., 2018).", and also the reference format here "Li Yung Lung et al., 2018".

**Our response:** Thanks. We have rewritten this sentence as: "*Recent studies have indicated the significance of geomorphologic factors, such as elevation and catchment slope, in influencing the export of DOC and riverine carbon cycling (Connolly et al., 2018; Li Yung Lung et al., 2018)*." The reference format here, "Li Yung Lung et al., 2018" is correct as the full name of the first author is "Joanna Y. S. Li Yung Lung" (P2 Line 58-59). The official citation is "Li Yung Lung, J. Y. S., Tank, S. E., Spence, C., Yang, D., Bonsal, B., McClelland, J. W., & Holmes, R. M. (2018). Seasonal and geographic variation in dissolved carbon biogeochemistry of rivers draining to the Canadian Arctic Ocean and Hudson Bay. Journal of Geophysical Research: Biogeosciences, 123, 3371–3386. https://doi.org/10.1029/2018JG004659"

L59: "characterized by greater releases of DOC", what does this mean? "Release of DOC", from where or to where? "Greater releases", the quantity or the flux or the rate?

**Our response:** Thanks. We have rewritten it to make it clear: "*Compared with high-relief catchments, low-relief regions with longer water residence time, stronger hydrologic connectivity to rivers, and greater development of wetlands are typically characterized by increased concentration of riverine DOC*" (P2 Line 59-62)

L60: "protein-like fluorescence DOM" should be "protein-like fluorescent DOM".

**Our response:** Thanks. We have replaced it with "*protein-like fluorescent DOM*". (P3 Line 63)

L134: The authors should check the value of the slopes ("mean catchment slope (from 14.3° to 25.5°)"). Is the magnitude right? Not one order less? I'm not familiar with this, however, it should be one order less considering the spatial scale (30-60 km) and the altitude difference (about 2 km).

**Our response:** Thanks for your concerns. We have checked the degree of slope. The study area is mountainous area, which leads to the high catchment slope. This magnitude is reasonable, as evidenced by other studies (e.g., (Harms et al., 2016)).

L189-190: The authors should check the definition of a254 and A254, and also the unit of absorption coefficients. The definition and calculation of absorbance (A254) and the absorption coefficient (a254) could refer Hu et al. (2002) and Li et al. (2017). Meanwhile, commonly the unit of a254 is m-1, not cm-1.

**Our response:** Thanks. We have checked the definition of a254 and A254 as you suggested and changed the unit of a254 to m$^{-1}$. (P7 Line 186)

Ref: Hu C, Muller-Karger F E, Zepp R G. Absorbance, absorption coefficient, and apparent quantum yield: A comment on common ambiguity in the use of these optical concepts[J]. Limnology and Oceanography, 2002, 47(4): 1261-1267.
Li P, Hur J. Utilization of UV-Vis spectroscopy and related data analyses for dissolved organic matter (DOM) studies: A

review[J]. Critical Reviews in Environmental Science and Technology, 2017, 47(3): 131-154.

L194-206: Have the authors considered the inner filter correction for fluorescence data?

**Our response:** Thanks. As we discussed earlier with another reviewer, we did not perform inner-filter correction because the inner filter effects could be ignored as the absorbance at 254 nm was lower than 0.3 m-1 (Ohno, 2002). According to Ohno (2002), "An exact correction using explicit correction factors for both primary and secondary inner filtration effects was shown to give humification index values that are concentration invariant when absorbance of the solution at 254 nm was less than approximately 0.3 unit." All the absorbance at 254 nm of the water samples in this study was lower than 0.3 m-1. We also tried to perform innerfilter correction to see the differences if inner-filter effect was not corrected. The differences (Absolute value of (uncorrected value - corrected value)/ corrected value *1000‰) were shown in the figure below. Most of the differences are lower than 10‰, indicating inner filtration effects can be ignored in this study.

[Figure]

Figures show the differences (Absolute value of (uncorrected value - corrected value)/ corrected value *1000‰) between inner-filter effect uncorrected and corrected values.

Reference:

Ohno T.: Fluorescence inner-filtering correction for determining the humification index of dissolved organic matter, Environ. Sci. Technol., 36, 742-746, doi:10.1021/es0155276, 2002.

L225: The authors should check the definition and calculation of SUVA254 in Table 2. In Weishaar et al. (2003), they use A254, not a254.

**Our response:** Thanks. According to (Weishaar et al., 2003) et al. (2003), "SUVA254 is defined as the UV absorbance at 254 nanometers measured in inverse meters (m$^{-1}$) divided by the DOC concentration measured in milligrams per liter (mg

L$^{-1}$).” In application, SUVA$_{254}$ was calculated by dividing the decadic absorption coefficient at 254 nm by DOC concentration (Luzius et al., 2018; Poulin et al., 2014; Weishaar et al., 2003). Thus, the SUVA$_{254}$ we provided in this study was correct.

L225: Will the authors consider the updated calculation and interpretation of fluorescence index (FI)? FI = Em470/Em520, at Ex 370 nm. The authors could refer to the updated reference (Cory R M, Miller M P, McKnight D M, et al. Effect of instrument-specific response on the analysis of fulvic acid fluorescence spectra[J]. Limnology and Oceanography: Methods, 2010, 8(2): 67-78.).
**Our response:** Thanks for your suggestions. I would like to update the calculation of fluorescence index as you suggested. However, we measured the emission wavelength from 280 to 500 nm at 2 nm increments, which makes the update impossible. We will measure the emission wavelength in a wider range in future research.

L229: “between” or “among”?
**Our response:** Thanks. The original sentence was removed in the revised manuscript as we have replaced DO with the pH.

L278-279: The term used here should be consistent with Figure 3. What does FI mean here? Fluorescence index? Then it should be related to Figure 3b according to the Y label of Figure 3b. But the authors related FI here with Figure 3d (“FI of DOM ranged from 1.66 to 1.94, averaging 1.78 (Fig. 3d)”), while the Y label for Figure 3d is freshness index.
**Our response:** Thanks. We are sorry for the misleading. We have updated this figure, and figure caption to show the DOM property correctly.

[Figure]

**Figure 3** Spatial variations in DOM property in the Yinjiang (Y), Shiqian (S), and Yuqing (Q) catchments. (a) SUVA$_{254}$, (b) fluorescence index (FI), (c) HIX, and (d) freshness index ($\beta/\alpha$). In each box plot, the end of the box represents the 25th and 75th percentiles, the blue solid dot represents the average, the horizontal red line represents the median, and whiskers represent 1.5 IQR. The magenta solid dot represents the outlier, which is outside of the 1.5 interquartile ranges. Different lowercase letters above the boxes denote significant differences across rivers based on statistical analysis with $p < 0.05$.

L294: The authors should provide the references to support/indicate why Cl- is anthropogenically derived here.
**Our response:** Thanks. We have provided the references in the revised manuscript: “*The environmental factors used in the*

*model were categorized into seven latent variables, including geomorphology (elevation and slope), anthropogenic activities (e.g., urban and agricultural land uses and anthropogenically derived $Cl^-$ ($Cl^-_{anthro}$, calculated as the total $Cl^-$ concentration minus atmospheric contributed $Cl^-$ concentration, which is the lowest $Cl^-$ concentration at site Y5 in the Yinjiang River; Gaillardet et al., 1997; Meybeck, 1983))*" (P9 Line 227-230)

Figure 5: The box for legend did not display well.

**Our response:** Thanks. We have deleted this figure as we added a correlation plot and PLS-PM model in the revised manuscript.

Figure 5: As stated in the introduction part (L80-89) and L393-394, agricultural and urban land use exerted different impacts on DOM abundance and composition, why the authors combined these two landuse patterns to demonstrate the anthropogenic impacts, rather than separating these two to provide detailed explanation?

**Our response:** Thanks. We combined these two landuse patterns is mainly due to two reasons. On the one hand, urban land use only accounts for a small proportion of the total land use area (Fig. 1c). On the other hand, urban rivers shares some similarities with agricultural rivers in many studies, so it is common to combine them as an indicate of human disturbance (e.g., (Butman et al., 2014)). To avoid misunderstanding on the difference between urban and agricultural rivers, we have deleted the following content in the revised manuscript: "*Although the DOM in urban rivers shares some similarities with agricultural rivers (such as microbial origins), the sources of DOM in urban rivers are much more complex, which may originate from urban point-source inputs (e.g., wastewater treatment facilities) and nonpoint source inputs (e.g., household sewage and petroleum-based hydrocarbons) (Hosen et al., 2014).*"

L352-354: Both geographical and anthropogenic factors influence the DOM abundance and composition. Here, the export of DOC with higher aromaticity may be due to the landuse pattern (Fig 7(a), and steeper catchments could be coincident. How to differentiate these factors here?

**Our response:** Thanks. We performed a stepwise multiple linear regression (MLR) modeling to identify significant environmental factors of DOM properties in the revised manuscript. Additionally, we performed the partial least squares path model (PLS-PM) to infer the direct and indirect effects of multiple factors (e.g., geomorphologic and anthropogenic impacts) on DOC concentrations. These analyses enable us to differentiate these factors. As shown in the revised manuscript: "*$SUVA_{254}$ showed an increasing trend with increasing mean catchment slope ($p < 0.001$; Fig. 4). Furthermore, there was a significant negative correlation between $SUVA_{254}$ and the proportion of urban and agricultural land uses ($p < 0.001$; Fig. 4). This is consistent with the constructed stepwise MLR models that urban and agricultural land uses and catchment slope were the best predictors of $SUVA_{254}$ (Table 4)*" (P13 Line 312-315); "*However, anthropogenic activities (0.49) demonstrated a comparable effect on fluorescent components, primarily through a direct pathway (0.37; Fig. S4b). Anthropogenic activities (-0.84) were the strongest driver for DOM optical parameters, although geomorphology (0.59) played a significant role in indirectly influencing DOM optical parameters (Fig. S5)*" (P14 Line 340-342). We can conclude that anthropogenic activities are more important in controlling the aromaticity of DOM than geomorphology.

L366: "increasing DOC concentrations … due to microbial degradation"? Please check the logic.

**Our response:** Thanks. We have revised it as: "*Previous studies have reported a decreasing $\delta^{13}C_{DOC}$ with increasing DOC concentrations (Fig. 4) in spring water (Nkoue Ndondo et al., 2020) and for TOC in soil profiles*" (P16 Line 376-377)

L401-402: The authors should provide more details for this statement "anthropogenic impacts can also decrease DOC concentrations". How?

**Our response:** Thanks. We have added details on the reasons for the decrease of DOC in the revised manuscript: "*Yet, anthropogenic impacts can also result in decreased DOC concentrations globally due to reduced organic carbon inputs into soils and enhanced SOC decomposition induced by warmer temperatures (Nagy et al., 2018; Spencer et al., 2019) or lead to undetectable changes in DOC concentrations (Veum et al., 2009).*" (P17 Line 404-407)

L428: "consistent with previous studies", how to explain this pattern?

**Our response:** Thanks. We have added more details to explain this pattern: "*Lower DOM aromaticity in the urban and agricultural streams and rivers was consistent with previous studies (Hosen et al., 2014; Kadjeski et al., 2020), which suggested a microbial origin for the DOM.*" (P18 Line 434-435)

L451-453: If "the weak positive correlation …indicated that DOC and POC may have been derived from the same source", what does the strong positive correlation indicate? The authors could rephrase this sentence to avoid ambiguity.

**Our response:** Thanks. We have deleted this sentence in the revised manuscript to avoid ambiguity. The correlation between DOC and POC is partly explained by: "Geomorphology is also associated with the reduction in water retention time due to rapid flows, leading to a lower input of terrestrially-derived DOC into rivers as discussed earlier. It is worth noting that the conversion of POC to DOC through dissolution and desorption (He et al., 2016) is also an important source of riverine DOC (Fig. 6)". (P19 Line 449-452)

---

## Author Response (AR4)

**Response to Editor**

Dear Professor Shen,

Thank you for the thoughtful review of our manuscript (No. bg-2022-217) and the chance to submit a revised version. We sincerely appreciate your valuable suggestions and generous assistance. We have incorporated the changes you recommended into the revised manuscript.

All changes have also been highlighted in yellow in the revised version of the manuscript and the Supplementary Information. We hope the revised version of the manuscript will be acceptable for publication in the journal Biogeosciences. We look forward to hearing from you.

Thank you very much for your kind consideration.

Sincerely,

Lishan Ran

On behalf of all the authors

**Response to editor**

Regarding the computation of SUVA254, it is accurate to utilize the decadic term of absorbance (i.e., absorbance normalized to path length, measured in m-1) for SUVA254 calculations, following the method outlined in Weishaar et al. 2003. However, in this context, the symbol representing the decadic absorption coefficient is italicized alpha (α), not the letter a. The letter a (or more often in its italic form) is typically employed to denote the Napierian absorption coefficient, where a is equal to α times ln10. To align with the correct notation, the authors can simply substitute the term "a254" with "α254" (alpha α in its italicized form) in equation (1) and add "decadic" before "absorption coefficient" in the subsequent line.

**Our response:** Thanks for your clarification on definition on SUVA254 calculations. We have substituted the term "a254" with "α254" throughout the manuscript and added "decadic" before "absorption coefficient" in the revised manuscript: "

Decadic absorbance values were used to calculate decadic absorption coefficients as below (Poulin et al., 2014):

$$\alpha_{254} = Abs_{254}/L, \tag{1}$$

Where, $\alpha_{254}$ is the decadic absorption coefficient ($m^{-1}$), $Abs_{254}$ is the absorbance at 254 nm, and L represents the path length (m). Specific UV absorbance at 254 nm (SUVA$_{254}$; reported in units of L mg $C^{-1}$ $m^{-1}$) was determined according to Weishaar et al. (2003; Table 1):

$$SUVA_{254} = \alpha_{254}/DOC. \tag{2}$$

" (P7 Line 180-186)

line 237: remove "were" in the title of Table 2.

**Our response:** Thanks. We have remove "were" in the title of Table 2: "DOM optical parameters used in this study." (P9 Line 235)

line 272: define "SD". I assume it stands for Standard Deviation.

**Our response:** Thanks. We have added the definition of SD in "2.4. Statistical analysis" as: "*Values are presented as the mean ± standard deviation (SD)*." (P8 Line 210)

lines 298-324: Section 3.3, I think it would be more informative to report r values in addition to p values. The r values help evaluate the strength and direction of the correction described in the text.

**Our response:** Thanks. We have added r values in "section 3.3" to strength our results. "Section 3.3" was shown below: "Significant pairwise interdependencies between DOC and catchment characteristics were identified in the three study rivers (Fig. 4). There is a strong negative correlation between DOC and SOC ($p < 0.001$, $r = -0.73$; Fig. 4), as well as average catchment slope ($p < 0.001$, $r = -0.67$). Conversely, DOC displayed a positive correlation with the proportion of urban and agricultural land uses ($p < 0.001$, $r = 0.62$), $Cl^-_{anthro}$ ($p < 0.01$, $r = 0.58$), and $NH_4^+$-N ($p < 0.01$, $r = 0.51$). Stepwise MLR models revealed that topsoil SOC and POC were the most effective predictors for explaining the spatial variation in DOC concentrations (Table 4), while catchment slope and $NH_4^+$-N exhibited the highest explanatory power for DOC concentrations when SOC was excluded from the models. Unlike DOC, a significant positive correlation with mean catchment slope was found for $\delta^{13}C_{DOC}$ ($p < 0.001$, $r = 0.76$; Fig. 4). In addition, there was a significant negative correlation between $\delta^{13}C_{DOC}$ and $NO_3^-$-N ($p < 0.001$, $r = -0.75$). Moreover, $\delta^{13}C_{DOC}$ was negatively correlated with DOC concentrations ($p < 0.01$, $r = -0.57$; Fig. 4), but positively correlated with $\delta^{13}C_{POC}$ in these three rivers ($p < 0.05$, $r = 0.51$). Similar to $\delta^{13}C_{DOC}$, $\Delta^{14}C_{DOC}$ was positively related to mean catchment slope ($p < 0.01$, $r = 0.91$) and $\Delta^{14}C_{POC}$ ($p < 0.05$, $r = 0.79$). Additionally, there was a positive correlation between $\Delta^{14}C_{POC}$ and catchment slope ($p < 0.05$, $r = 0.79$), and no significant correlations were detected between $\Delta^{14}C_{POC}$ and the proportion of urban and agricultural land uses or ions that reflect human disturbances (e.g., $Cl^-_{anthro}$, $NH_4^+$-N, and $NO_3^-$-N; $p > 0.05$; Fig. 4).

SUVA$_{254}$ showed an increasing trend with increasing mean catchment slope ($p < 0.001$, $r = 0.77$; Fig. 4). Furthermore, there was a significant negative correlation between SUVA$_{254}$ and the proportion of urban and agricultural land uses ($p < 0.001$, $r = -0.83$; Fig. 4). This is consistent with the constructed stepwise MLR models that urban and agricultural land uses and catchment slope were the best predictors of SUVA$_{254}$ (Table 4). Although no significant correlation was observed between the fluorescence indexes and catchment slope, they (except for FI) were found to be closely related to land use

patterns (Fig. 4). For example, HIX had a positive correlation with urban and agricultural land uses ($p < 0.001$, $r = 0.61$; Fig. 4), while β/α had a negative correlation with urban and agricultural land uses ($p < 0.01$, $r = -0.52$) and water pH ($p < 0.001$, $r = -0.66$). In addition, the fluorescence components did not exhibit significant variations with changing catchment slope ($p > 0.05$; Fig. 4), but the percentage of C1 and C2 were positively ($p < 0.05$, $r = 0.47$) or negatively ($p < 0.01$, $r = -0.55$) related to the proportion of urban and agricultural land uses. Urban and agricultural land uses were also identified as predictors for DOM optical indexes (i.e., HIX; Table 4) and fluorescent components (i.e., C1 and C2). However, unlike C1 and C2, C3 was not significantly correlated with urban and agricultural land uses ($p > 0.05$; Fig. 4), but its variation can be partially explained by $NO_3^-$-N concentrations and POC (Table 4)."

line 348: Figure 6 is of low resolution; the text appears blurry.
**Our response:** Thanks. We have replaced it with a high resolution.

line 383: 2014 should be 2015.
**Our response:** Thanks. We have replaced "2014" to "2015".

line 465: Suggestion: insert "compiled and" before "shown"
**Our response:** Thanks. We have inserted "compiled and" before "shown" as you suggested: "*To provide a deeper insight into the DOC characteristics of the study rivers, DOC concentrations and the carbon isotopes of DOC in global mountainous rivers are compiled and shown in Table 5*." (P20 Line 466-467)

line 499: Regarding the statement on the availability of data, please refer to our data policy (https://www.biogeosciences.net/policies/data_policy.html). Authors are required to provide a statement on how their underlying research data can be accessed. If the data are not publicly accessible, a detailed explanation of why this is the case is required. The best way to provide access to data is by depositing them (as well as related metadata) in FAIR-aligned reliable public data repositories, assigning digital object identifiers, and properly citing data sets as individual contributions.
**Our response:** Thanks. We have make most of the data used in this study availability to the public. We'd like to add the statement on the availability of data in the revised manuscript: "*Data availability*. Water chemistry, isotopes, and DOM properties data used in this study are available online at https://doi.org/10.25442/hku.24433354. Other data are available from the corresponding author Lishan Ran upon request at lsran@hku.hk."